# Greed is Good:
# A Unifying Perspective on Guided Generation

**Zander W. Blasingame**
Clarkson University
`blasinzw@clarkson.edu`

**Chen Liu**
Clarkson University
`cliu@clarkson.edu`

## Abstract

Training-free guided generation is a widely used and powerful technique that allows the end user to exert further control over the generative process of flow/diffusion models. Generally speaking, two families of techniques have emerged for solving this problem for *gradient-based guidance*: namely, *posterior guidance* (*i.e.*, guidance by projecting the current sample to the target distribution via the target prediction model) and *end-to-end guidance* (*i.e.*, guidance by performing backpropagation throughout the entire ODE solve). In this work, we show that these two seemingly separate families can actually be *unified* by looking at the posterior guidance as a *greedy strategy* of *end-to-end guidance*. We explore the theoretical connections between these two families and provide an in-depth theoretical understanding of these two techniques relative to the *continuous ideal gradients*. Motivated by this analysis, we then show a method for *interpolating* between these two families enabling a trade-off between compute and accuracy of the guidance gradients. We then validate this work on several inverse image problems and property-guided molecular generation.

## 1 Introduction

Guided generation greatly extends the utility of state-of-the-art generative models by allowing the end user to exert greater control over the generative process, ultimately making the tool more useful in a wide variety of applications ranging from conditional generation, editing of samples, inverse problems &c. We focus particularly on a subset of neural differential equations that model *affine probability paths*, in other words, diffusion and flow models due to their widespread adoption in a large variety of practical tasks. *E.g.*, audio (H. Liu et al. 2023; Schneider et al. 2024), images (Rombach et al. 2022; Black Forest Labs 2024), biometrics (Blasingame and C. Liu 2024c), molecules (Hoogeboom et al. 2022; Ben-Hamu et al. 2024), proteins (Watson et al. 2023; Skreta et al. 2025), &c.

We can divide the guided generation techniques into two broad categories: conditional training and training-free methods. The former of these two requires the training of the underlying diffusion/flow model on additional conditional information, either as a part of the training or at a later time as additional fine-tuning (Ho and Salimans 2021; J. Song, Meng, and Ermon 2021; Hu et al. 2022). The latter category instead makes use of some known guidance function defined on the data distribution and incorporates this information back to the model to influence the generative process. These training-free techniques can be further broken down into two sub-categories, *i.e.*, posterior and end-to-end guidance. The former class of techniques uses a simple estimation of the posterior distribution that can be easily found in diffusion models (Chung, J. Kim, et al. 2023) and *some* flow models (*cf*. Lipman, Havasi, et al. 2024, Section 4.8). This simple posterior estimate can then be fed into a guidance function to construct a gradient w.r.t. to the current timestep. We refer to this category as *posterior guidance* as they use this posterior estimate to perform the guidance process. This can then be used to update the ODE solve as a form of classifier guidance (Chung, J. Kim, et al. 2023; Yu et al.

39th Conference on Neural Information Processing Systems (NeurIPS 2025).

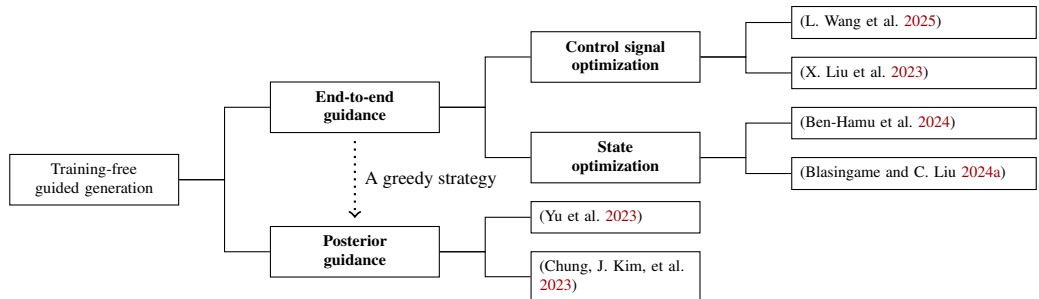

Figure 1: The greedy perspective as a unification of separate families in the taxonomy of training-free guided generation. We provide a more detailed version of this in Figure 5.

2023). The latter class of techniques, in contrast, performs backpropagation throughout the entire sampling process of the flow/diffusion model (Ben-Hamu et al. 2024; Blasingame and C. Liu 2024a). We refer to this category as *end-to-end guidance* as it performs backpropagation throughout the *entire* sampling trajectory.

The aim of this work is to bring these two seemingly disparate family of techniques together into a *single unified view*.

> Our key insight is that we can *bridge* between techniques that use posterior sampling and techniques that use end-to-end optimization for guidance by viewing the former as a *greedy strategy* on the latter.

**Contributions.** In light of this insight, we compare several state-of-the-art techniques from this perspective, showing how this perspective yields a unified and flexible framework for viewing guided generation with flow/diffusion models. We perform a detailed analysis of this greedy strategy, showing that it is not only a unifying view, but that it actually makes *good* decisions under certain scenarios. We then show a perspective which allows one to move between these two classes of guided generation techniques, opening up an exciting and novel design space. Lastly, we conduct some numerical experiments on inverse image problems and molecule generation.

## 2 Preliminaries

Flow models (Lipman, R. T. Q. Chen, et al. 2023) are a highly popular class of generative models that model the generative process as a neural *ordinary differential equation* (ODE) (R. T. Chen et al. 2018). Consider two $\mathbb{R}^d$-valued random variables: $\boldsymbol{X}_0 \sim p(\boldsymbol{x})$ and $\boldsymbol{X}_1 \sim q(\boldsymbol{x})$, denoting the *source* (noise) and *target* (data) distributions, respectively. Then consider a time-dependent vector field $\boldsymbol{u} \in \mathcal{C}^{1,r}([0,1] \times \mathbb{R}^d; \mathbb{R}^d)[1]$ with $r \geq 1$ which determines a time-dependent flow $\Phi_t \in \mathcal{C}^{1,r}([0,1] \times \mathbb{R}^d; \mathbb{R}^d)$ which satisfies the ODE

$$\Phi_0(\boldsymbol{x}) = \boldsymbol{x}, \quad \frac{\mathrm{d}}{\mathrm{d}t}\Phi_t(\boldsymbol{x}) = \boldsymbol{u}(t, \Phi_t(\boldsymbol{x})). \tag{1}$$

This is known as a $\mathcal{C}^r$-flow and this flow is diffeomorphism in its second argument for all $t \in [0,1]$. For notational simplicity let $\boldsymbol{u}_t(\boldsymbol{x}) \mapsto \boldsymbol{u}(t, \boldsymbol{x})$. A special case of flow models are known as *affine probability paths* and are defined as $\boldsymbol{X}_t = \alpha_t \boldsymbol{X}_0 + \sigma_t \boldsymbol{X}_1$ with schedule $(\alpha_t, \sigma_t)$. We provide more details on flow models in Appendix B.1.[2]

---

[1]For notational simplicity, we let $\mathcal{C}^{k_1, k_2, \ldots, k_n}(X_1 \times X_2 \times \cdots \times X_n; Y)$ denote the set of continuous functions that are $k_i$-times differentiable in the $i$-th argument mapping from $(X_1 \times X_2 \times \cdots \times X_n)$ to $Y$, if $Y$ is omitted, then $Y = \mathbb{R}$.

[2]Without loss of generality we consider flow models which subsume the ODE formulation of diffusion models.

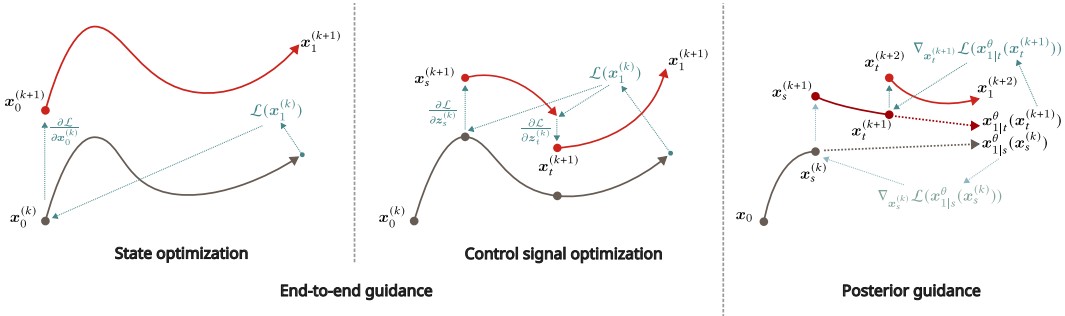

Figure 2: Visual comparison of different training-free guided generation techniques.

# 3 An overview of training-free guidance with gradients

We explore techniques for solving *training-free* guidance problems—this is in contrast with techniques like classifier (Dhariwal and Nichol 2021; Y. Song, Sohl-Dickstein, et al. 2021) and classifier-free (Ho and Salimans 2021) guidance—which use some off-the-shelf guidance function $\mathcal{L} \in \mathcal{C}^1(\mathbb{R}^d)$ defined on the output of the flow model. *I.e.*, we wish to optimize the ODE solve such that the output $\boldsymbol{x}_1$ minimizes $\mathcal{L}$. Suppose we have numerical scheme (Euler, RK4, DPM-Solver, &c.) denoted

$$
\begin{aligned}
&\boldsymbol{\Phi} : \mathbb{R} \times \mathbb{R} \times \mathbb{R}^d \times \mathcal{C}(\mathbb{R} \times \mathbb{R}^d; \mathbb{R}^d) \to \mathbb{R}^d, \\
&\boldsymbol{\Phi}(t_n, t_{n+1}, \boldsymbol{x}_n, \boldsymbol{u}_t^\theta) \mapsto \boldsymbol{x}_{n+1}.
\end{aligned}
\tag{2}
$$

For simplicity we will omit the explicit dependency of the numerical scheme on $\boldsymbol{u}_t^\theta$ and assume it implicitly; likewise, let $\boldsymbol{\Phi}_h(t_n, \cdot, \cdot) \mapsto \boldsymbol{\Phi}(t_n, t_{n+1}, \cdot, \cdot)$ where $h = t_{n+1} - t_n$. We write this objective more formally below in Equation (3).

> **Problem statement.** Given some $t_1 \in [0, 1)$ and step size regime $\{t_1 < t_2 < \ldots < t_N = 1\}$ solve:
>
> $$
> \begin{aligned}
> &\text{Find a sequence} && \{\boldsymbol{x}_n\}_{n=1}^N && \text{which minimizes } \mathcal{L}(\boldsymbol{x}_N), \\
> &\text{subject to} && \boldsymbol{x}_{n+1} = \boldsymbol{\Phi}(t_{n+1}, t_n, \boldsymbol{x}_n).
> \end{aligned}
> \tag{3}
> $$

Next, we will detail two popular families of techniques for solving the problem mentioned above. We illustrate the relationships between these different families in Figure 1, a taxonomy of training-free guidance methods. We note that these two seemingly separate branches can be unified back into a single branch, by the viewing posterior guidance techniques as a greedy strategy of the later. Likewise, we provide a visual overview of the guidance mechanisms in Figure 2.

## 3.1 Posterior guidance

A popular technique for *training-free* guidance is what we will term *posterior guidance* (Chung, J. Kim, et al. 2023; Yu et al. 2023). The key idea behind this strategy is to use the parameterized target prediction model $\boldsymbol{x}_{1|t}^\theta(\boldsymbol{x})$, *i.e.*, the expected value of the posterior distribution given $\boldsymbol{X}_t = \boldsymbol{x}$, to provide a guidance gradient of the form $\nabla_{\boldsymbol{x}} \mathcal{L}(\boldsymbol{x}_{1|t}^\theta(\boldsymbol{x}))$ for some guidance function $\mathcal{L} \in \mathcal{C}^1(\mathbb{R}^d)$. For literature working with score-based generative models (Y. Song, Sohl-Dickstein, et al. 2021), this interpretation arose from the famous Tweedie's formula (Stein 1981; Efron and N. R. Zhang 2011). Thus, for each $\boldsymbol{x}_n$ in the ODE solve, we add guidance to it in the form of posterior guidance gradient.

## 3.2 End-to-end optimization for guidance

Another popular class of techniques is what we will term *end-to-end guidance* (Ben-Hamu et al. 2024; Blasingame and C. Liu 2024a), *i.e.*, techniques which perform guidance by optimizing the initial condition $\boldsymbol{x}_0$ w.r.t. the guidance function $\mathcal{L}$; such techniques require performing backpropagation through a neural ODE. Fittingly, we will import notations and terminology from the study of *neural differential equations* (Kidger 2022) to discuss these techniques. The first technique for performing this kind of guidance is known as *discretize-then-optimize* (DTO) where the numerical

scheme (*cf*. Equation (2)) is part of the computation graph of the model reverse-mode automatic differentiation (Linnainmaa 1976) is applied, *i.e.*, *vanilla backpropagation*. The memory cost of such techniques, however, is $\mathcal{O}(n)$, prompting researchers to explore the second method known as *optimize-then-discretize* (OTD) which instead solves *another* ODE in *reverse-time* which models the continuous-time dynamics of reverse-mode differentiation, this is called the *continuous adjoint method* (R. T. Chen et al. 2018; *cf*. Kidger 2022, Section 5.1.2).

Given a flow model $\boldsymbol{u}_\theta \in \mathcal{C}^{1,1}([0,1] \times \mathbb{R}^d; \mathbb{R}^d)$ that is Lipschitz continuous in its second argument and the solution $\boldsymbol{x} : [0,1] \to \mathbb{R}^d, \boldsymbol{x}_t \mapsto \boldsymbol{x}(t)$, let $\boldsymbol{a}_{\boldsymbol{x}} \coloneqq \partial \mathcal{L} / \partial \boldsymbol{x}_t$ denote the *adjoint state*. Then $\boldsymbol{a}_{\boldsymbol{x}}(t)$ can be found by solving the continuous adjoint equation:

$$\boldsymbol{a}_{\boldsymbol{x}}(1) = \frac{\partial \mathcal{L}}{\partial \boldsymbol{x}_1}, \quad \frac{\mathrm{d}\boldsymbol{a}_{\boldsymbol{x}}}{\mathrm{d}t}(t) = -\boldsymbol{a}_{\boldsymbol{x}}(t)^\top \frac{\partial \boldsymbol{u}_t^\theta}{\partial \boldsymbol{x}}(\boldsymbol{x}_t). \tag{4}$$

*N.B.*, this technique was first proposed by Pontryagin et al. (1963) and popularized for neural differential equations by R. T. Chen et al. (2018). This approach has a constant memory cost $\mathcal{O}(1)$; however, this comes with the cost of several drawbacks related to the numerical scheme. While these issues are not particularly relevant to our theoretical analyses, we note them in Appendix E for the ML practitioner.

## 4   A greedy perspective on guidance

Now returning back to our problem statement from Equation (3), the end-to-end guidance techniques amount to optimizing the initial condition $\boldsymbol{x}_0$ in light of the entire solution trajectory admitted by the numerical scheme. A natural question we consider for problems of this form is that rather than finding the full sequence $\{\boldsymbol{x}_n\}$, can we make use of local information instead? *I.e.*,

> **Key insight 1.** Rather than solving the full ODE from $\boldsymbol{x}_t$, what if we greedily took a locally optimal step at each $\boldsymbol{x}_t$ instead?

Formally, we define a greedy strategy is the following augmentation to the numerical scheme from Equation (2) as

$$\boldsymbol{x}_n^{\mathcal{G}} = \mathcal{G}(t_n, \boldsymbol{x}_n, \boldsymbol{u}_{t_n}^\theta), \tag{5}$$

$$\boldsymbol{x}_{n+1} = \boldsymbol{\Phi}(t_n, t_{n+1}, \boldsymbol{x}_n^{\mathcal{G}}), \tag{6}$$

where $\mathcal{G}$ is the *greedy action* which makes its decision from only information available at time $t_n$.

Now in particular we are interested in a specific greedy action, *i.e.*, posterior guidance. We define this greedy action as the solution to the following iterative process with initial value $\boldsymbol{x}_n^{(0)} = \boldsymbol{x}_n$ which solves

$$\boldsymbol{x}_n^{(k+1)} = \boldsymbol{x}_n^{(k)} - \eta \nabla \mathcal{L}\left(\boldsymbol{x}_{1|t_n}^\theta(\boldsymbol{x}_n^{(k)})\right), \tag{7}$$

for some sufficient number $k > 0$ and learning rate $\eta > 0$.

By construction this greedy action is the popular strategy of posterior guidance. The rest of this section is then devoted to exploring the connections between this greedy action and end-to-end guidance schemes. More, succinctly we state our insight below:

> **Key insight 2.** Posterior guidance can be viewed as Euler schemes within the DTO or OTD backpropagation schemes.

To make our analysis simpler, let us write the flow from $s$ to $t$ in terms of the target prediction model. The flow from time $s$ to time $t$ can then be expressed as the integral of the right-hand side of Equation (19) over time. Thus, the flow is now expressed as a semi-linear integral equation with linear term $a_t \boldsymbol{x}$ and non-linear term $b_t \boldsymbol{x}_{1|t}^\theta(\boldsymbol{x})$. Due to this semi-linear structure, we apply the same technique of *exponential integrators* (Hochbruck and Ostermann 2010) that has been successfully used to simplify numerical solvers for diffusion models (Lu et al. 2022a; Q. Zhang and Y. Chen 2023; Gonzalez et al. 2024). *N.B.*, the full derivations and proofs for this section can be found in Appendix B.

Let $\gamma_t \coloneqq \alpha_t/\sigma_t$ denote the signal-to-noise ratio (SNR), then $\gamma_t$ is a monotonically increasing sequence in $t$, due to the properties of $(\alpha_t, \sigma_t)$ (*cf.* Equation (17)) and thus has an inverse $t_\gamma$ such that $t_\gamma(\gamma(t)) = t$. With abuse of notation, we let $\boldsymbol{x}_\gamma \coloneqq \boldsymbol{x}_{t_\gamma(\gamma)}$ and $\boldsymbol{x}_{1|\gamma}^\theta(\cdot) = \boldsymbol{x}_{1|t_\gamma(\gamma)}^\theta(\cdot)$. As such, we can rewrite the solution to the flow model in terms of $\gamma$ by making use of exponential integrators, which we show in Proposition 4.1 with the full proof provided in Appendix B.3.

> **Proposition 4.1** (Exact solution of affine probability paths). *Given an initial value of $\boldsymbol{x}_s$ at time $s \in [0,1]$ the solution $\boldsymbol{x}_t$ at time $t \in [0,1]$ of an ODE governed by the vector field in Equation (18) is:*
> $$\boldsymbol{x}_t = \frac{\sigma_t}{\sigma_s}\boldsymbol{x}_s + \sigma_t \int_{\gamma_s}^{\gamma_t} \boldsymbol{x}_{1|\gamma}^\theta(\boldsymbol{x}_\gamma)\, \mathrm{d}\gamma. \tag{8}$$

**Remark 4.1.** This result bears some similarity to Lu et al. (2022b, Propostion 5.1); however, they integrate w.r.t. the log-SNR; their result can be recovered, *mutatis mutandis*, with the identity $\lambda_t = \log \gamma_t$.

## 4.1 Greedy guidance as an Euler scheme

Now equipped with this simplified form, we begin to draw connections between end-to-end guidance and our greedy strategy. In Proposition 4.2 we show that the greedy action in Equation (7) can be interpreted as backpropagation via a DTO scheme with an Euler step of size $h = \gamma_1 - \gamma_t$.

> **Proposition 4.2** (Greedy as an explicit Euler scheme within DTO). *For some trajectory state $\boldsymbol{x}_t$ at time $t$, the greedy gradient given by $\nabla_{\boldsymbol{x}}\mathcal{L}(\boldsymbol{x}_{1|t}^\theta(\boldsymbol{x}))$ is the DTO scheme with an explicit Euler discretization with step size $h = \gamma_1 - \gamma_t$.*

Now we examine greedy action from the perspective of an OTD scheme. In Proposition 4.3 we show that a greedy strategy can be viewed as the first iteration of a fixed-point method of an implicit Euler discretization of the continuous adjoint equations.

> **Proposition 4.3** (Greedy as an implicit Euler scheme within OTD). *For some trajectory state $\boldsymbol{x}_t$ at time $t$, the greedy gradient given by $\nabla_{\boldsymbol{x}_t}\mathcal{L}(\boldsymbol{x}_{1|t}^\theta(\boldsymbol{x}_t))$ is an implicit Euler discretization of the continuous adjoint equations for the true gradients with step size $h = \gamma_1 - \gamma_t$.*

*Proof sketch.* First, we use the technique of exponential integrators to simplify the continuous adjoint equations. Then we perform a first-order Taylor expansion around $\gamma_t$, which is equivalent to an implicit Euler scheme, as we calculate the gradient flow from $1$ to $t$. The full proof is provided in Appendix B.5. □

## 5 Is greed good?

A natural question to ask in light of this discussion on taking this greedy action is why even bother backpropagating through the ODE solve at all for guidance? After all, we could simply run the optimization process directly in the data space (*cf.* Equation (7)). So why perform end-to-end guidance or this greedy action at all? *N.B.*, the full derivations and proofs for this section may be found in Appendix C.

We begin by examining the structure of the gradient $\nabla_{\boldsymbol{x}}\mathcal{L}(\Phi_{t,1}^\theta(\boldsymbol{x}))$. By the chain rule we observe the following:[3]
$$\nabla_{\boldsymbol{x}}\mathcal{L}\left(\Phi_{t,1}^\theta(\boldsymbol{x})\right) = \nabla_{\boldsymbol{x}}\Phi_{t,1}^\theta(\boldsymbol{x})^\top \nabla_{\boldsymbol{x}_1}\mathcal{L}\left(\Phi_{t,1}^\theta(\boldsymbol{x}_1)\right). \tag{9}$$

The question then is what is the behavior of $\nabla_{\boldsymbol{x}}\Phi_{t,1}^\theta(\boldsymbol{x})$? We answer this in Theorem 5.1 below, providing an integral equation for $\nabla_{\boldsymbol{x}}\Phi_{s,t}^\theta(\boldsymbol{x})$.

---

[3]Let $\nabla_{\boldsymbol{x}_1}$ be shorthand for the gradient w.r.t. the output $\Phi_{t,1}^\theta(\boldsymbol{x})$.

**Theorem 5.1** (Jacobian matrices of affine Gaussian probability paths). *For the standard affine Gaussian probability path with flow model $\Phi_{s,t}^{\theta}(\boldsymbol{x})$, the Jacobian matrix $\nabla_{\boldsymbol{x}}\Phi_{s,t}(\boldsymbol{x})$ as function of $\boldsymbol{x}$ is given as the solution to*

$$\nabla_{\boldsymbol{x}}\Phi_{s,t}^{\theta}(\boldsymbol{x}) = \frac{\sigma_t}{\sigma_s}\boldsymbol{I} + \sigma_t\int_s^t \dot{\gamma}_u\frac{\gamma_u}{\sigma_u}\text{Var}_{1|u}(\Phi_{s,u}^{\theta}(\boldsymbol{x}))\nabla_{\boldsymbol{x}}\Phi_{s,u}^{\theta}(\boldsymbol{x})\,\mathrm{d}u, \tag{10}$$

*where*

$$\text{Var}_{1|t}(\boldsymbol{x}) = \mathbb{E}_{p_{1|t}(\boldsymbol{x}_1|\boldsymbol{x})}\left[(\boldsymbol{x}_1 - \boldsymbol{x}_{1|t}^{\theta}(\boldsymbol{x}))(\boldsymbol{x}_1 - \boldsymbol{x}_{1|t}^{\theta}(\boldsymbol{x}))^{\top}\right]. \tag{11}$$

**Remark 5.1.** From Theorem 5.1 we observe the Jacobian-vector product $\nabla_{\boldsymbol{x}}\Phi_{s,t}^{\theta}(\boldsymbol{x})^{\top}\boldsymbol{v}$ corresponds to an integral of covariance projections applied to $\boldsymbol{v}$.[4]

Thus, we see that the continuous-time backpropagation process through the flow model is a projection of the loss by a covariance matrix into the directions of highest variance, *i.e.*, the guidance encourages the state to evolve within states on the data manifold. We elaborate on this more in Appendix C.2. While this is a nice observation we cannot solve such an integral in practice. What about our greedy strategy, how does it impact the loss function?

## 5.1 Dynamics of gradient guidance

We now consider how the output of the flow model will change under greedy guidance. In particular, we are interested in how $\Phi_{t,1}^{\theta}(\boldsymbol{x})$ changes under the following gradient step

$$\boldsymbol{x}' = \boldsymbol{x} - \eta\nabla_{\boldsymbol{x}}\mathcal{L}\left(\boldsymbol{x}_{1|t}^{\theta}(\boldsymbol{x})\right). \tag{12}$$

To do this, we make use of the Gateaux differential (Gâteaux 1913) which allows us to define the differential that describes how the output of the flow model $\boldsymbol{x}_1$ evolves with changes to $\boldsymbol{x}$ at time $t$. We present the result to this question in Proposition 5.2 below.

**Proposition 5.2** (Dynamics of greedy gradient guidance). *Consider the standard affine Gaussian probability paths model trained to zero loss. The Gateaux differential of $\boldsymbol{x}$ at some time $t \in [0,1]$ in the direction of the gradient $\nabla_{\boldsymbol{x}}\mathcal{L}\left(\boldsymbol{x}_{1|t}^{\theta}(\boldsymbol{x})\right)$ is given by*

$$\delta_{\boldsymbol{x}}^{\mathcal{G}}\Phi_{t,1}^{\theta}(\boldsymbol{x}) = -\nabla_{\boldsymbol{x}}\Phi_{t,1}^{\theta}(\boldsymbol{x})\nabla_{\boldsymbol{x}}\boldsymbol{x}_{1|t}^{\theta}(\boldsymbol{x})^{\top}\nabla_{\boldsymbol{x}_1}\mathcal{L}(\boldsymbol{x}_1). \tag{13}$$

**Remark 5.2.** Recall that from Theorem 5.1 and (Ben-Hamu et al. 2024, Proposition 4.1) we know that both $\nabla_{\boldsymbol{x}}\Phi_{t,1}^{\theta}(\boldsymbol{x})$ and $\nabla_{\boldsymbol{x}}\boldsymbol{x}_{1|t}^{\theta}(\boldsymbol{x})$ consist of covariance matrices, thus the dynamics of greedy gradient guidance are governed by this covariance projection of the loss.

Next, we ask what is the difference between the *idealized* gradient $\nabla_{\boldsymbol{x}}\Phi_{t,1}^{\theta}(\boldsymbol{x})$ and the greedy gradient $\nabla_{\boldsymbol{x}}\boldsymbol{x}_{1|t}^{\theta}(\boldsymbol{x})$? Intuitively, we find that it is bound by the local truncation error, *i.e.*, $\mathcal{O}(h^2)$ which we show below.

**Theorem 5.3** (Dynamics of gradient vs greedy guidance). *The difference between the dynamics of gradient guidance in Proposition C.4 and greedy gradient guidance in Proposition 5.2 for a point $\boldsymbol{x}$ at time $t$ with guidance function $\mathcal{L} \in \mathcal{C}^1(\mathbb{R}^d)$ is bounded by $\mathcal{O}(h^2)$ where $h := \gamma_1 - \gamma_t$, i.e.,*

$$\left\|\nabla_{\boldsymbol{x}}\Phi_{t,1}^{\theta}(\boldsymbol{x}) - \nabla_{\boldsymbol{x}}\boldsymbol{x}_{1|t}^{\theta}(\boldsymbol{x})\right\| = \mathcal{O}(h^2). \tag{14}$$

An important question is whether a greedy strategy makes *good* decisions at each timestep. *I.e.*, if we make a good decision at time $t$, does that ensure that an optimal solution was made in the sense of $\Phi_{1|t}^{\theta}(\boldsymbol{x}_t)$. A natural way to examine this question is to consider whether convergence in the local case implies convergence of the whole solution trajectory. We find that up to a bound dependent on the

---

[4]Readers familiar with the work of Ben-Hamu et al. (2024) may notice some similarities between our result Theorem 5.1 and Ben-Hamu et al. (2024, Theorem 4.2). We discuss this more in Remark C.3.

step size, convergence in the greedy solution implies convergence in the flow, which we state more formally in Theorem 5.4.

> **Theorem 5.4** (Greedy convergence). *For affine probability paths, if there exists a sequence of states $\boldsymbol{x}_t^{(n)}$ at time $t$ such that it converges to the locally optimal solution $\boldsymbol{x}_{1|t}^{\theta}(\boldsymbol{x}_t^{(n)}) \to \boldsymbol{x}_1^*$. Then the solution, $\Phi_{1|t}^{\theta}(\boldsymbol{x}_t^{(n)})$, converges to a neighborhood of size $\mathcal{O}(h^2)$ centered at $\boldsymbol{x}_1^*$.*

## 6 Beyond Euler

Motivated by this connection between the powerful, but expensive, end-to-end guidance techniques and posterior guidance techniques, we ask is there a middle-ground between them? A natural extension would be to consider something beyond the Euler scheme from the previous section, *e.g.*, applying the midpoint method or two Euler steps. To motivate this discussion more rigorously we present Theorem 6.1, which shows that for any explicit single-step Runge-Kutta solver, the error between the *ideal* gradient and this estimated gradient is on the order of the local truncation error of the underlying numerical solver.

> **Theorem 6.1** (Truncation error of single-step gradients). *Let $\boldsymbol{\Phi}$ be an explict Runge-Kutta solver of order $\alpha > 0$ of a flow model with flow $\Phi_{s,t}^{\theta}(\boldsymbol{x})$. Then for any $t \in [0, 1]$,*
>
> $$\left\| \nabla_{\boldsymbol{x}} \Phi_{t,1}^{\theta}(\boldsymbol{x}) - \nabla_{\boldsymbol{x}} \boldsymbol{\Phi}_{t,1}(\boldsymbol{x}) \right\| = \mathcal{O}(h^{\alpha+1}), \tag{15}$$
>
> *where $h = 1 - t$.*

> **Key insight 3.** We can use a higher-order solver to move between posterior and end-to-end guidance exchanging compute and gradient accuracy.

This theoretical tool enables us to move between posterior and full end-to-end guidance choosing whichever point between compute and accuracy happens to be most suitable, hopefully opening a larger design space for solving interesting problems. Additional discussions and the full derivations are found in Appendix D.

## 7 Experiments

Motivated by the theoretical connections from the previous sections we apply the greedy posterior strategy (Euler) to several problems using flow/diffusion models, as well as several methods lying in the in between space of end-to-end guidance and posterior guidance, namely, a single-step midpoint scheme and 2-step Euler scheme.

### 7.1 Inverse problems for images

A common application of posterior guidance has been in solving inverse problems (Y. Song, Sohl-Dickstein, et al. 2021; Chung, Sim, and Ye 2022) (*cf*. Appendix H). As such, we explore several inverse problems in the image domain. In particular, we explore a set of inverse image problems on a subset of 100 images from the FFHQ (Karras, Laine, and Aila 2019) $256 \times 256$ dataset. We make use of the pre-trained diffusion model from Chung, J. Kim, et al. (2023) trained on the FFHQ dataset.

**Inverse problems and metrics.** Following (B. Zhang et al. 2025) we conduct experiments on the following linear tasks: super resolution, Gaussian deblurring, motion deblurring, inpainting (with a box mask), and inpainting (with a 70% random mask); along with three non-linear problems: phase retrieval, high dynamic range (HDR) reconstruction, and non-linear deblurring. We use the standard evaluation metrics of *peak signal-to-noise-ratio* (PSNR), *structural similarity index measure* (SSIM), *Learned Perceptual Image Patch Similarity* (LPIPS) (R. Zhang et al. 2018), and *Fréchet Inception Distance* (FID) (Heusel et al. 2017). We solve the probability flow ODE with the midpoint scheme and 20 discretization steps; further configuration details are reported in Appendix I.1.

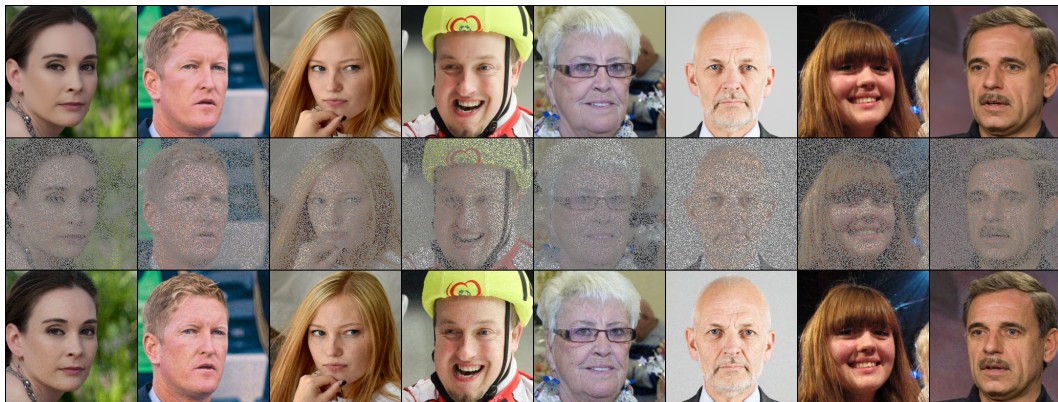

Figure 3: Qualitative visualization of using posterior guidance to solve an inverse problem on the task of inpainting with a 70% random mask. Top row is the ground truth, middle row is the measurement, and the bottom row is the reconstruction.

Table 1: A snapshot of the quantitative results for solving inverse image problems on FFHQ. We report the mean performance (PSNR, SSIM, and LPIPS) across 100 validation images along with the FID. All tasks are using a noisy measurement with noise level $\beta_{\boldsymbol{y}} = 0.05$. The full results are found in Table 6.

| Task | Method | PSNR ($\uparrow$) | SSIM ($\uparrow$) | LPIPS ($\downarrow$) | FID ($\downarrow$) |
|---|---|---|---|---|---|
| Inpaint (random) | Greedy (Euler) | 30.87 | 0.823 | 0.141 | 40.73 |
| | Greedy (midpoint) | 31.03 | 0.816 | 0.139 | 38.80 |
| | Greedy (2-step Euler) | 30.80 | 0.811 | 0.144 | 39.23 |
| | DAPS | 31.12 | 0.844 | 0.098 | 32.17 |
| | DPS | 25.46 | 0.823 | 0.203 | 69.20 |
| Gaussian deblurring | Greedy (Euler) | 28.01 | 0.766 | 0.182 | 57.04 |
| | Greedy (midpoint) | 28.36 | 0.776 | 0.185 | 58.55 |
| | Greedy (2-step Euler) | 28.18 | 0.774 | 0.181 | 57.18 |
| | DAPS | 29.19 | 0.817 | 0.165 | 53.33 |
| | DPS | 25.87 | 0.764 | 0.219 | 79.75 |

**Results.** We present some qualitative results on reconstructing images from a random mask in Section 7. Quantitatively, we present a snapshot of our full results (*cf*. Table 6) on the inpainting with random mask and Gaussian deblurring tasks. For reference we include the standard DPS (Chung, J. Kim, et al. 2023) and the recent state-of-the-art DAPS (B. Zhang et al. 2025). We observe that the posterior guidance strategy works well performing closer to DAPS than DPS. Interestingly, on these tasks the extra compute and smaller truncation error of the midpoint and 2-step Euler did not lead to any noticeable performance gains. We report further results in Appendix J.2 along with additional analysis and discussion.

**Ablations on discretization steps.** As we discussed in Section 6 we can improve performance by taking more step sizes. We preform a more involved ablation of this design axis on the *high-dynamic range* (HRD) reconstruction experiment detailed in Table 2. Additionally, we also report the results from the RED-diff (Mardani et al. 2024) algorithm. Following B. Zhang et al. (2025) for the *non-linear* inverse problem of HDR reconstruction we perform 4 runs per algorithm and report the mean and standard deviation. We notice that the Greedy (5-step Euler) performs very well beating the SOTA DAPS algorithm, even the 2-step and 3-step perform variants as well or better than DAPs on this problem. Increasing the number of discretization steps leads to better performance (*cf*. Theorem 6.1). Interestingly, the standard deviation decreases as well with the results becoming more consistent. We also compared to a full DTO run with vary step sizes, *i.e.*, end-to-end optimization with a vary number of steps used in calculating the gradient (the full 20 are used for the final sampling). We do observe a similar trend of increasing performance as we increase the number of steps, however, it far under-performs the greedy strategy for a similar compute budget.

Table 2: Further ablations on the number of discretization steps on the non-linear HDR inverse problem.

| Method | PSNR ($\uparrow$) | SSIM ($\uparrow$) | LPIPS ($\downarrow$) | FID ($\downarrow$) |
|---|---|---|---|---|
| DAPS | $27.12_{\pm 3.53}$ | $0.752_{\pm 0.041}$ | $0.162_{\pm 0.072}$ | 42.97 |
| DPS | $22.73_{\pm 6.07}$ | $0.591_{\pm 0.141}$ | $0.264_{\pm 0.156}$ | 112.82 |
| RED-diff | $22.16_{\pm 3.41}$ | $0.512_{\pm 0.083}$ | $0.258_{\pm 0.089}$ | 108.32 |
| | | | | |
| Greedy (Euler) | $25.07_{\pm 4.25}$ | $0.776_{\pm 0.126}$ | $0.173_{\pm 0.070}$ | 43.25 |
| Greedy (2-step Euler) | $26.32_{\pm 4.34}$ | $0.802_{\pm 0.111}$ | $0.173_{\pm 0.065}$ | 38.64 |
| Greedy (3-step Euler) | $27.17_{\pm 4.21}$ | $0.820_{\pm 0.096}$ | $0.154_{\pm 0.062}$ | 36.07 |
| Greedy (4-step Euler) | $27.89_{\pm 4.10}$ | $0.828_{\pm 0.092}$ | $0.151_{\pm 0.061}$ | 36.94 |
| Greedy (5-step Euler) | $\mathbf{28.27}_{\pm 4.01}$ | $\mathbf{0.831}_{\pm 0.088}$ | $\mathbf{0.149}_{\pm 0.059}$ | **35.35** |
| | | | | |
| DTO (1-step) | $13.16_{\pm 1.15}$ | $0.372_{\pm 0.083}$ | $0.521_{\pm 0.059}$ | 108.39 |
| DTO (2-step) | $14.91_{\pm 1.23}$ | $0.372_{\pm 0.080}$ | $0.483_{\pm 0.061}$ | 98.93 |
| DTO (4-step) | $16.37_{\pm 1.38}$ | $0.455_{\pm 0.082}$ | $0.457_{\pm 0.066}$ | 93.52 |
| DTO (8-step) | $16.37_{\pm 1.38}$ | $0.455_{\pm 0.082}$ | $0.457_{\pm 0.066}$ | 93.52 |

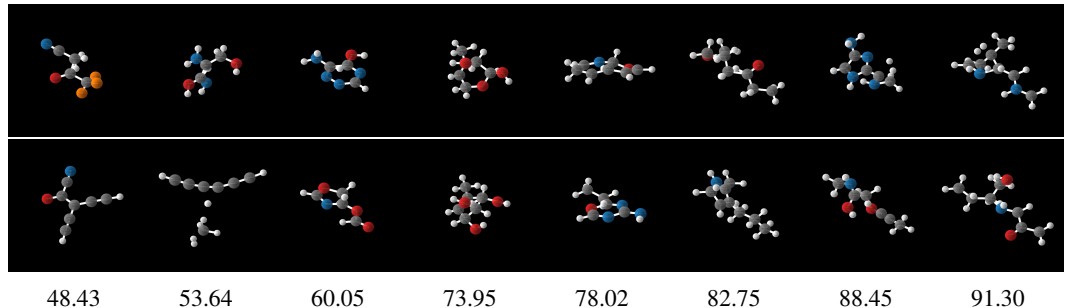

| 48.43 | 53.64 | 60.05 | 73.95 | 78.02 | 82.75 | 88.45 | 91.30 |

Figure 4: Qualitative visualization of controlled generated molecules for various polarizability ($\alpha$) levels. Top row is generated using end-to-end guidance with a DTO scheme and the bottom row is generated using greedy guidance.

## 7.2 Molecule generation for QM9

We also illustrate the core ideas with some experiments in controllable molecule generation on the QM9 dataset (Ruddigkeit et al. 2012), a popular molecular dataset containing small molecules with up to 29 atoms. Following Hoogeboom et al. (2022) and Ben-Hamu et al. (2024), we perform the conditional generation of molecules with specified quantum chemical property values. In particular, we target the following properties: polarizability $\alpha$, orbital energies $\varepsilon_{\mathrm{HOMO}}, \varepsilon_{\mathrm{LUMO}}$ and their gap $\Delta\varepsilon$, dipole moment $\mu$, and heat capacity $C_v$. The property classifiers were trained following the methodology outlined in Hoogeboom et al. (2022). The underlying flow model is an unconditional equivariant flow matching model with *conditional optimal transport* path (Lipman, Havasi, et al. 2024, Section 4.7; *cf.* Tong, Malkin, et al. 2023; Tong, FATRAS, et al. 2024), *i.e.*, the EquiFM (Y. Song, Gong, et al. 2023) model. We solve the ODE with Euler's method and 50 discretization steps; further configuration details are reported in Appendix I.1. Further details are provided in Appendix I.2.

**Metrics.** To evaluate the guided generation we calculate the *mean absolute error* (MAE) between the predicted property value of the generated molecule by the property classifier and the target property value (Satorras et al. 2021). Additionally in Appendix J.1 we report the quality of the generated molecules by evaluating the atom stability (the percentage of atoms with correct valency) and molecule stability (the percentage of molecules where all atoms are stable).

**Results.** In Section 7.1 we present a visual comparison between molecules generated targeting different polarizability $\alpha$ values using a DTO end-to-end guidance scheme (essentially D-Flow) and the posterior guidance scheme. Notice that as $\alpha$ increases the compactness of the molecules

Table 3: Quantitative evaluation of conditional molecule generation. The MAE is reported for each molecule property (lower is better).

| Property | $\alpha$ | $\Delta\varepsilon$ | $\varepsilon_{\mathrm{HOMO}}$ | $\varepsilon_{\mathrm{LUMO}}$ | $\mu$ | $C_v$ |
| Unit | $\mathrm{Bohr}^2$ | meV | meV | meV | D | $\frac{\mathrm{cal}}{\mathrm{K \cdot mol}}$ |
|---|---|---|---|---|---|---|
| Greedy (Euler) | 11.282 | 1265 | 725 | 1092 | 1.559 | 6.469 |
| Greedy (midpoint) | 5.313 | 1196 | 599 | 1057 | 1.417 | 2.967 |
| Greedy (2-step Euler) | 5.667 | 1205 | 695 | 1222 | 1.491 | 2.767 |
| Greedy (3-step Euler) | 5.098 | 1152 | 600 | 1152 | 1.384 | 3.229 |
| Greedy (5-step Euler) | 4.177 | 1083 | 571 | 939 | 1.328 | 2.332 |
| | | | | | | |
| DTO (1-step) | 13.049 | $989 \times 10^{12}$ | 681 | 86.512 | 1.666 | 15.144 |
| DTO (2-step) | 6.113 | 1359 | 666 | 1199 | 1.533 | 3.757 |
| DTO (4-step) | 6.115 | 1294 | 668 | 1190 | 1.406 | 2.829 |
| DTO (8-step) | 4.549 | 1070 | 608 | 1078 | 1.247 | 2.594 |
| DTO (16-step) | 3.454 | 817 | 608 | 939 | 1.177 | 2.003 |
| DTO (32-step) | 2.912 | 750 | 410 | 666 | 0.721 | 1.566 |
| DTO (40-step) | 2.384 | 625 | 372 | 556 | 0.719 | 1.425 |
| DTO (50-step) | 1.404 | 401 | 176 | 373 | 0.372 | 0.866 |
| EquiFM | 9.525 | 1494 | 622 | 1523 | 1.628 | 6.689 |
| Lower bound | 0.10 | 64 | 39 | 46 | 0.043 | 0.040 |

generated by a DTO scheme decreases. This trend is less noticeable for the posterior guided samples. We report quantitative results in Table 3. We report the unguided EquiFM generated molecules as an upper bound and include the theoretical lower bounds from Ben-Hamu et al. (2024). It is here that we notice a sharp decrease in performance from using posterior guidance. In particular the greedy (Euler) strategy is is highly unstable even performing worse than the unguided model on the $\alpha$ property. The introduction of an additional step in the form of either midpoint or 2-step Euler does seem to improve performance; although the significance varies property to property. We observe that the midpoint method seems to perform slightly better than the 2-step Euler. We performed experiments with Ralston's third-order method and a fourth-order Runge-Kutta scheme but noticed significant instability in comparison to just taking more steps, we posit this is due to the large step size and that a hybrid scheme like that employed by Moufad et al. (2025) might be a reasonable solution to such problems, but ultimately we leave that question up to future work. Moreover, we observe that increasing the number of steps generally improves performance with greedy (5-step Euler) performing the best among all the greedy guidance strategies. The gradients of DTO (50-step) strategy are the *ideal* and perfect gradients w.r.t. the flow model as the numerical solver takes 50 discretization steps and thus form the *upper bound of performance*. We see that as the DTO strategy incorporates more discretization steps the results converge to the upper bound along this particular design axis; in particular, we notice that the greedy strategy does well in the the regime of less discretization steps.

## 8    Conclusion

In this paper we present a unifying view of two different families of guided generation: end-to-end guidance and posterior guidance from the lens of a greedy algorithm. We present numerous theoretical connections tying these two families together. Our theoretical analysis shows that there might be some reason to believe that such a cheap approximation of the gradient can be reasonable for *certain* tasks. By exploiting the theoretical connections we created, we investigate guidance techniques which lie in between these two families giving rise to an exciting novel design space. We then conduct several experiments on inverse image problems and on controlled molecule generation to illustrate this new design space. We hope that our findings can help future researchers find the optimal spot between computational cost and accuracy of gradients for guidance problems.

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

## Organization of the appendix

In Appendix A we discuss previous approaches by exploring posterior guidance and end-to-end guidance in greater detail to provide a more comprehensive overview of how this greedy perspective connects these various works. Appendix B is devoted to the proofs and derivations from Section 4 in the main paper. Likewise, Appendices C and D is devoted to proofs and derivations from Sections 5 and 6 respectively. In Appendix E we discuss some important practical issues when using OTD for guidance, which we believe several to be useful background for the reader. We provide some additional connections between posterior guidance and control signal optimization in Appendix F that we were unable to include in the main paper. Appendix H is devoted to providing a brief background on inverse problems. Likewise, Appendix I is devoted to discussing the implementation details of the numerical experiments in Section 7 and providing a background for the experiments. In Appendix J we include additional results that we could not fit into the main paper. Lastly, in Appendix K we discuss the limitations and broader impacts of this research.

## Appendices

## Overview of theoretical results

For convenience we provide a list of theorems to make navigating the theoretical results easier.

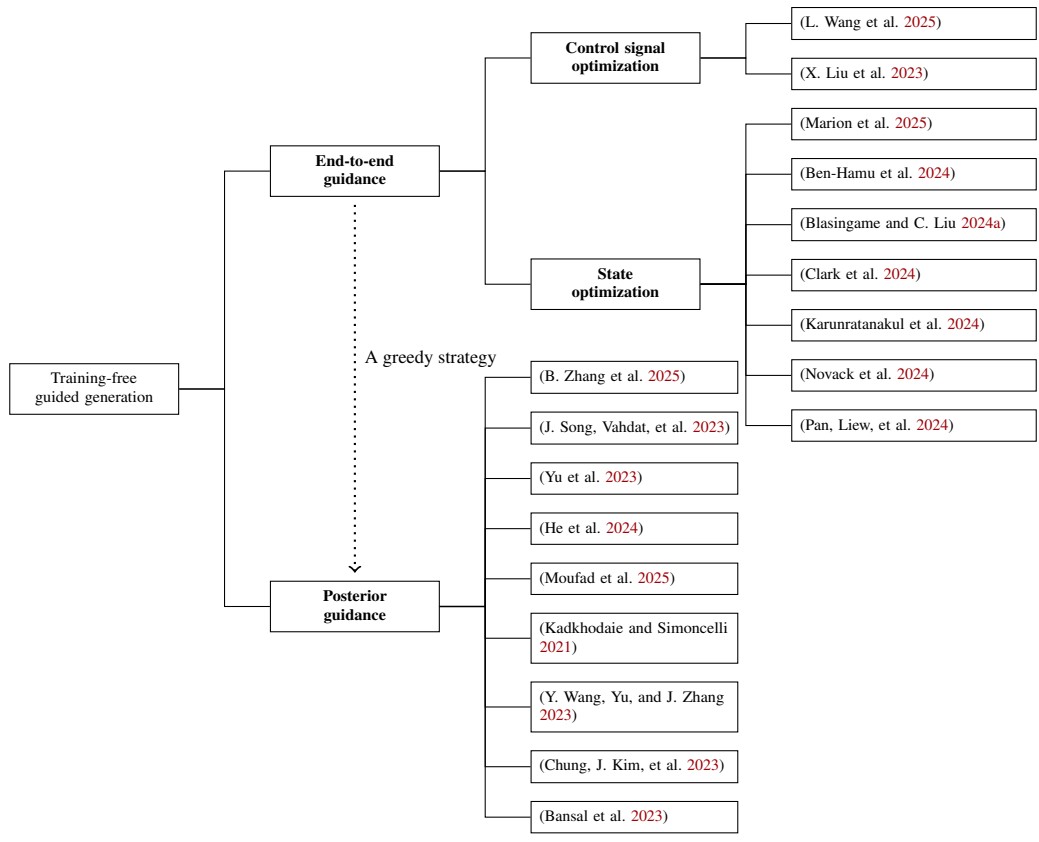

Figure 5: A more detailed taxonomy of *training-free guided generation* methods from Figure 1 from the main paper.

# A   Related works

We provide a brief summary of previous work exploring either posterior guidance or end-to-end guidance strategies. In Figure 5 we provide a more detailed taxonomy of training-free methods for gradient-based guided generation based on Figure 1 from the main paper.

## A.1   Posterior guidance

Recent work in flow/diffusion models has explored the guidance using this strategy; we highlight a few notable examples. Diffusion Posterior Sampling (DPS) (Chung, J. Kim, et al. 2023) is a guidance method that uses Tweedie's formula (Stein 1981) to estimate the gradient of some guidance function defined in the output state w.r.t. the noisy state, *i.e.*, $\mathbb{E}[\boldsymbol{X}_1 | \boldsymbol{X}_t = \boldsymbol{x}]$. Likewise, the work of Bansal et al. (2023), He et al. (2024), Y. Wang, Yu, and J. Zhang (2023), and Yu et al. (2023) explores similar concepts by employing Tweedie's formula for diffusion models. Most of these works have explored using the SDE (or Markov chain) formulation of diffusion models rather than the ODE formulation, which is what we primarily focused on in our analysis.

**Correcting the guidance trajectory.**    Several works have explored extensions to the DPS framework by using multiple steps of an SDE solver to correct *errors* made by the guidance steps. In particular, FreeDoM (Yu et al. 2023) explores the usage of a *time-reversal* strategy repeated for a set number of times in each sampling step to correct possible guidance errors. Likewise, recent work by B. Zhang et al. (2025) explored modeling Langevin dynamics on top of a diffusion ODE to correct measurement errors in inverse problems. A significant number of the proposed methods which use posterior guidance arise from solving inverse problems (*cf*. Daras et al. 2024).

**Scheduled hyperparameters.** Researchers realized that extra performance can be gained in such problems by scheduling hyperparameters like the learning rate (or guidance strength) at different timesteps in the numerical scheme (Moufad et al. 2025; Yu et al. 2023).

**Beyond Euler.** Recent work by Moufad et al. (2025) explores an extension to (Chung, J. Kim, et al. 2023) by using a two-step method to estimate the guidance gradient. This is mostly closely related to the *greedy (2-step Euler)* method from the main paper, although they use a stochastic sampling method, so it would be more akin to taking two Euler-Maruyama steps.

### A.2 End-to-end guidance

Within the last year, many researchers have explored backpropagation through flow/diffusion models for controllable generation. As mentioned in the main paper, the two main strategies for solving such a problem is a DTO or OTD scheme (*cf*. Appendix E).

**Discretize-then-optimize.** FlowGrad proposed by X. Liu et al. (2023) uses a DTO scheme to optimize an additional control signal (more details on this later) to perform guidance with flow models. Although the analysis of Ben-Hamu et al. (2024) makes use of the continuous adjoint equations, in practice they use the *generally* preferred approach of DTO with gradient checkpointing.[5] Likewise, Clark et al. (2024), Karunratanakul et al. (2024), and Novack et al. (2024) all use gradient checkpointing with DTO to perform backpropagation through the flow/diffusion model.

**Optimize-then-discretize.** Another stream of work has explored the use of continuous adjoint equations to perform the backpropagation. The advantage of such approaches is the $\mathcal{O}(1)$ memory cost, and we enumerate the drawbacks in Appendix E, but suffice to say there are several. To the best of our knowledge, the first work to explore this was Nie et al. (2022) which used OTD with SDEs for the adversarial purification task. More general work came later by Ben-Hamu et al. (2024), Blasingame and C. Liu (2024a), and Pan, Liew, et al. (2024). More specifically, Pan, Liew, et al. (2024) and Pan, Yan, et al. (2023) explore bespoke solvers for the continuous adjoint equations of diffusion ODEs. Blasingame and C. Liu (2024a) extends these works by developing bespoke solvers for diffusion ODEs and SDEs and performs more theoretical analysis of the problem in the SDE setting. Marion et al. (2025) explore using the continuous adjoint equations as a part of a larger bi-level optimization scheme for guided generation. The work of Ben-Hamu et al. (2024) extends the analysis of continuous adjoint equations for diffusion models to flow-based models and provides an alternative perspective to the analysis performed in the earlier works. Recent work by L. Wang et al. (2025) explores an extension of Ben-Hamu et al. (2024) to Riemannian manifolds which incorporates a control signal to the vector field and optimizes both the solution state and *co-state*, they call their approach OC-Flow.

Parallel to these works (conceptually) is the work of Wallace, Gokul, Ermon, et al. (2023) who uses EDICT (Wallace, Gokul, and Naik 2023), an invertible formulation of diffusion models, to perform backpropagation through the diffusion model. Although not presented or viewed this way in the original work, the later work by Blasingame and C. Liu (2024a) showed that this approach can be viewed as a specific discretization scheme of continuous adjoint equations. We note that the EDICT solver, while reversible, is a zeroth-order solver and has poor convergence properties (*cf*. F. Wang et al. 2024).

**Control signal optimization.** We discuss this in more detail in Appendix F, but there are several works that explore the optimization of an additional control signal $z(t)$ rather than the solution trajectory $x(t)$; namely, X. Liu et al. (2023) and L. Wang et al. (2025).

## B  A greedy perspective

We present the proofs and derivations associated with Section 4.

---

[5]See https://docs.kidger.site/diffrax/api/adjoints/ for an excellent summary of such design considerations and why DTO is generally preferable over OTD.

## B.1 Additional details on flow models

Applying this flow to the random variable $\boldsymbol{X}_0$ we define a *continuous-time Markov process* $\{\boldsymbol{X}_t\}_{t \in [0,1]}$ with mapping $\boldsymbol{X}_t = \Phi_t(\boldsymbol{X}_0)$. The *goal*, then, is to learn a flow $\Phi_t$ such that $\boldsymbol{X}_1 = \Phi_1(\boldsymbol{X}_0) \sim q(\boldsymbol{x})$. This procedure amounts to learning a neural network parameterized vector field $\boldsymbol{u}^\theta \in \mathcal{C}^{1,r}([0,1] \times \mathbb{R}^d; \mathbb{R}^d)$; this learning procedure can be performed efficiently through a *simulation-free* training process known as *flow matching* (Lipman, R. T. Q. Chen, et al. 2023) or more generally *generator matching* (Holderrieth et al. 2025).

Throughout the rest of this paper we will assume a standard flow model trained to zero loss and we denote the parameterized flow model via $\Phi_t^\theta(\boldsymbol{x})$. We let $\Phi_{s,t}(\boldsymbol{x}) = (\Phi_t \circ \Phi_s^{-1})(\boldsymbol{x})$ denote the flow from time $s$ to time $t$, $s, t \in [0,1]$.

**Affine probability paths.** A special subset of flow models, are flows which model an *affine probability path*, *i.e.*, given a schedule $(\alpha_t, \sigma_t)$ the random process $\{\boldsymbol{X}_t\}$ is described via the affine equation

$$\boldsymbol{X}_t = \alpha_t \boldsymbol{X}_1 + \sigma_t \boldsymbol{X}_0, \tag{16}$$

where $\alpha_t, \sigma_t \in \mathcal{C}^\infty([0,1]; [0,1])$ which satisfy

$$\alpha_0 = \sigma_1 = 0, \quad \alpha_1 = \sigma_0 = 1, \quad \forall t \in (0,1) \ [\dot{\alpha}_t > 0, \ \dot{\sigma}_t < 0]. \tag{17}$$

The *marginal vector field* can then be expressed as the following conditional expectation:

$$\boldsymbol{u}_t(\boldsymbol{x}) = \mathbb{E}[\dot{\alpha}_t \boldsymbol{X}_1 + \dot{\sigma}_t \boldsymbol{X}_0 | \boldsymbol{X}_t = \boldsymbol{x}]. \tag{18}$$

This *nice* form of the marginal vector field enables use to rewrite the vector field in the forms of either source (Ho, Jain, and Abbeel 2020) or target (Kingma et al. 2021) prediction as

$$\boldsymbol{u}_t(\boldsymbol{x}) = \underbrace{\frac{\dot{\beta}_t}{\beta_t}}_{=a_t} \boldsymbol{x} + \underbrace{\frac{\sigma_t \dot{\alpha}_t - \dot{\sigma}_t \alpha_t}{\beta_t}}_{=b_t} \boldsymbol{f}_t(\boldsymbol{x}), \tag{19}$$

where $\beta_t = -\alpha_t$ for source prediction with $\boldsymbol{f}_t(\boldsymbol{x}) = \boldsymbol{x}_{0|t}(\boldsymbol{x}) = \mathbb{E}[\boldsymbol{X}_0 | \boldsymbol{X}_t = \boldsymbol{x}]$ and $\beta_t = \sigma_t$ for target prediction with $\boldsymbol{f}_t(\boldsymbol{x}) = \boldsymbol{x}_{1|t}(\boldsymbol{x}) = \mathbb{E}[\boldsymbol{X}_1 | \boldsymbol{X}_t = \boldsymbol{x}]$; and $a_t, b_t$ are useful shorthands to be used later.

**Remark B.1.** The probability flow ODE formulation of *diffusion models* (Y. Song, Sohl-Dickstein, et al. 2021) is subsumed by flow models, and represents a model with an affine Gaussian probability paths (AGGP), *i.e.*, $(\boldsymbol{X}_0, \boldsymbol{X}_1) \sim \pi_{0,1}(\boldsymbol{x}_0, \boldsymbol{x}_1) = p(\boldsymbol{x}_0)q(\boldsymbol{x}_1)$ with $p(\boldsymbol{x}) = \mathcal{N}(\boldsymbol{x}|\boldsymbol{0}, \sigma^2 \boldsymbol{I})$ (Lipman, Havasi, et al. 2024). Thus without loss of generality we consider flow models of affine probability paths.[6]

## B.2 Assumptions

Throughout the norm $\|\cdot\|$ corresponds to the Euclidean norm $\|\cdot\|_2$. Additionally, we make the following (mild) regularity assumptions:

**Assumption B.1.** *The function $a_t := \frac{\dot{\sigma}_t}{\sigma_t}$ is integrable in $[0,1]$.*

**Assumption B.2.** *The total derivatives $\frac{\mathrm{d}^n}{\mathrm{d}\gamma^n}\left[\boldsymbol{x}_{1|\gamma}^\theta(\boldsymbol{x})\right]$ exist and are continuous for $0 \le n \le k-1$.*

Assumption B.1 is necessary for the simplification that we perform with exponential integrators and Ben-Hamu et al. (2024) make the same assumption in their analysis of the continuous adjoint equations for affine probability paths. Assumption B.2 is to ensure that we can take a Taylor expansion of $\boldsymbol{x}_{1|\gamma}^\theta(\boldsymbol{x})$.

## B.3 Proof of Proposition 4.1

We restate Proposition 4.1 below.

---

[6]Clearly, diffusion models which solve the reverse-time SDE are different and require a separate analysis.

**Proposition 4.1** (Exact solution of affine probability paths). *Given an initial value of $\boldsymbol{x}_s$ at time $s \in [0,1]$ the solution $\boldsymbol{x}_t$ at time $t \in [0,1]$ of an ODE governed by the vector field in Equation (18) is:*

$$\boldsymbol{x}_t = \frac{\sigma_t}{\sigma_s}\boldsymbol{x}_s + \sigma_t \int_{\gamma_s}^{\gamma_t} \boldsymbol{x}_{1|\gamma}^{\theta}(\boldsymbol{x}_\gamma)\, \mathrm{d}\gamma. \tag{8}$$

*Proof.* Recall that we uniquely define a flow model through the vector field $\boldsymbol{u} \in \mathcal{C}^{1,1}([0,1] \times \mathbb{R}^d; \mathbb{R}^d)$. The vector field which models the affine conditional flow with schedule $(\alpha_t, \sigma_t)$, is defined as

$$\boldsymbol{u}_t^{\theta}(\boldsymbol{x}) = \mathbb{E}[\dot{\alpha}_t \boldsymbol{X}_1 + \dot{\sigma}_t \boldsymbol{X}_0 | \boldsymbol{X}_t = \boldsymbol{x}]. \tag{20}$$

With some simple algebra, we can rewrite the vector field in terms of $\hat{\boldsymbol{x}}_{1|t}$,

$$\boldsymbol{u}_t^{\theta}(\boldsymbol{x}) = a_t \boldsymbol{x} + b_t \boldsymbol{x}_{1|t}^{\theta}(\boldsymbol{x}),$$
$$a_t = \frac{\dot{\sigma}_t}{\sigma_t} \qquad b_t = \dot{\alpha}_t - \alpha_t \frac{\dot{\sigma}_t}{\sigma_t}. \tag{21}$$

Now using this definition we can rewrite the solution for $\boldsymbol{x}_t$ from $\boldsymbol{x}_s$ in terms of $\hat{\boldsymbol{x}}_{1|t}$,

$$\boldsymbol{x}_t = \boldsymbol{x}_s + \int_s^t \boldsymbol{u}_\tau^{\theta}(\boldsymbol{x}_\tau)\, \mathrm{d}\tau, \tag{22}$$

$$\boldsymbol{x}_t = \boldsymbol{x}_s + \int_s^t a_\tau \boldsymbol{x}_\tau + b_\tau \boldsymbol{x}_{1|\tau}^{\theta}(\boldsymbol{x}_\tau)\, \mathrm{d}\tau. \tag{23}$$

Note the semi-linear form of the integral equation. We can exploit this structure using the technique of *exponential integrators*, (see Gonzalez et al. 2024; Lu et al. 2022a; Q. Zhang and Y. Chen 2023), to simplify Equation (23), under Assumption B.1, to

$$\boldsymbol{x}_t = e^{\int_s^t a_u\, \mathrm{d}u}\boldsymbol{x}_s + \int_s^t e^{\int_\tau^t a_u\, \mathrm{d}u} b_\tau \boldsymbol{x}_{1|\tau}^{\theta}(\boldsymbol{x}_\tau)\, \mathrm{d}\tau. \tag{24}$$

Now, the integrating factor simplifies quite nicely to

$$e^{\int_s^t a_u\, \mathrm{d}u} = e^{\int_s^t \frac{\dot{\sigma}_u}{\sigma_u}\, \mathrm{d}u} = e^{\int_{\sigma_s}^{\sigma_t} \frac{1}{\sigma}\, \mathrm{d}\sigma} = \frac{\sigma_t}{\sigma_s}, \tag{25}$$

such that Equation (24) becomes

$$\boldsymbol{x}_t = \frac{\sigma_t}{\sigma_s}\boldsymbol{x}_s + \sigma_t \int_s^t \frac{b_\tau}{\sigma_\tau} \boldsymbol{x}_{1|\tau}^{\theta}(\boldsymbol{x}_\tau)\, \mathrm{d}\tau. \tag{26}$$

We can simplify $b_t/\sigma_t$ to find:

$$\frac{b_t}{\sigma_t} = \frac{\dot{\alpha}_t \sigma_t - \alpha_t \dot{\sigma}_t}{\sigma_t^2} = \frac{\mathrm{d}}{\mathrm{d}t}\left(\frac{\alpha_t}{\sigma_t}\right) = \frac{\mathrm{d}}{\mathrm{d}t}\gamma_t, \tag{27}$$

where $\gamma_t := \alpha_t/\sigma_t$, *i.e.*, the signal-to-noise ratio. As such, we can rewrite Equation (26) with a change of variables $\boldsymbol{x}_\gamma = \boldsymbol{x}_{\gamma_t^{-1}(\gamma)} = \boldsymbol{x}_t$,

$$\boldsymbol{x}_t = \frac{\sigma_t}{\sigma_s}\boldsymbol{x}_s + \sigma_t \int_{\gamma_s}^{\gamma_t} \boldsymbol{x}_{1|\gamma}^{\theta}(\boldsymbol{x}_\gamma)\, \mathrm{d}\gamma, \tag{28}$$

concluding the proof. $\square$

## B.4 Proof of Proposition 4.2

**Proposition 4.2** (Greedy as an explicit Euler scheme within DTO). *For some trajectory state $\boldsymbol{x}_t$ at time $t$, the greedy gradient given by $\nabla_{\boldsymbol{x}}\mathcal{L}(\boldsymbol{x}_{1|t}^{\theta}(\boldsymbol{x}))$ is the DTO scheme with an explicit Euler discretization with step size $h = \gamma_1 - \gamma_t$.*

*Proof.* From Proposition 4.1 we see that using the target prediction model to estimate $\boldsymbol{x}_1$ is akin to taking a first-order approximation of the flow. More specifically, under Assumption B.2 we can construct a $(k-1)$-th Taylor expansion of Equation (8) with:

$$\boldsymbol{x}_t = \frac{\sigma_t}{\sigma_s}\boldsymbol{x}_s + \sigma_t \sum_{n=0}^{k-1} \frac{\mathrm{d}^n}{\mathrm{d}\gamma^n}\left[\boldsymbol{x}_{1|\gamma}^\theta(\boldsymbol{x}_\gamma)\right]_{\gamma=\gamma_s} \int_{\gamma_s}^{\gamma_t} \frac{(\gamma-\gamma_s)^n}{n!}\,\mathrm{d}\gamma + \mathcal{O}(h^{k+1}), \tag{29}$$

$$= \frac{\sigma_t}{\sigma_s}\boldsymbol{x}_s + \sigma_t \sum_{n=0}^{k-1} \frac{\mathrm{d}^n}{\mathrm{d}\gamma^n}\left[\boldsymbol{x}_{1|\gamma}^\theta(\boldsymbol{x}_\gamma)\right]_{\gamma=\gamma_s} \frac{h^{n+1}}{(n+1)!} + \mathcal{O}(h^{k+1}), \tag{30}$$

where $h := \gamma_t - \gamma_s$ is the step size. Then it follows that for $k = 1$ the first-order discretization of the flow, omitting high-order error terms becomes,

$$\boldsymbol{x}_t \approx \tilde{\boldsymbol{x}}_t = \frac{\sigma_t}{\sigma_s}\boldsymbol{x}_s + (\alpha_t + \frac{\sigma_t\alpha_s}{\sigma_s})\boldsymbol{x}_{1|s}^\theta(\boldsymbol{x}_s). \tag{31}$$

In the limit as $t \to 1$ we have $\tilde{\boldsymbol{x}}_t = \boldsymbol{x}_{1|s}^\theta(\boldsymbol{x}_s)$.[7] Thus, the greedy gradient is a DTO scheme with an explicit Euler discretization with step size $h = \gamma_1 - \gamma_t$. $\qquad\square$

## B.5 Proof of Proposition 4.3

We restate Proposition 4.3 below.

> **Proposition 4.3** (Greedy as an implicit Euler scheme within OTD). *For some trajectory state $\boldsymbol{x}_t$ at time $t$, the greedy gradient given by $\nabla_{\boldsymbol{x}_t}\mathcal{L}(\boldsymbol{x}_{1|t}^\theta(\boldsymbol{x}_t))$ is an implicit Euler discretization of the continuous adjoint equations for the true gradients with step size $h = \gamma_1 - \gamma_t$.*

For clarity we restate the definition of the continuous adjoint equations. Let $\boldsymbol{u}_\theta \in \mathcal{C}^{1,1}([0,1]\times\mathbb{R}^d;\mathbb{R}^d)$ be a model that models the vector field of some ODE and be Lipschitz continuous in its second argument. Let $\boldsymbol{x}: [0,1] \to \mathbb{R}^d$ be the solution to the ODE with the initial condition $\boldsymbol{x}_0 \in \mathbb{R}^d$, $\dot{\boldsymbol{x}}_t = \boldsymbol{u}_\theta(t, \boldsymbol{x}_t)$. For some scalar-valued loss function $\mathcal{L} \in \mathcal{C}^2(\mathbb{R}^d)$ in $\boldsymbol{x}_1$, let $\boldsymbol{a}_{\boldsymbol{x}} := \partial\mathcal{L}/\partial\boldsymbol{x}_t$ denote the gradient. Then $\boldsymbol{a}_{\boldsymbol{x}}$ and related quantity $\boldsymbol{a}_\theta := \partial\mathcal{L}/\partial\theta$ can be found by solving an augmented ODE of the form,

$$\begin{aligned}
\boldsymbol{a}_{\boldsymbol{x}}(1) &= \frac{\partial\mathcal{L}}{\partial\boldsymbol{x}_1}, & \frac{\mathrm{d}\boldsymbol{a}_{\boldsymbol{x}}}{\mathrm{d}t}(t) &= -\boldsymbol{a}_{\boldsymbol{x}}(t)^\top\frac{\partial\boldsymbol{u}_\theta}{\partial\boldsymbol{x}}(t, \boldsymbol{x}_t), \\
\boldsymbol{a}_\theta(1) &= \boldsymbol{0}, & \frac{\mathrm{d}\boldsymbol{a}_\theta}{\mathrm{d}t}(t) &= -\boldsymbol{a}_{\boldsymbol{x}}(t)^\top\frac{\partial\boldsymbol{u}_\theta}{\partial\theta}(t, \boldsymbol{x}_t).
\end{aligned} \tag{32}$$

Now we present the proof.

*Proof.* The adjoint state can be simplified by rewriting the vector field in terms of the target prediction model to find

$$\frac{\mathrm{d}\boldsymbol{a}_{\boldsymbol{x}}}{\mathrm{d}t}(t) = -a_t\boldsymbol{a}_{\boldsymbol{x}}(t) - b_t\boldsymbol{a}_{\boldsymbol{x}}(t)^\top\frac{\partial\boldsymbol{x}_{1|t}^\theta(\boldsymbol{x}_t)}{\partial\boldsymbol{x}_t}. \tag{33}$$

We can express this *backwards-in-time* ODE as an integral equation in the form of

$$\begin{aligned}
\boldsymbol{a}_{\boldsymbol{x}}(s) &= \boldsymbol{a}_{\boldsymbol{x}}(t) - \int_t^s a_\tau\boldsymbol{a}_{\boldsymbol{x}}(t) + b_\tau\boldsymbol{a}_{\boldsymbol{x}}(\tau)^\top\frac{\partial\boldsymbol{x}_{1|\tau}^\theta(\boldsymbol{x}_\tau)}{\boldsymbol{x}_\tau}\,\mathrm{d}\tau, \\
&= \boldsymbol{a}_{\boldsymbol{x}}(t) + \int_s^t a_\tau\boldsymbol{a}_{\boldsymbol{x}}(t) + b_\tau\boldsymbol{a}_{\boldsymbol{x}}(\tau)^\top\frac{\partial\boldsymbol{x}_{1|\tau}^\theta(\boldsymbol{x}_\tau)}{\partial\boldsymbol{x}_\tau}\,\mathrm{d}\tau. \quad\text{(time-reversal)}
\end{aligned} \tag{34}$$

---

[7]Note that despite $\sigma_t \to 0$ the asymptotic behavior is well-defined (see Ben-Hamu et al. 2024).

Using the technique of exponential integrators we rewrite the integral as

$$\boldsymbol{a_x}(s) = e^{\int_s^t a_u \, \mathrm{d}u} \boldsymbol{a_x}(t) + \int_s^t e^{\int_\tau^t a_u \, \mathrm{d}u} b_\tau \boldsymbol{a_x}(\tau)^\top \frac{\partial \boldsymbol{x}_{1|\tau}^\theta(\boldsymbol{x}_\tau)}{\partial \boldsymbol{x}_\tau} \, \mathrm{d}\tau,$$

$$= \frac{\sigma_t}{\sigma_s} \boldsymbol{a_x}(t) + \sigma_t \int_s^t \frac{b_\tau}{\sigma_\tau} \boldsymbol{a_x}(\tau)^\top \frac{\partial \boldsymbol{x}_{1|\tau}^\theta(\boldsymbol{x}_\tau)}{\partial \boldsymbol{x}_\tau} \, \mathrm{d}\tau,$$

$$= \frac{\sigma_t}{\sigma_s} \boldsymbol{a_x}(t) + \sigma_t \int_{\gamma_s}^{\gamma_t} \boldsymbol{a_x}(\gamma)^\top \frac{\partial \boldsymbol{x}_{1|\gamma}^\theta(\boldsymbol{x}_\gamma)}{\partial \boldsymbol{x}_\gamma} \, \mathrm{d}\gamma. \tag{35}$$

By Assumption B.2 it follows that the vector-Jacobian product has $(k-1)$-th total derivatives, allowing us to define a first-order Taylor expansion around $\gamma_s$:

$$\boldsymbol{a_x}(s) = \frac{\sigma_t}{\sigma_s} \boldsymbol{a_x}(t) + (\alpha_t - \frac{\sigma_t}{\sigma_s}\alpha_s) \boldsymbol{a_x}(s)^\top \frac{\partial \hat{\boldsymbol{x}}_{1|s}(\boldsymbol{x}_s)}{\partial \boldsymbol{x}_s} + \mathcal{O}(h^2). \tag{36}$$

Thus, the first-order approximation of the adjoint state at time $t$ with a step size of $h = \gamma_1 - \gamma_t$ is the implicit equation

$$\boldsymbol{a_x}(t) = \boldsymbol{a_x}(t)^\top \frac{\partial \hat{\boldsymbol{x}}_{1|t}(\boldsymbol{x}_t)}{\partial \boldsymbol{x}_t}. \tag{37}$$

Now to solve the implicit equation we can use the fixed-point iteration method. Let $\boldsymbol{a_x}(t)^{(0)} = \boldsymbol{a_x}(1)$, then the first iteration has

$$\boldsymbol{a_x}(t)^{(1)} = \boldsymbol{a_x}(1)^\top \frac{\partial \hat{\boldsymbol{x}}_{1|t}(\boldsymbol{x}_t)}{\partial \boldsymbol{x}_t} = \nabla_{\boldsymbol{x}_t} \mathcal{L}(\hat{\boldsymbol{x}}_{1|t}(\boldsymbol{x}_t)). \tag{38}$$

Thus, we have shown that the greedy gradients are equivalent to the first iteration of an implicit Euler discretization of the continuous adjoint equations.

$\square$

# C Dynamics of guidance

In this section we detail some of the formalisms omitted in the main paper concerning the dynamics of the gradient flow and greedy gradients.

We begin by re-establishing some useful prior results. Ben-Hamu et al. (2024, Proposition 4.1) showed that the gradient of the target prediction model is proportional to the variance of the random variable defined by $p_{1|t}(\boldsymbol{x}_1|\boldsymbol{x})$, we restate their result below.

**Lemma C.1** (Gradient of target prediction model). *For affine Gaussian probability paths, the gradient of the target prediction model $\boldsymbol{x}_{1|t}^\theta(\boldsymbol{x})$ w.r.t. $\boldsymbol{x}$ is proportional to the variance of $p_{1|t}(\boldsymbol{x}_1|\boldsymbol{x})$, i.e.,*

$$\nabla_{\boldsymbol{x}} \boldsymbol{x}_{1|t}^\theta(\boldsymbol{x}) = \frac{\alpha_t}{\sigma_t^2} \mathrm{Var}_{1|t}(\boldsymbol{x}), \tag{39}$$

*where*

$$\mathrm{Var}_{1|t}(\boldsymbol{x}) = \mathbb{E}_{p_{1|t}(\boldsymbol{x}_1|\boldsymbol{x})} \left[ (\boldsymbol{x}_1 - \boldsymbol{x}_{1|t}^\theta(\boldsymbol{x}))(\boldsymbol{x}_1 - \boldsymbol{x}_{1|t}^\theta(\boldsymbol{x}))^\top \right]. \tag{40}$$

**Remark C.1.** This can be written more generally in terms of the (pushforward) differential $D_x \boldsymbol{x}_{1|t}^\theta(\boldsymbol{x})$ where the underlying spaces are smooth manifolds and $\boldsymbol{x}_{1|t}^\theta$ is a smooth map between them (Ben-Hamu et al. 2024). In this section, we only consider flow models defined in Euclidean spaces, and so we opt not to elaborate on this generalization.

We restate a well-known result below in Lemma C.2 regarding the continuous-time analogue to forward-mode autodifferentiation, or in other words, forward sensitivity.

**Lemma C.2** (Dynamics of Jacobian matrices for flows). *Let $\boldsymbol{x}_0 \in \mathbb{R}^d$ and let $\boldsymbol{f} \in \mathcal{C}^{1,1}([0,T] \times \mathbb{R}^d; \mathbb{R}^d)$ be uniformly Lipschitz in $\boldsymbol{x}$. Let $\boldsymbol{x} : [0,T] \to \mathbb{R}^d$ be the unique solution to*

$$\boldsymbol{x}(0) = \boldsymbol{x}_0, \qquad \frac{\mathrm{d}\boldsymbol{x}}{\mathrm{d}t}(t) = \boldsymbol{f}(t, \boldsymbol{x}(t)). \tag{41}$$

Let $\Phi_{s,t}(\boldsymbol{x})$, $s,t \in [0,T]$ denote the flow associated with Equation (41). Then let $\boldsymbol{J}_s(t) := \nabla_{\boldsymbol{x}}\Phi_{s,t}(\boldsymbol{x})$ denote the Jacobian matrices, where $\boldsymbol{J}_s : [s,T] \to \mathbb{R}^{d \times d}$ solve the differential equation

$$\boldsymbol{J}_s(s) = \boldsymbol{I}, \qquad \frac{\mathrm{d}\boldsymbol{J}_s}{\mathrm{d}t}(t) = \nabla_{\boldsymbol{x}}\boldsymbol{f}(t, \Phi_{s,t}(\boldsymbol{x}(s)))\boldsymbol{J}_s(t), \qquad (42)$$

where $\nabla_{\boldsymbol{x}}\boldsymbol{f}(t, \cdot)$ refers to the gradient w.r.t. the second argument.

**Remark C.2.** This result is well known and has been extended to *controlled differential equations* (Friz and Victoir 2010, Theorem 4.4) and *rough differential equations* (Friz and Victoir 2010, Theorem 11.3). Kidger (2022, Theorem 5.8) discusses this result for neural ODEs.

## C.1 Proof of Theorem 5.1

We restate Theorem 5.1 below.

---

**Theorem 5.1** (Jacobian matrices of affine Gaussian probability paths). *For the standard affine Gaussian probability path with flow model $\Phi_{s,t}^{\theta}(\boldsymbol{x})$, the Jacobian matrix $\nabla_{\boldsymbol{x}}\Phi_{s,t}(\boldsymbol{x})$ as function of $\boldsymbol{x}$ is given as the solution to*

$$\nabla_{\boldsymbol{x}}\Phi_{s,t}^{\theta}(\boldsymbol{x}) = \frac{\sigma_t}{\sigma_s}\boldsymbol{I} + \sigma_t \int_s^t \dot{\gamma}_u \frac{\gamma_u}{\sigma_u}\mathrm{Var}_{1|u}(\Phi_{s,u}^{\theta}(\boldsymbol{x}))\nabla_{\boldsymbol{x}}\Phi_{s,u}^{\theta}(\boldsymbol{x}) \, \mathrm{d}u, \qquad (10)$$

*where*

$$\mathrm{Var}_{1|t}(\boldsymbol{x}) = \mathbb{E}_{p_{1|t}(\boldsymbol{x}_1|\boldsymbol{x})}\left[(\boldsymbol{x}_1 - \boldsymbol{x}_{1|t}^{\theta}(\boldsymbol{x}))(\boldsymbol{x}_1 - \boldsymbol{x}_{1|t}^{\theta}(\boldsymbol{x}))^{\top}\right]. \qquad (11)$$

---

This proof follows a similar technique to that used by Blasingame and C. Liu (2024a) to simplify adjoint equations for diffusion models using exponential integrators.

*Proof.* Now recall Lemma C.2 which discusses the dynamics of Jacobian matrices for flows, rewriting this as an integral equation yields:

$$\nabla_{\boldsymbol{x}}\Phi_{s,t}^{\theta}(\boldsymbol{x}) = \boldsymbol{I} + \int_s^t \nabla_{\boldsymbol{x}_u}\boldsymbol{u}_u^{\theta}(\Phi_{s,u}^{\theta}(\boldsymbol{x}))\nabla_{\boldsymbol{x}}\Phi_{s,u}^{\theta}(\boldsymbol{x}) \, \mathrm{d}u. \qquad (43)$$

Now recall the definition of the marginal vector field in terms of the target prediction model (*cf.* Equation (19)) which we use to rewrite Equation (43) as

$$\nabla_{\boldsymbol{x}}\Phi_{s,t}^{\theta}(\boldsymbol{x}) = \boldsymbol{I} + \int_s^t \nabla_{\boldsymbol{x}_u}a_u\Phi_{s,u}^{\theta}(\boldsymbol{x})\nabla_{\boldsymbol{x}}\Phi_{s,u}^{\theta}(\boldsymbol{x}) + \nabla_{\boldsymbol{x}_u}b_u\boldsymbol{x}_{1|u}^{\theta}(\Phi_{s,u}^{\theta}(\boldsymbol{x}))\nabla_{\boldsymbol{x}}\Phi_{s,u}^{\theta}(\boldsymbol{x}) \, \mathrm{d}u,$$

$$\overset{(i)}{=} \boldsymbol{I} + \int_s^t a_u\nabla_{\boldsymbol{x}}\Phi_{s,u}^{\theta}(\boldsymbol{x}) + b_u\nabla_{\boldsymbol{x}_u}\boldsymbol{x}_{1|u}^{\theta}(\Phi_{s,u}^{\theta}(\boldsymbol{x}))\nabla_{\boldsymbol{x}}\Phi_{s,u}^{\theta}(\boldsymbol{x}) \, \mathrm{d}u, \qquad (44)$$

where (i) holds by $\nabla_{\boldsymbol{x}_u}\Phi_{s,u}^{\theta}(\boldsymbol{x}) = \boldsymbol{I}$. Next we can make use of the popular technique of *exponential integrators* to simplify Equation (43) in combination with Equation (19). Thus, the integral equation in Equation (44) becomes

$$\nabla_{\boldsymbol{x}}\Phi_{s,t}^{\theta}(\boldsymbol{x}) = \Lambda_a(s,t)\boldsymbol{I} + \int_s^t \Lambda_a(u,t)b_u\nabla_{\boldsymbol{x}_u}\boldsymbol{x}_{1|u}^{\theta}(\Phi_{s,u}^{\theta}(\boldsymbol{x}))\nabla_{\boldsymbol{x}}\Phi_{s,u}^{\theta}(\boldsymbol{x}) \, \mathrm{d}u, \qquad (45)$$

where $\Lambda_a(s,t) := \exp \int_s^t a_u \, \mathrm{d}u$ is the integrating factor. This simplifies to $\Lambda_a(s,t) = \sigma_t/\sigma_s$. Using this, Equation (45) can be simplified to

$$\nabla_{\boldsymbol{x}}\Phi_{s,t}^{\theta}(\boldsymbol{x}) = \frac{\sigma_t}{\sigma_s}\boldsymbol{I} + \sigma_t \int_s^t \frac{b_u}{\sigma_u}\nabla_{\boldsymbol{x}_u}\boldsymbol{x}_{1|u}^{\theta}(\Phi_{s,u}^{\theta}(\boldsymbol{x}))\nabla_{\boldsymbol{x}}\Phi_{s,u}^{\theta}(\boldsymbol{x}) \, \mathrm{d}u. \qquad (46)$$

Now we can apply Lemma C.1 to further simplify Equation (46) to find

$$\nabla_{\boldsymbol{x}}\Phi_{s,t}^{\theta}(\boldsymbol{x}) = \frac{\sigma_t}{\sigma_s}\boldsymbol{I} + \sigma_t \int_s^t \frac{\alpha_u}{\sigma_u^3}b_u\mathrm{Var}_{1|u}(\Phi_{s,u}^{\theta}(\boldsymbol{x}))\nabla_{\boldsymbol{x}}\Phi_{s,u}^{\theta}(\boldsymbol{x}) \, \mathrm{d}u. \qquad (47)$$

Next we simplify the coefficient $\alpha_u b_u / \sigma_u^3$ in the integral term. Let $\gamma_t := \alpha_t / \sigma_t$ equal the signal-to-noise-ratio. Then we observe

$$
\begin{aligned}
b_t \frac{\alpha_t}{\sigma_t^3} &= \left( \dot{\alpha}_t - \alpha_t \frac{\dot{\sigma}_t}{\sigma_t} \right) \frac{\alpha_t}{\sigma_t^3}, \\
&= \frac{\dot{\alpha}_t \sigma_t - \dot{\sigma}_t \alpha_t}{\sigma_t^3} \frac{\alpha_t}{\sigma_t^2}, \\
&\overset{(i)}{=} \frac{\mathrm{d}}{\mathrm{d}t} \left[ \frac{\alpha_t}{\sigma_t} \right] \frac{\alpha_t}{\sigma_t} \frac{1}{\sigma_t}, \\
&\overset{(ii)}{=} \dot{\gamma}_t \frac{\gamma_t}{\sigma_t},
\end{aligned}
\tag{48}
$$

where (i) holds by the quotient rule and (ii) holds by definition of $\gamma_t$. Using this simplification we can perform a change-of-variables to simplify the gradient resulting in

$$
\nabla_{\boldsymbol{x}} \Phi_{s,t}^\theta(\boldsymbol{x}) = \frac{\sigma_t}{\sigma_s} \boldsymbol{I} + \sigma_t \int_s^t \dot{\gamma}_u \frac{\gamma_u}{\sigma_u} \mathrm{Var}_{1|u}(\Phi_{s,u}^\theta(\boldsymbol{x})) \nabla_{\boldsymbol{x}} \Phi_{s,u}^\theta(\boldsymbol{x}) \, \mathrm{d}u.
\tag{49}
$$

$\square$

**Remark C.3.** Readers familiar with the work of Ben-Hamu et al. (2024) may notice some similarities between our result Theorem 5.1 and Ben-Hamu et al. (2024, Theorem 4.2). The difference between the two is that the former is a simplified integral equation; whereas, the latter is the exact solution and no longer requires solving an ODE. However, this later solution does require solving a time-ordered exponential which requires a formal truncated series expansion, *e.g.*, Magnus expansion.

Theorem 5.1 is closely related to Ben-Hamu et al. (2024, Theorem 4.2) which we restate below within the context of our notational conventions.[8]

**Theorem C.3.** *For the standard affine Gaussian probability path, the differential of $\Phi_{0,1}^\theta(\boldsymbol{x})$ as of function of $\boldsymbol{x}$ is*

$$
\nabla_{\boldsymbol{x}} \Phi_{0,1}^\theta(\boldsymbol{x}) = \sigma_1 \mathcal{T} \exp \left[ \int_0^1 \frac{1}{2} \dot{\gamma}_t^2 \mathrm{Var}_{1|t}(\boldsymbol{x}) \, \mathrm{d}t \right],
\tag{50}
$$

*where $\mathcal{T} \exp$ denotes the time-ordered exponential.*

The time-ordered exponential[9] (Grossman and Katz 1972) is defined as

$$
\begin{aligned}
\mathcal{T} \exp \left[ \int_t^1 \boldsymbol{A}(s) \, \mathrm{d}s \right] &= \sum_{n=0}^\infty \frac{1}{n!} \int_t^1 \mathrm{d}s_1 \cdots \int_t^1 \mathrm{d}s_n \ \mathcal{T}\{\boldsymbol{A}(s_1) \ldots \boldsymbol{A}(s_n)\}, \\
&= \sum_{n=0}^\infty \int_t^1 \mathrm{d}s_1 \int_t^{s_1} \mathrm{d}s_2 \cdots \int_t^{s_{n-1}} \mathrm{d}s_n \ \boldsymbol{A}(s_1) \boldsymbol{A}(s_2) \ldots \boldsymbol{A}(s_n),
\end{aligned}
\tag{51}
$$

and the solution can be found the Dyson series (Sakurai and Napolitano 2020) or Magnus expansion (Magnus 1954), which are truncated in practice. The meta-operator $\mathcal{T}$ denotes the time-ordering (Dyson 1949), *e.g.*, consider the time-ordering of two operators $\boldsymbol{A}, \boldsymbol{B}$:

$$
\mathcal{T}\{\boldsymbol{A}(s_1)\boldsymbol{B}(s_2)\} := \begin{cases} \boldsymbol{A}(s_1)\boldsymbol{B}(s_2) & \text{if } s_1 > s_2, \\ \pm \boldsymbol{B}(s_2)\boldsymbol{A}(s_1) & \text{otherwise.} \end{cases}
\tag{52}
$$

For more details we refer the reader to Weinberg (1995).

## C.2 Dynamics of gradient guidance

We state this more formally below in Proposition C.4.

---

[8]With abuse of notation let $\dot{\gamma}_t^2$ denote the time derivative of $\gamma_t^2$.

[9]This is closely related to the *Peano-Baker series* (see Frazer, Duncan, and Collar 1938, Section 7.5).

**Proposition C.4** (Dynamics of gradient guidance). *Consider the standard affine Gaussian probability paths model trained to zero loss. The Gateaux differential of $\boldsymbol{x}$ at some time $t \in [0, 1]$ in the direction of the gradient $\nabla_{\boldsymbol{x}} \mathcal{L} \left( \Phi_{t,1}^{\theta}(\boldsymbol{x}) \right)$ is given by*

$$\delta_{\boldsymbol{x}} \Phi_{t,1}^{\theta}(\boldsymbol{x}) = -\nabla_{\boldsymbol{x}} \Phi_{t,1}^{\theta}(\boldsymbol{x}) \nabla_{\boldsymbol{x}} \Phi_{t,1}^{\theta}(\boldsymbol{x})^{\top} \nabla_{\boldsymbol{x}_1} \mathcal{L}(\boldsymbol{x}_1). \tag{53}$$

Thus the behavior of $\boldsymbol{x}_1$ when guided by $\mathcal{L}$ is determined by the operator $\nabla_{\boldsymbol{x}} \Phi_{t,1}^{\theta}(\boldsymbol{x})$ which iteratively projects the gradient of the loss function by the covariance matrix $\mathrm{Var}_{1|t}(\boldsymbol{x})$. Put another way:

Performing gradient guidance with $\mathcal{L}$ at time $t < 1$ amounts to guidance which follows the target distribution $p(\boldsymbol{X}_1)$ by projecting $\nabla_{\boldsymbol{x}_1} \mathcal{L}(\boldsymbol{x}_1)$ onto to the target distribution via the local covariance matrix.

It is for this reason that it is undesirable to simply perform guidance in the data space as we are likely to deviate from this target distribution. From Equation (53) we know that applying the gradient at earlier timesteps causes the initial gradient $\nabla_{\boldsymbol{x}_1} \mathcal{L}(\boldsymbol{x}_1)$ to be projected into high-variance directions of the target distribution causing the guided sample to stay closer to the true target distribution.

The next question is: how does $\boldsymbol{x}_1$ change when $\boldsymbol{x}$ is updated with our greedy guidance strategy?

### C.3 Proof of Proposition C.4

We restate Proposition C.4 below.

**Proposition C.4** (Dynamics of gradient guidance). *Consider the standard affine Gaussian probability paths model trained to zero loss. The Gateaux differential of $\boldsymbol{x}$ at some time $t \in [0, 1]$ in the direction of the gradient $\nabla_{\boldsymbol{x}} \mathcal{L} \left( \Phi_{t,1}^{\theta}(\boldsymbol{x}) \right)$ is given by*

$$\delta_{\boldsymbol{x}} \Phi_{t,1}^{\theta}(\boldsymbol{x}) = -\nabla_{\boldsymbol{x}} \Phi_{t,1}^{\theta}(\boldsymbol{x}) \nabla_{\boldsymbol{x}} \Phi_{t,1}^{\theta}(\boldsymbol{x})^{\top} \nabla_{\boldsymbol{x}_1} \mathcal{L}(\boldsymbol{x}_1). \tag{53}$$

*Proof.* This can be shown from a straightforward derivation:

$$
\begin{aligned}
\delta_{\boldsymbol{x}} \Phi_{t,1}^{\theta}(\boldsymbol{x}) &\stackrel{(i)}{=} \left. \frac{\mathrm{d}}{\mathrm{d}\eta} \right|_{\eta=0} \Phi_{t,1}^{\theta} \left( \boldsymbol{x} - \eta \nabla_{\boldsymbol{x}} \mathcal{L} \left( \Phi_{t,1}^{\theta}(\boldsymbol{x}) \right) \right), \\
&\stackrel{(ii)}{=} -\nabla_{\boldsymbol{x}} \Phi_{t,1}^{\theta} \left( \boldsymbol{x} - \eta \nabla_{\boldsymbol{x}} \mathcal{L} \left( \Phi_{t,1}^{\theta}(\boldsymbol{x}) \right) \right) \nabla_{\boldsymbol{x}} \mathcal{L} \left( \Phi_{t,1}^{\theta}(\boldsymbol{x}) \right) \Big|_{\eta=0}, \\
&= -\nabla_{\boldsymbol{x}} \Phi_{t,1}^{\theta}(\boldsymbol{x}) \nabla_{\boldsymbol{x}} \mathcal{L} \left( \Phi_{t,1}^{\theta}(\boldsymbol{x}) \right), \\
&\stackrel{(iii)}{=} -\nabla_{\boldsymbol{x}} \Phi_{t,1}^{\theta}(\boldsymbol{x}) \nabla_{\boldsymbol{x}} \Phi_{t,1}^{\theta}(\boldsymbol{x})^{\top} \nabla_{\boldsymbol{x}_1} \mathcal{L}(\boldsymbol{x}_1), 
\end{aligned}
\tag{54}
$$

where (i) holds by the definition of the Gateaux differential, (ii) holds by the chain rule, and (iii) holds by a substitution of Equation (9) with the simplification of $\boldsymbol{x}_1 = \Phi_{t,1}^{\theta}(\boldsymbol{x})$. $\square$

### C.4 Proof of Proposition 5.2

We restate Proposition 5.2 below.

**Proposition 5.2** (Dynamics of greedy gradient guidance). *Consider the standard affine Gaussian probability paths model trained to zero loss. The Gateaux differential of $\boldsymbol{x}$ at some time $t \in [0, 1]$ in the direction of the gradient $\nabla_{\boldsymbol{x}} \mathcal{L} \left( \boldsymbol{x}_{1|t}^{\theta}(\boldsymbol{x}) \right)$ is given by*

$$\delta_{\boldsymbol{x}}^{\mathcal{G}} \Phi_{t,1}^{\theta}(\boldsymbol{x}) = -\nabla_{\boldsymbol{x}} \Phi_{t,1}^{\theta}(\boldsymbol{x}) \nabla_{\boldsymbol{x}} \boldsymbol{x}_{1|t}^{\theta}(\boldsymbol{x})^{\top} \nabla_{\boldsymbol{x}_1} \mathcal{L}(\boldsymbol{x}_1). \tag{13}$$

*Proof.* This can be shown from a straightforward derivation:

$$\delta_{\boldsymbol{x}}^{\mathcal{G}}\Phi_{t,1}^{\theta}(\boldsymbol{x}) \stackrel{(i)}{=} \frac{\mathrm{d}}{\mathrm{d}\eta}\bigg|_{\eta=0} \Phi_{t,1}^{\theta}\left(\boldsymbol{x} - \eta\nabla_{\boldsymbol{x}}\mathcal{L}\left(\boldsymbol{x}_{1|t}^{\theta}(\boldsymbol{x})\right)\right),$$

$$\stackrel{(ii)}{=} -\nabla_{\boldsymbol{x}}\Phi_{t,1}^{\theta}\left(\boldsymbol{x} - \eta\nabla_{\boldsymbol{x}}\mathcal{L}\left(\Phi_{t,1}^{\theta}(\boldsymbol{x})\right)\right)\nabla_{\boldsymbol{x}}\mathcal{L}\left(\boldsymbol{x}_{1|t}^{\theta}(\boldsymbol{x})\right)\bigg|_{\eta=0},$$

$$= -\nabla_{\boldsymbol{x}}\Phi_{t,1}^{\theta}(\boldsymbol{x})\nabla_{\boldsymbol{x}}\mathcal{L}\left(\boldsymbol{x}_{1|t}^{\theta}(\boldsymbol{x})\right),$$

$$\stackrel{(iii)}{=} -\nabla_{\boldsymbol{x}}\Phi_{t,1}^{\theta}(\boldsymbol{x})\nabla_{\boldsymbol{x}}\boldsymbol{x}_{1|t}^{\theta}(\boldsymbol{x})^{\top}\nabla_{\boldsymbol{x}_1}\mathcal{L}(\boldsymbol{x}_1), \tag{55}$$

where (i) holds by the definition of the Gateaux differential, (ii) holds by the chain rule, and (iii) holds by the chain rule. $\qquad\square$

We note an interesting corollary below.

**Corollary C.4.1** (Dynamics of gradient vs greedy guidance). *The difference between the dynamics of gradient guidance in Proposition C.4 and greedy gradient guidance in Proposition 5.2 for a point* $\boldsymbol{x}$ *at time* $t$ *with guidance function* $\mathcal{L} \in \mathcal{C}^1(\mathbb{R}^d)$ *is*

$$\left\|\delta_{\boldsymbol{x}}\Phi_{t,1}^{\theta}(\boldsymbol{x}) - \delta_{\boldsymbol{x}}^{\mathcal{G}}\Phi_{t,1}^{\theta}(\boldsymbol{x})\right\| = \left\|\nabla_{\boldsymbol{x}}\Phi_{t,1}^{\theta}(\boldsymbol{x})\left(\nabla_{\boldsymbol{x}}\Phi_{t,1}^{\theta}(\boldsymbol{x}) - \nabla_{\boldsymbol{x}}\boldsymbol{x}_{1|t}^{\theta}(\boldsymbol{x})\right)^{\top}\nabla_{\boldsymbol{x}_1}\mathcal{L}(\boldsymbol{x}_1)\right\|. \tag{56}$$

## C.5 Proof of Theorem 5.3

We restate Theorem 5.3 below.

> **Theorem 5.3** (Dynamics of gradient vs greedy guidance). *The difference between the dynamics of gradient guidance in Proposition C.4 and greedy gradient guidance in Proposition 5.2 for a point* $\boldsymbol{x}$ *at time* $t$ *with guidance function* $\mathcal{L} \in \mathcal{C}^1(\mathbb{R}^d)$ *is bounded by* $\mathcal{O}(h^2)$ *where* $h := \gamma_1 - \gamma_t$, *i.e.,*
>
> $$\left\|\nabla_{\boldsymbol{x}}\Phi_{t,1}^{\theta}(\boldsymbol{x}) - \nabla_{\boldsymbol{x}}\boldsymbol{x}_{1|t}^{\theta}(\boldsymbol{x})\right\| = \mathcal{O}(h^2). \tag{14}$$

*Proof.* From Corollary C.4.1 it is clear that the difference between $\delta_{\boldsymbol{x}}\Phi_{t,1}^{\theta}(\boldsymbol{x})$ and $\delta_{\boldsymbol{x}}^{\mathcal{G}}\Phi_{t,1}^{\theta}(\boldsymbol{x})$ amounts to the difference between the true gradient and gradient of the target prediction model. Recall Theorem 5.1 which enables to write the gradient as the solution to an integral equation:

$$\nabla_{\boldsymbol{x}}\Phi_{t,1}^{\theta}(\boldsymbol{x}) = \frac{\sigma_1}{\sigma_t}\boldsymbol{I} + \sigma_1\int_t^1 \dot{\gamma}_u\frac{\gamma_u}{\sigma_u}\mathrm{Var}_{1|u}(\Phi_{s,u}^{\theta}(\boldsymbol{x}))\nabla_{\boldsymbol{x}}\Phi_{t,u}^{\theta}(\boldsymbol{x})\,\mathrm{d}u. \tag{57}$$

Now as $\sigma_t \to 0$ as $t \to 1$, we can simplify the integral equation

$$\nabla_{\boldsymbol{x}}\Phi_{t,1}^{\theta}(\boldsymbol{x}) = \sigma_1\int_t^1 \dot{\gamma}_u\frac{\gamma_u}{\sigma_u}\mathrm{Var}_{1|u}(\Phi_{t,u}^{\theta}(\boldsymbol{x}))\nabla_{\boldsymbol{x}}\Phi_{t,u}^{\theta}(\boldsymbol{x})\,\mathrm{d}u, \tag{58}$$

and then by rewriting the integral in terms of $\mathrm{d}\gamma = \dot{\gamma}_u\mathrm{d}u$ we find

$$\nabla_{\boldsymbol{x}}\Phi_{t,1}^{\theta}(\boldsymbol{x}) = \sigma_1\int_{\gamma_t}^{\gamma_1} \frac{\gamma}{\sigma_\gamma}\mathrm{Var}_{1|\gamma}(\Phi_{\gamma_t,\gamma}^{\theta}(\boldsymbol{x}))\nabla_{\boldsymbol{x}}\Phi_{\gamma_t,\gamma}^{\theta}(\boldsymbol{x})\,\mathrm{d}\gamma. \tag{59}$$

Next we take a first-order Taylor expansion of $\frac{1}{\sigma_\gamma}\mathrm{Var}_{1|\gamma}(\Phi_{\gamma_t,\gamma}^{\theta}(\boldsymbol{x}))\nabla_{\boldsymbol{x}}\Phi_{\gamma_t,\gamma}^{\theta}(\boldsymbol{x})$ centered at $\gamma_t$ which yields:

$$\frac{\gamma}{\sigma_\gamma}\mathrm{Var}_{1|\gamma}(\Phi_{\gamma_t,\gamma}^{\theta}(\boldsymbol{x}))\nabla_{\boldsymbol{x}}\Phi_{\gamma_t,\gamma}^{\theta}(\boldsymbol{x}) = \frac{\gamma_t}{\sigma_t}\mathrm{Var}_{1|t}(\boldsymbol{x}) + \mathcal{O}(\gamma - \gamma_t). \tag{60}$$

For this analysis, it is actually more convenient to include the $\gamma$ term as part of the Taylor expansion rather than computing it in closed form in the integral. Now plugging Equation (60) into Equation (59)

yields

$$\nabla_{\boldsymbol{x}}\Phi_{t,1}^{\theta}(\boldsymbol{x}) = \sigma_1 \int_{\gamma_t}^{\gamma_1} \gamma \frac{1}{\sigma_t} \mathrm{Var}_{1|t}(\boldsymbol{x}) + \mathcal{O}(\gamma - \gamma_t) \, \mathrm{d}\gamma,$$

$$\overset{(i)}{=} \sigma_1 \frac{\gamma_t}{\sigma_t} \mathrm{Var}_{1|t}(\boldsymbol{x}) \int_{\gamma_t}^{\gamma_1} \mathrm{d}\gamma + \mathcal{O}(h^2),$$

$$= \sigma_1 \frac{\gamma_t}{\sigma_t} \mathrm{Var}_{1|t}(\boldsymbol{x}) (\gamma_1 - \gamma_t) + \mathcal{O}(h^2), \tag{61}$$

where (i) holds with $h := \gamma_1 - \gamma_t$. Then, with a little algebra we have

$$\nabla_{\boldsymbol{x}}\Phi_{t,1}^{\theta}(\boldsymbol{x}) = \sigma_1 \frac{\alpha_t}{\sigma_t^2} (\gamma_1 - \gamma_t) \mathrm{Var}_{1|t}(\boldsymbol{x}) + \mathcal{O}(h^2),$$

$$= \sigma_1 \frac{\alpha_t}{\sigma_t^2} \left( \frac{\alpha_1}{\sigma_1} - \frac{\alpha_t}{\sigma_t} \right) \mathrm{Var}_{1|t}(\boldsymbol{x}) + \mathcal{O}(h^2),$$

$$= \frac{\alpha_t}{\sigma_t^2} \left( \alpha_1 - \sigma_1 \frac{\alpha_t}{\sigma_t} \right) \mathrm{Var}_{1|t}(\boldsymbol{x}) + \mathcal{O}(h^2),$$

$$\overset{(i)}{=} \frac{\alpha_t}{\sigma_t^2} \mathrm{Var}_{1|t}(\boldsymbol{x}) + \mathcal{O}(h^2), \tag{62}$$

where (i) holds by the boundary conditions of the schedule (*cf*. Equation (17)). Now recall Lemma C.1 which states:

$$\nabla_{\boldsymbol{x}}\boldsymbol{x}_{1|t}^{\theta}(\boldsymbol{x}) = \frac{\alpha_t}{\sigma_t^2} \mathrm{Var}_{1|t}(\boldsymbol{x}). \tag{63}$$

Thus from Equation (62) and Equation (63) it is easy to see that

$$\left\| \nabla_{\boldsymbol{x}}\Phi_{t,1}^{\theta}(\boldsymbol{x}) - \nabla_{\boldsymbol{x}}\boldsymbol{x}_{1|t}^{\theta}(\boldsymbol{x}) \right\| = \mathcal{O}(h^2), \tag{64}$$

holds and thus

$$\left\| \delta_{\boldsymbol{x}}\Phi_{t,1}^{\theta}(\boldsymbol{x}) - \delta_{\boldsymbol{x}}^{\mathcal{G}}\Phi_{t,1}^{\theta}(\boldsymbol{x}) \right\| = \mathcal{O}(h^2). \tag{65}$$

$\square$

## C.6  Proof of Theorem 5.4

We restate Theorem 5.4 below.

> **Theorem 5.4** (Greedy convergence). *For affine probability paths, if there exists a sequence of states $\boldsymbol{x}_t^{(n)}$ at time $t$ such that it converges to the locally optimal solution $\boldsymbol{x}_{1|t}^{\theta}(\boldsymbol{x}_t^{(n)}) \to \boldsymbol{x}_1^*$.*
> *Then the solution, $\Phi_{1|t}^{\theta}(\boldsymbol{x}_t^{(n)})$, converges to a neighborhood of size $\mathcal{O}(h^2)$ centered at $\boldsymbol{x}_1^*$.*

*Proof.* By Assumption B.2, we can take a $(k-1)$-th order Taylor expansion around $\gamma_t$ of the flow in Equation (8) to obtain

$$\Phi_{1|t}^{\theta}(\boldsymbol{x}_t) = \frac{\sigma_1}{\sigma_t}\boldsymbol{x}_t + \sigma_1 \int_{\gamma_t}^{\gamma_1} \sum_{n=0}^{k-1} \frac{\mathrm{d}^n}{\mathrm{d}\gamma^n} \left[ \boldsymbol{x}_{1|\gamma}^{\theta}(\boldsymbol{x}_\gamma) \right]_{\gamma=\gamma_t} \frac{(\gamma - \gamma_t)^n}{n!} \, \mathrm{d}\gamma + \mathcal{O}(h^{k+1}),$$

$$= \frac{\sigma_1}{\sigma_t}\boldsymbol{x}_t + \sigma_1 \sum_{n=0}^{k-1} \frac{\mathrm{d}^n}{\mathrm{d}\gamma^n} \left[ \boldsymbol{x}_{1|\gamma}^{\theta}(\boldsymbol{x}_\gamma) \right]_{\gamma=\gamma_t} \int_{\gamma_t}^{\gamma_1} \frac{(\gamma - \gamma_t)^n}{n!} \, \mathrm{d}\gamma + \mathcal{O}(h^{k+1}),$$

$$= \frac{\sigma_1}{\sigma_t}\boldsymbol{x}_t + \sigma_1 \sum_{n=0}^{k-1} \frac{\mathrm{d}^n}{\mathrm{d}\gamma^n} \left[ \boldsymbol{x}_{1|\gamma}^{\theta}(\boldsymbol{x}_\gamma) \right]_{\gamma=\gamma_t} \frac{h^{n+1}}{(n+1)!} + \mathcal{O}(h^{k+1}), \tag{66}$$

where $h := \gamma_1 - \gamma_t$ is the stepsize. Let $k = 1$, then we have:

$$\Phi_{1|t}^{\theta}(\boldsymbol{x}_t) = \frac{\sigma_1}{\sigma_t}\boldsymbol{x}_n + \sigma_1 \hat{\boldsymbol{x}}_{1|t}(\boldsymbol{x}_t)h + \mathcal{O}(h^2), \tag{67}$$

$$= \frac{\sigma_1}{\sigma_t}\boldsymbol{x}_n + (\alpha_1 - \frac{\sigma_1\alpha_t}{\sigma_t})\hat{\boldsymbol{x}}_{1|t}(\boldsymbol{x}_t) + \mathcal{O}(h^2). \tag{68}$$

By definition $\sigma_1 = 0$ and $\alpha_1 = 1$, then

$$\Phi_{1|t}^\theta(\boldsymbol{x}_t) = \hat{\boldsymbol{x}}_{1|t}(\boldsymbol{x}_t) + \mathcal{O}(h^2), \tag{69}$$

which is equivalent to

$$\left\| \Phi_{1|t}^\theta(\boldsymbol{x}_t) - \hat{\boldsymbol{x}}_{1|t}(\boldsymbol{x}_t) \right\| \le C_1 h^2, \tag{70}$$

for some constant $C_1 > 0$. Since $\boldsymbol{x}_{1|t}^\theta(\boldsymbol{x}_t^{(n)}) \to \boldsymbol{x}_1^*$ we know that for any $\epsilon > 0$ there exists some $n \ge N$ such that $\|\boldsymbol{x}_1^* - \boldsymbol{x}_{1|t}^\theta(\boldsymbol{x}_t^{(n)})\| < \epsilon$. Thus,

$$\left\| \Phi_{1|t}^\theta(\boldsymbol{x}_t^{(n)}) - \boldsymbol{x}_1^* \right\| \le \left\| \Phi_{1|t}^\theta(\boldsymbol{x}_t^{(n)}) - \boldsymbol{x}_{1|t}^\theta(\boldsymbol{x}_t^{(n)}) \right\| + \left\| \boldsymbol{x}_1^* - \boldsymbol{x}_{1|t}^\theta(\boldsymbol{x}_t^{(n)}) \right\| < \underbrace{\epsilon + C_1 h^2}_{:=C_2}. \tag{71}$$

Therefore, $\Phi_{1|t}(\boldsymbol{x}_t^{(n)})$ converges to a point inside a neighborhood centered at $\boldsymbol{x}_1^*$ with radius $\mathcal{O}(h^2)$. $\square$

# D  Beyond Euler

In this section we provide the full proofs and derivations for Section 6 in the main paper.

## D.1  Proof of Theorem 6.1

Before showing Theorem 6.1 we show a more general version below.

> **Theorem D.1** (Local truncation error of discretize-then-optimize gradients). *Let $\boldsymbol{\Phi}$ be an explicit Runge-Kutta solver of order $\alpha > 0$ to the ODE*
>
> $$\boldsymbol{x}(0) = \boldsymbol{x}_0, \qquad \frac{\mathrm{d}\boldsymbol{x}}{\mathrm{d}t}(t) = \boldsymbol{u}_\theta(t, \boldsymbol{x}(t)), \tag{72}$$
>
> *on $[0, T]$ which satisfies the regularity conditions for the Picard-Lindelöf theorem. Let $\Phi_{s,t}^\theta(\boldsymbol{x})$ denote the flow from $s$ to $t$, for any $s, t \in [0, T]$ admitted by the ODE. Then,*
>
> $$\left\| \nabla_{\boldsymbol{x}} \Phi_{s,t}^\theta(\boldsymbol{x}) - \nabla_{\boldsymbol{x}} \boldsymbol{\Phi}_{s,t}(\boldsymbol{x}) \right\| = \mathcal{O}(h^{\alpha+1}). \tag{73}$$

*Proof.* Consider an explicit $k$-stage Runge-Kutta method given by

$$\boldsymbol{u}_{n,j} = \boldsymbol{u}_\theta \left( t_n + c_j h, \boldsymbol{x}_n + h \sum_{i=1}^{j} a_{j,i} \boldsymbol{u}_{n,i} \right), \qquad j = 1, 2, \ldots, k \tag{74}$$

$$\boldsymbol{x}_{n+1} = \boldsymbol{x}_n + h \sum_{j=1}^{k} b_j \boldsymbol{u}_{n,j}, \tag{75}$$

where $a_{j,i}, b_j, c_j$ are all given via the *Butcher Tableau* (Stewart 2022, Section 6.1.4). Now, we consider a single step from time $s$ to time $t$ with initial value $\boldsymbol{x}$ and step size $h := t - s$. Then, the gradient is

$$\begin{aligned}
\nabla_{\boldsymbol{x}} \boldsymbol{\Phi}_{s,t}(\boldsymbol{x}) &= \nabla_{\boldsymbol{x}} \boldsymbol{x} + h \sum_{j=1}^{k} b_j \nabla_{\boldsymbol{x}} \boldsymbol{u}_\theta \left( s + c_j h, \boldsymbol{x} + h \sum_{i=1}^{j} a_{j,i} \boldsymbol{u}_i \right), \\
&= \boldsymbol{I} + h \sum_{j=1}^{k} b_j \left[ \nabla_{\hat{\boldsymbol{x}}_j} \boldsymbol{u}_\theta(s + c_j h, \hat{\boldsymbol{x}}_j) \left( \boldsymbol{I} + h \sum_{i=1}^{j} a_{j,i} \nabla_{\boldsymbol{x}} \boldsymbol{u}_i \right) \right],
\end{aligned} \tag{76}$$

where we let

$$\hat{\boldsymbol{x}}_j = \boldsymbol{x} + h \sum_{i=1}^{j} a_{j,i} \boldsymbol{u}_i. \tag{77}$$

Next, recall Lemma C.2 which gives the following ODE

$$\boldsymbol{J}_s(s) = \boldsymbol{I}, \qquad \frac{\mathrm{d}\boldsymbol{J}_s}{\mathrm{d}t}(t) = \nabla_{\boldsymbol{x}}\boldsymbol{u}_\theta(t, \Phi_{s,t}(\boldsymbol{x}))\boldsymbol{J}_s(t). \tag{78}$$

Next, we augmented the ODE above with the underyling ODE for the solution state, $\dot{\boldsymbol{x}}(t) = \boldsymbol{u}_\theta(t, \boldsymbol{x}(t))$. We now apply the same Runge-Kutta solver to this augmented ODE for the Jacobian matrices which yields

$$\boldsymbol{U}_j = \boldsymbol{I} + h\sum_{j=1}^{k} b_j \left[ \nabla_{\hat{\boldsymbol{x}}_j}\boldsymbol{u}_\theta\left(s + c_j h, \hat{\boldsymbol{x}}_j\right) \left(\boldsymbol{I} + h\sum_{i=1}^{j} a_{j,i}\nabla_{\boldsymbol{x}}\boldsymbol{u}_i\right) \right]. \tag{79}$$

Clearly, Equation (79) and Equation (105) are equivalent. Now as the underlying numerical solver has local truncation error $\mathcal{O}(h^{\alpha+1})$ we find that

$$\left\|\nabla_{\boldsymbol{x}}\Phi_{s,t}^\theta(\boldsymbol{x}) - \nabla_{\boldsymbol{x}}\boldsymbol{\Phi}_{s,t}(\boldsymbol{x})\right\| = \mathcal{O}(h^{\alpha+1}). \tag{80}$$

$\square$

**Remark D.1.** This result is intuitive as differentiation is a linear operator. However simple, we believe the insight is useful on the discussion of using DTO/OTD/posterior methods for guidance and thus include it here.

**Remark D.2.** Theorem D.1 shows that DTO and OTD are really just two sides of the same coin and that one of the main differences is the choice of end points when discretizing.

**Remark D.3.** Onken and Ruthotto (2020, Appendix A) made similar observations; however, it is for only of the case of Euler.

---

**Theorem 6.1** (Truncation error of single-step gradients). *Let $\boldsymbol{\Phi}$ be an explict Runge-Kutta solver of order $\alpha > 0$ of a flow model with flow $\Phi_{s,t}^\theta(\boldsymbol{x})$. Then for any $t \in [0, 1]$,*

$$\left\|\nabla_{\boldsymbol{x}}\Phi_{t,1}^\theta(\boldsymbol{x}) - \nabla_{\boldsymbol{x}}\boldsymbol{\Phi}_{t,1}(\boldsymbol{x})\right\| = \mathcal{O}(h^{\alpha+1}), \tag{15}$$

*where $h = 1 - t$.*

---

*Proof.* This follows as a corollary of Theorem D.1. $\square$

**Corollary D.1.1** (Convergence of a $\alpha$-th order posterior gradient). *For affine probability paths, if there exists a sequence of states $\boldsymbol{x}_t^{(n)}$ at time $t$ such that it converges to the locally optimal solution $\Phi_{t,1}^\theta(\boldsymbol{x}_t^{(n)}) \to \boldsymbol{x}_1^*$. Then solution, $\Phi_{1|t}^\theta(\boldsymbol{x}_t^{(n)})$, converges to a neighborhood of size $\mathcal{O}(h^{\alpha+1})$ centered at $\boldsymbol{x}_1^*$.*

*Proof.* This follows as a straightforward derivation from Theorem D.1. $\square$

**Corollary D.1.2** (Dynamics of $\alpha$-th order posterior gradient). *Consider the standard affine Gaussian probability paths model trained to zero loss. Let $\boldsymbol{\Phi}$ be an explicit Runga-Kutta solver of order $\alpha > 0$ of a flow model with flow $\Phi_{s,t}^\theta(\boldsymbol{x})$. The Gateaux differential of $\boldsymbol{x}$ at some time $t \in [0, 1]$ in the direction of the gradient $\nabla_{\boldsymbol{x}}\mathcal{L}(\boldsymbol{\Phi}_{t,1}(\boldsymbol{x}))$ is given by*

$$\delta_{\boldsymbol{x}}^{\boldsymbol{\Phi}}(\boldsymbol{x}) = -\nabla_{\boldsymbol{x}}\Phi_{t,1}^\theta(\boldsymbol{x})\nabla_{\boldsymbol{x}}\boldsymbol{\Phi}_{t,1}(\boldsymbol{x})^\top\nabla_{\boldsymbol{x}_1}\mathcal{L}(\boldsymbol{x}_1). \tag{81}$$

*Proof.* This follows straightforwardly from Proposition 5.2 and Theorem 6.1. $\square$

## D.2 A useful reparameterization of the flow model

We present a useful reparameterization of the flow model, which is a parallel result to Proposition 4.1.

**Proposition D.2** (Reparameterized for the target prediction model of affine probability paths). *The ODE governed by the vector field in Equation* (18) *can be reparameterized as*

$$\frac{\mathrm{d}\boldsymbol{y}_\gamma}{\mathrm{d}\gamma} = \sigma_0 \boldsymbol{x}_{1|\zeta}^\theta \left( \frac{\sigma_\gamma}{\sigma_0} \boldsymbol{y}_\zeta \right), \tag{82}$$

*where* $\boldsymbol{y}_t = \frac{\sigma_0}{\sigma_t} \boldsymbol{x}_t$.

*Proof.* The ODE governed by the vector field in Equation (18) can be written as

$$\frac{\mathrm{d}\boldsymbol{x}_t}{\mathrm{d}t} = a_t \boldsymbol{x}_t + b_t \boldsymbol{x}_{1|t}^\theta(\boldsymbol{x}_t). \tag{83}$$

Now we can use the technique of exponential integrators to rewrite the ODE as

$$\frac{\mathrm{d}}{\mathrm{d}t} \left[ e^{\int_0^t -a_u \, \mathrm{d}u} \boldsymbol{x}_t \right] = e^{\int_0^t -a_u \, \mathrm{d}u} b_t \boldsymbol{x}_{1|t}^\theta(\boldsymbol{x}_t). \tag{84}$$

The exponential term can be simplified to

$$e^{\int_0^t -a_u \, \mathrm{d}u} = \frac{\sigma_0}{\sigma_t}. \tag{85}$$

We introduce a *change-of-variables*, $\boldsymbol{y}_t = \frac{\sigma_0}{\sigma_t} \boldsymbol{x}_t$. Thus, the ODE becomes

$$\frac{\mathrm{d}\boldsymbol{y}_t}{\mathrm{d}t} = \frac{\sigma_0}{\sigma_t} b_t \boldsymbol{x}_{1|t}^\theta \left( \frac{\sigma_t}{\sigma_0} \boldsymbol{y}_t \right). \tag{86}$$

Next, recall that $b_t/\sigma_t = \dot{\gamma}_t$ (*cf.* Equation (27)) which enables a change of integration variable:

$$\frac{\mathrm{d}\boldsymbol{y}_\gamma}{\mathrm{d}\gamma} = \sigma_0 \boldsymbol{x}_{1|\gamma}^\theta \left( \frac{\sigma_\gamma}{\sigma_0} \boldsymbol{y}_\gamma \right). \tag{87}$$

$\square$

**Remark D.4.** Recall that, often, for affine probability paths we let $\sigma_0 = 1$, further simplifying Proposition D.2 to

$$\frac{\mathrm{d}\boldsymbol{y}_\gamma}{\mathrm{d}\gamma} = \boldsymbol{x}_{1|\gamma}^\theta \left( \sigma_\gamma \boldsymbol{y}_\gamma \right). \tag{88}$$

**Remark D.5.** Proposition D.2 is a tangential result to the prior result of Pan, Liew, et al. (2024, Equation (11)) which was for diffusion models and was developed w.r.t. the source prediction model rather than the target prediction model and was solved in reverse-time.[10]

This parameterization in Proposition D.2 can be combined with Theorem D.1 to construct a DTO approximation of the gradient with truncation error $(\gamma_t - \gamma_s)^{\alpha+1}$.

# E   Notes on using OTD in practice

While the OTD approach has become quite popular after the work of R. T. Chen et al. (2018), several later works have noticed several key issues that we wish to note for ML practitioners.

Recall our prototypical neural ODE (or flow model) of the form

$$\frac{\mathrm{d}\boldsymbol{x}}{\mathrm{d}t}(t) = \boldsymbol{u}_\theta(t, \boldsymbol{x}(t)), \tag{89}$$

and assume it is defined on the interval $[0, T]$ and the flow model statifies the usual regularity conditions. Then, the continuous adjoint equations (Kidger 2022, Theorem 5.2) are:

$$\begin{aligned} \boldsymbol{a}_{\boldsymbol{x}}(T) &= \frac{\partial \mathcal{L}}{\partial \boldsymbol{x}_T}, & \frac{\mathrm{d}\boldsymbol{a}_{\boldsymbol{x}}}{\mathrm{d}t}(t) &= -\boldsymbol{a}_{\boldsymbol{x}}(t)^\top \frac{\partial \boldsymbol{u}_\theta}{\partial \boldsymbol{x}}(t, \boldsymbol{x}(t)), \\ \boldsymbol{a}_\theta(T) &= \boldsymbol{0}, & \frac{\mathrm{d}\boldsymbol{a}_\theta}{\mathrm{d}t}(t) &= -\boldsymbol{a}_{\boldsymbol{x}}(t)^\top \frac{\partial \boldsymbol{u}_\theta}{\partial \theta}(t, \boldsymbol{x}(t)), \end{aligned} \tag{90}$$

where $\boldsymbol{a}_{\boldsymbol{x}}(t) := \partial \mathcal{L}/\partial \boldsymbol{x}(t)$ and $\boldsymbol{a}_\theta(0) := \partial \mathcal{L}/\partial \theta$.

---

[10]Technically forward-time due to the conventions of diffusion models.

Table 4: Comparison of different strategies for performing backpropagation through flow models. For the complexity analysis $n$ denotes the number of discretization steps and $d$ the dimensionality of the state. Note, for accuracy we mean there are no truncation errors. Note that whilst in general the stability of reversible solvers is quite poor, there are *some* solvers which have a non-trival region of stability.

| Method | Time | Memory | Accurate gradients | Stability |
|---|---|---|---|---|
| DTO | $\mathcal{O}(n)$ | $\mathcal{O}(nd^2)$ | ✓ | - |
| DTO + recursive checkpointing | $\mathcal{O}(n \log n)$ | $\mathcal{O}(d^2 \log n)$ | ✓ | - |
| OTD + stored trajectory | $\mathcal{O}(n)$ | $\mathcal{O}(nd + d^2)$ | ✓ | - |
| OTD + reversible solver | $\mathcal{O}(n)$ | $\mathcal{O}(d^2)$ | ✓ | ? |
| OTD | $\mathcal{O}(n)$ | $\mathcal{O}(d^2)$ | ✗ | ✗ |

**Truncation errors.** One area of concern is the potential mismatch between the forward trajectory $\{\boldsymbol{x}_{t_i}\}_{i=1}^N$ and the backward trajectory $\{\tilde{\boldsymbol{x}}_{t_i}\}_{i=1}^N$ when performing the backwards solve. *E.g.*, consider an explicit Euler scheme

$$\boldsymbol{x}_{t_{i+1}} = \boldsymbol{x}_{t_i} + (t_{i+1} - t_i)\boldsymbol{u}_\theta(t_i, \boldsymbol{x}_{t_i}). \tag{91}$$

The same scheme when applied to solving the backward trajectory would yield,

$$\tilde{\boldsymbol{x}}_{t_i} = \tilde{\boldsymbol{x}}_{t_{i+1}} + (t_i - t_{i+1})\boldsymbol{u}_\theta(t_{i+1}, \tilde{\boldsymbol{x}}_{t_{i+1}}). \tag{92}$$

Clearly, there is no guarantee that these two trajectories match during the forward and backward solve introducing a source of error. One potential solution is to use an *algebraically reversible solver* (see Blasingame and C. Liu 2025; Kidger et al. 2021; McCallum and Foster 2024) which guarantees that the forward and backward trajectory match *perfectly*. Another option is to store the forward trajectory $\{\boldsymbol{x}_{t_i}\}_{i=1}^N$ in memory and use *interpolated adjoints* if the backward timesteps do not perfectly align with the forward timesteps (see S. Kim et al. 2021).

**Stability concerns.** Consider the simple ODE, $\dot{y}(t) = \lambda y(t)$ defined on $t \in [0, T]$ with $y(0) = y_0$ and $\lambda < 0$. Clearly, most ODE solvers with a non-trivial region of stability (see Harier and Wanner 2002, Definition 2.1) will solve this ODE without an issue, as the errors will decrease exponentially with $\lambda < 0$. However, in the backwards in time solve from $y(T)$ the errors will *grow exponentially*. It can be shown that the adjoint state suffers from similar stability issues. The local behavior of a differential equation is described through the eigenvalues of the Jacobian of the vector field (see Butcher 2016). For $\boldsymbol{x}_t$ this is given by $\frac{\partial \boldsymbol{u}_\theta}{\partial \boldsymbol{x}}$ and for $\boldsymbol{a}_{\boldsymbol{x}}$ this is given by

$$\frac{\partial}{\partial \boldsymbol{a}_{\boldsymbol{x}}}\left(-\boldsymbol{a}_{\boldsymbol{x}}(t)^\top \frac{\partial \boldsymbol{u}_\theta}{\partial \boldsymbol{x}}(t, \boldsymbol{x}(t))\right) = -\frac{\partial \boldsymbol{u}_\theta}{\partial \boldsymbol{x}}(t, \boldsymbol{x}(t)). \tag{93}$$

Clearly, the Jacobians for $\boldsymbol{a}_{\boldsymbol{x}}$ and $\boldsymbol{x}_t$ solved in reverse-time are identical, meaning the stability of the backward solve is pushed onto the solve for the adjoint state (see Kidger 2022, Section 5.1.2.4) for more details. Reversible solvers eliminate truncation errors, but tend to suffer from poor stability, *e.g.*, the region of stability for reversible Heun applied to neural ODEs is the complex interval $[-i, i]$ (Kidger et al. 2021). Recent work by McCallum and Foster (2024), however, has shown a strategy for constructing reversible solvers with a non-trivial region of stability.

**Recommendations.** In light of these concerns we propose we consider to be best practices for deciding what scheme to use.

Generally, the best choice is DTO when memory allows as it is the most *accurate* in terms of the forward discretization. If memory is an issue then using a clever checkpointing scheme (Griewank 1992; Griewank and Walther 2000; Stumm and Walther 2010) can help alleviate such issues in exchange for additional compute time. The recursive checkpointing strategy in combination with DTO is actually the default (and recommended) implementation in the `Diffrax` library. Alternatively, one could store the forward trajectory in memory and then apply the OTD scheme on these stored states (not activations). This strategy of caching the forward trajectory is quite popular and was used by Blasingame and C. Liu (2024a) and Domingo-Enrich et al. (2025) in practice when solving the continuous adjoint equations. Another option is to use an algebraically reversible solver in

conjunction with OTD. Lastly, one could use vanilla OTD, which we should mention can actually work reasonably well depending on the application despite the concerns listed above.

In Table 4 we summarize the discussion of this section and hope it is helpful to the reader.

# F  On control signal optimization

Rather than optimizing the trajectory of the solution or the initial condition, several works (X. Liu et al. 2023; L. Wang et al. 2025) have explored the guidance from the perspective of optimal control (Kirk 2004). In essence this technique first injects an additional control signal, $z \in \mathcal{C}^1(\mathbb{R}; \mathbb{R}^d)$, to the vector field, $u_t^\theta$, such that

$$\frac{\mathrm{d}x_t}{\mathrm{d}t} = u_t^\theta(x_t) + z(t). \tag{94}$$

Thus, instead of optimizing $\{x_t\}_{t \in [0,t]}$ directly, this control signal can instead be optimized, serving as one of the key insights in (X. Liu et al. 2023; L. Wang et al. 2025). *I.e.*, suppose we have a neural ODE with vector field $u_t^\theta(x)$, then we can write the optimization problem as

$$\min_z \quad \mathcal{L}(x_T) + \lambda \int_0^T \|z(t)\| \, \mathrm{d}t, \tag{95}$$
$$\text{s.t.} \quad x_T = x_0 + \int_0^T u_t^\theta(x_t) + z(t) \, \mathrm{d}t.$$

The next natural question then is to ask about the behavior of a greedy strategy applied to $z(t)$. To simplify the analysis, we now consider a control signal applied to the posterior model $x_{1|t}^\theta$ such that it is replaced by $x_{1|t}^\theta(x_t) + z(t)$ which amounts to simply rescaling $z(t)$ from Equation (94) with $b_t$. From this construction, it should be clear that the greedy gradient for the control signal is merely $\nabla_{\tilde{x}_1} \mathcal{L}(\tilde{x}_1)$. If using the original formulation where the control signal is applied to the vector field, rather than the denoiser, the gradient is simply scaled by a weighting function dependent on time. Note that this approach is similar to the greedy approach taken by Blasingame and C. Liu (2024b); however, they inject the control signal to the source prediction model rather than the target prediction model.

## F.1  Continuous adjoint equations for control signals

We can model the gradient for this signal by augmenting the continuous adjoint equations with the adjoint state $a_z(t) := \partial \mathcal{L} / \partial z(t)$. In Theorem F.1 we show that this gradient is simply an integral of the adjoint state $a_x(t)$.

> **Theorem F.1** (Continuous adjoint equations for the control term). *Let $u_t^\theta \in \mathcal{C}^{1,1}([0,T] \times \mathbb{R}^{d_x}; \mathbb{R}^{d_x})$ be a parameterization of some time-dependent vector field of a neural ODE that is Lipschitz continuous in its second argument, and let $z \in \mathcal{C}^1([0,1]; \mathbb{R}^d)$ be an additional control signal such that the new dynamics are given by Equation (94). Let $a_z(t) := \partial \mathcal{L} / \partial z(t)$, then*
>
> $$a_z(t) = -\int_T^t a_x(s) \, \mathrm{d}s. \tag{96}$$

Our proof follows the structure of the modern proof of Pontryagin's original result (Pontryagin et al. 1963) presented by (R. T. Chen et al. 2018); and is similar to the form used by Blasingame and C. Liu (2024a, Theorem 2.2).

*Proof.* For notational clarity, we use the notation $x(t) = x_t$. We define the augmented state on $[0, T]$ as

$$\frac{\mathrm{d}}{\mathrm{d}t} \begin{bmatrix} x \\ z \end{bmatrix}(t) = f_{\mathrm{aug}} = \begin{bmatrix} u_\theta(t, x(t)) + z(t) \\ \frac{\mathrm{d}z}{\mathrm{d}t}(t) \end{bmatrix}, \tag{97}$$

and the augmented adjoint state as

$$a_{\mathrm{aug}}(t) := \begin{bmatrix} a_x \\ a_z \end{bmatrix}(t). \tag{98}$$

The Jacobian of $\boldsymbol{f}_{\mathrm{aug}}$ has form

$$\frac{\partial \boldsymbol{f}_{\mathrm{aug}}}{\partial [\boldsymbol{x}, \boldsymbol{z}]} = \begin{bmatrix} \frac{\partial \boldsymbol{u}_\theta(t, \boldsymbol{x}(t))}{\partial \boldsymbol{x}} & \mathbf{1} \\ \mathbf{0} & \mathbf{0} \end{bmatrix}. \tag{99}$$

The evolution of the adjoint state is given by

$$\frac{\mathrm{d}\boldsymbol{a}_{\mathrm{aug}}}{\mathrm{d}t}(t) = - \begin{bmatrix} \boldsymbol{a}_{\boldsymbol{x}} & \boldsymbol{a}_{\boldsymbol{z}} \end{bmatrix}(t) \frac{\partial \boldsymbol{f}_{\mathrm{aug}}}{\partial [\boldsymbol{x}, \boldsymbol{z}]}(t). \tag{100}$$

Therefore, $\boldsymbol{a}_{\boldsymbol{u}}(t)$ evolves with

$$\boldsymbol{a}_{\boldsymbol{u}}(T) = \mathbf{0}, \qquad \frac{\mathrm{d}\boldsymbol{a}_{\boldsymbol{u}}}{\mathrm{d}t}(t) = -\boldsymbol{a}_{\boldsymbol{x}}(t), \tag{101}$$

thereby finishing the proof. $\qquad\square$

## G   Implementation details

We discuss how to implement the greedy strategy.

### G.1   The construction of the greedy guidance schemes

Recall that the general Butcher tableau for a $k$-stage explicit RK scheme (Stewart 2022, Section 6.1.4) is written as

$$
\begin{array}{c|ccccc}
c_1 & & & & & \\
c_2 & a_{2,1} & & & & \\
c_3 & a_{3,1} & a_{3,2} & & & \\
\vdots & \vdots & \vdots & \ddots & & \\
c_k & a_{k,1} & a_{k,2} & \cdots & a_{(k-1),k} & \\
\hline
& b_1 & b_2 & \cdots & b_{k-1} & b_k
\end{array}
\quad = \quad
\begin{array}{c|c}
c & a \\
\hline
& b
\end{array}. \tag{102}
$$

Thus a single-step is given by

$$\boldsymbol{u}_{n,j} = \boldsymbol{u}_\theta \left( t_n + c_j h, \boldsymbol{x}_n + h \sum_{i=1}^{j} a_{j,i} \boldsymbol{u}_{n,i} \right), \qquad j = 1, 2, \ldots, k \tag{103}$$

$$\boldsymbol{x}_{n+1} = \boldsymbol{x}_n + h \sum_{j=1}^{k} b_j \boldsymbol{u}_{n,j}, \tag{104}$$

where $a_{j,i}, b_j, c_j$ are all given via the *Butcher Tableau* (Stewart 2022, Section 6.1.4). Now, we consider a single step from time $s$ to time $t$ with initial value $\boldsymbol{x}$ and step size $h := t - s$. Then, the gradient is

$$\nabla_{\boldsymbol{x}} \boldsymbol{\Phi}_{s,t}(\boldsymbol{x}) = \nabla_{\boldsymbol{x}} \boldsymbol{x} + h \sum_{j=1}^{k} b_j \nabla_{\boldsymbol{x}} \boldsymbol{u}_\theta \left( s + c_j h, \boldsymbol{x} + h \sum_{i=1}^{j} a_{j,i} \boldsymbol{u}_i \right),$$

$$= \boldsymbol{I} + h \sum_{j=1}^{k} b_j \left[ \nabla_{\hat{\boldsymbol{x}}_j} \boldsymbol{u}_\theta(s + c_j h, \hat{\boldsymbol{x}}_j) \left( \boldsymbol{I} + h \sum_{i=1}^{j} a_{j,i} \nabla_{\boldsymbol{x}} \boldsymbol{u}_i \right) \right], \tag{105}$$

where we let

$$\hat{\boldsymbol{x}}_j = \boldsymbol{x} + h \sum_{i=1}^{j} a_{j,i} \boldsymbol{u}_i. \tag{106}$$

Which can easily be found through standard reverse-mode autodifferentiation frameworks; likewise, the gradients for multiple Euler steps can be found.

## G.2   A PyTorch pseudocode illustration

Below in Code G.1 we provide an example PyTorch implementation of the greedy guidance strategy with an Euler scheme for sampling the main ODE.

```python
# posterior step
def posterior_step(t, x, n_steps=1)
    tlist = [0, 1]
    dt = tlist[1] - t

    if method == 'euler':
        # Euler step
        x = x + model(t, x) * dt
    elif method == 'midpoint':
        # Midpoint step
        x = x + dt * model(t + 0.5 * dt, x + 0.5 * dt * model(t, x))
    elif method == 'ralston3':
        # Ralston's third order
        k1 = model(t, x)
        k2 = model(t + 0.5 * dt, x + 0.5 * dt * k1)
        k3 = model(t + 0.75 * dt, x + 0.75 * dt * k2)

        x = x + dt * (2/9 * k1 + 1/3 * k2 + 4/9 * k3)
    elif method == 'heun3':
        # Heun's third order
        k1 = model(t, x)
        k2 = model(t + 1/3 * dt, x + 1/3 * dt * k1)
        k3 = model(t + 2/3 * dt, x + 2/3 * dt * k2)

        x = x + dt * (1/4 * k1 + 3/4 * k3)

    elif method == 'rk4':
        # RK-4 step
        k1 = model(t, x)
        k2 = model(t + 0.5*dt, x + 0.5*dt*k1)
        k3 = model(t + 0.5*dt, x + 0.5*dt*k2)
        k4 = model(t + dt, x + dt*k3)

        x = x + dt * (1/6 * k1 + 1/3 * k2 + 1/3 * k3 + 1/6 * k4)

    elif method == 'multiple_euler':
        # k Euler steps
        Dt = tlist[1] - t
        dt = Dt / n_steps
        ts = torch.linspace(t, tlist[1], n_steps + 1)

        for t in ts:
            x = x + model(t, x) * dt

    return x

# posterior guidance pseudocode
# assumed dt and loss are defined

xt = x0

for t in timesteps:
    # quick fix
    xt_opt = xt.detach().clone().requires_grad_(True)

    if optim == 'sgd':
        optimizer = torch.optim.SGD([xt_opt], lr=lr)
    else:
        optimizer = torch.optim.LBFGS([xt_opt], max_iter=max_iter, lr=lr, line_search_fn='strong_wolfe')

    for step in range(opt_steps):
        optimizer.zero_grad()

        x1hat = posterior_step(t, xt_opt)

        loss(x1hat).mean().backward()
        optimizer.step()

    with torch.no_grad():
        xt = xt_opt + model(t, xt_opt) * dt

return xt
```

Code G.1: Example implementation of greedy guidance

# H   A brief introduction to inverse problems

Inverse problems cover a large class of scientific problems (Chung, J. Kim, et al. 2023) that encompass scenarios where a partial measurement $y$ is made of $x$. When the mapping $x \mapsto y$ is not an injection,

recovering $\boldsymbol{x}$ from $\boldsymbol{y}$ becomes an ill-posed inverse problem. Generally, the relationship between the underlying sample $\boldsymbol{x}$ and the measurement $\boldsymbol{y}$ is given by

$$\boldsymbol{y} = \mathcal{A}(\boldsymbol{x}) + \boldsymbol{\eta}, \qquad \boldsymbol{y}, \boldsymbol{\eta} \in \mathbb{R}^{d_y}, \boldsymbol{x} \in \mathbb{R}^{d_x}, \tag{107}$$

where $\mathcal{A} : \mathbb{R}^{d_x} \to \mathbb{R}^{d_y}$ is the forward measurement operator and $\boldsymbol{\eta} \sim (0, \beta_{\boldsymbol{y}}^2 \boldsymbol{I})$ is the measurement noise.

> The inverse problem then is to find $p(\boldsymbol{x}|\boldsymbol{y})$.

More details on these types of problems can be found in Chung, J. Kim, et al. (2023), Moufad et al. (2025), and B. Zhang et al. (2025).

### H.1 Inverse problems and diffusion models

Recall that the ODE formulation of diffusion models is just a particular type of affine Gaussian probability path (Lipman, Havasi, et al. 2024). Following the conventions of the EDM model (Karras, Aittala, et al. 2022) we write this ODE formulation, known in the literature as the *probability flow ODE*, below in

$$\mathrm{d}\boldsymbol{x}_t = -\dot{\sigma}_t \sigma_t \nabla_{\boldsymbol{x}_t} \log p(\sigma_t, \boldsymbol{x}_t) \, \mathrm{d}t, \tag{108}$$

where $p(\sigma_t, \boldsymbol{x}_t)$ is the joint distribution of $\boldsymbol{x}_t$ at noise level $\sigma_t$.[11] *N.B.*, for diffusion models $\mathrm{d}t$ is a *negative* timestep and we integrate in reverse-time from $T$ to $0$. These models are also called *score-based generative models* due to learning the score function $\nabla_{\boldsymbol{x}_t} \log p(\sigma_t, \boldsymbol{x}_t)$.

One of the insights of Chung, J. Kim, et al. (2023) and Y. Song, Sohl-Dickstein, et al. (2021) is to apply Bayes' theorem for inverse problems to score-based generative models, *i.e.*,

$$p(\boldsymbol{x}|\boldsymbol{y}) = \frac{p(\boldsymbol{y}|\boldsymbol{x})p(\boldsymbol{x})}{p(\boldsymbol{y})}, \tag{109}$$

$$\nabla_{\boldsymbol{x}} \log p(\boldsymbol{x}|\boldsymbol{y}) = \nabla_{\boldsymbol{x}} \log p(\boldsymbol{x}) + \nabla_{\boldsymbol{x}} \log p(\boldsymbol{y}|\boldsymbol{x}). \tag{110}$$

Adapting this for diffusion models, assuming $\mathcal{A}$ is defined on $\boldsymbol{x}_0$ (the output), we have

$$\nabla_{\boldsymbol{x}_t} \log p(\sigma_t, \boldsymbol{x}_t|\boldsymbol{y}) = \nabla_{\boldsymbol{x}_t} \log p(\sigma_t, \boldsymbol{x}_t) + \nabla_{\boldsymbol{x}_t} \log p(\boldsymbol{y}|\boldsymbol{x}_t, \sigma_t). \tag{111}$$

The unconditional score term is the regular score function learned by diffusion models and thus is *appropriately* learned; however, the other term is much more difficult to work with. The approach of Chung, J. Kim, et al. (2023) is to use an approximation of

$$p(\boldsymbol{y}|\boldsymbol{x}_t, \sigma_t) = \mathbb{E}_{\boldsymbol{x}_0 \sim p(\boldsymbol{x}_0|\boldsymbol{x}_t)}[p(\boldsymbol{y}|\boldsymbol{x}_0, \sigma_0)], \tag{112}$$

via Tweedie's formula (Stein 1981) to write

$$p(\boldsymbol{y}|\boldsymbol{x}_t, \sigma_t) \approx p(\boldsymbol{y}|\mathbb{E}[\boldsymbol{x}_0|\boldsymbol{x}_t], \sigma_0). \tag{113}$$

The approximation error can be quantified by the Jensen gap (Chung, J. Kim, et al. 2023, Theorem 1).

## I   Experimental details

We provide additional details of the experiments performed in Section 7. *N.B.*, for all experiments we used fixed random seeds between the different software components to ensure a fair comparison.

### I.1   Inverse image problems

**Inverse problems.**   The inverse problems are implemented in the same way as in B. Zhang et al. (2025). We reiterate some of the important settings below. For Gaussian and motion deblurring we made use of kernels of size $61 \times 61$ with standard deviations of 3.0 and 0.5 respectively. The box inpainting task makes use of a random box of size $128 \times 128$ to mask the original images, while the random inpainting task randomly masks each pixel with a probability of 70% following (B. Song et al. 2024). The measure for the high dynamic range reconstruction problem is defined as

$$\boldsymbol{y} \sim \mathcal{N}(\mathrm{clip}(\alpha \boldsymbol{x}_0, -1, 1), \beta_{\boldsymbol{y}}^2 \boldsymbol{I}), \tag{114}$$

with $\alpha = 2$.

---

[11]This $\sigma_t$ is not the same as the $\sigma_t$ from the scheduler $(\alpha_t, \sigma_t)$ used in the main paper.

**Diffusion model.** We make use of the pre-trained diffusion model from Chung, J. Kim, et al. (2023), trained on the FFHQ $256 \times 256$ dataset. We focus on the probability flow ODE formulation popularized by Karras, Aittala, et al. (2022) known as EDM described as

$$\mathrm{d}\boldsymbol{x}_t = -\dot{\sigma}_t \sigma_t \nabla_{\boldsymbol{x}_t} \log p(\sigma_t, \boldsymbol{x}_t) \, \mathrm{d}t. \tag{115}$$

Following Ben-Hamu et al. (2024), we employ a midpoint scheme to solve this ODE in *reverse-time* with $N = 20$ steps. We use the noise schedule $\sigma_t = t$ which means $\dot{\sigma}_t = 1$. The discretized noise schedule $\{\sigma_n\}_{n=1}^N$ is given by the following polynomial interpolation

$$\sigma_n = \left( \sigma_{\max}^{\frac{1}{\rho}} + \frac{n}{N-1} \left( \sigma_{\min}^{\frac{1}{\rho}} - \sigma_{\max}^{\frac{1}{\rho}} \right) \right)^{\rho}. \tag{116}$$

We use $\rho = 7$, $T = \sigma_{\max} = 100$, and $\epsilon = \sigma_{\min} = 0.01$ for all experiments and integrate over $[\epsilon, T]$. *N.B.*, truncating the integration domain at $\epsilon$ rather than 0 is quite common in diffusion models (Y. Song, Dhariwal, et al. 2023).

**Hyperparameters.** Unlike previous works (B. Zhang et al. 2025) we did not adjust the hyperparameters per task and left them the same throughout. The learning rate was set at $\eta = 1$ for all experiments, and we performed $n_{\mathrm{opt}} = 50$ optimization steps with the stock implementation of the `torch.optim.SGD` method for each step of the ODE solve. We set $\beta_{\boldsymbol{y}} = 0.05$ for all tasks.

**Ablation study.** For the ablation study in Table 2 we used the L-BGFS optimizer over the standard SGD optimizer used in the main experiments (for the greedy guidance runs). For full DTO we used SGD as it provided better performance over L-BGFS in that scenario. *N.B.*, due to compute limitations we couldn't run DTO for step sizes larger than 8. Importantly, we fix the maximum number of optimization steps between the greedy and DTO strategies; for greedy we take 5 optimization step per step in the ODE solver, so a 100 in total. Likewise, for DTO we take a 100 optimization steps in total.

## I.2 Molecule generation for QM9

We follow the experimental methodology taken in previous work (Ben-Hamu et al. 2024; L. Wang et al. 2025) and follow the conditional generation pipeline used by Hoogeboom et al. (2022). An equivariant *graph neural network* (GNN) was trained for each property on half of the QM9 dataset, serving as a classifier—this model was then used as a guidance function during the experiments. The EquiFM (Y. Song, Gong, et al. 2023) model was trained on the whole QM9 training set and was used as the underlying flow model for the experiments. Following L. Wang et al. (2025), the test time properties were sampled from the whole training set; in contrast to Ben-Hamu et al. (2024).

Following Ben-Hamu et al. (2024) we used the L-BFGS algorithm (D. C. Liu and Nocedal 1989) with 5 optimizer steps and 5 inner steps with a linear search, in particular we used the stock PyTorch implementation `torch.opt.LBFGS`. For the DTO experiment we used a learning rate of $\eta = 1$. We tried this for the posterior guidance experiments but encountered severe instability. We found that a learning rate of $\eta = 0.001$ seemed to work better.

Recall that Proposition 5.2 states that the greedy gradient is scaled by the covariance projection. This effect is lessened as $t \to 1$, thus in later timesteps the greedy gradient is more likely to push samples off the data manifold. We observed this, with exploding losses even at small learning rates. To remedy this, we took inspiration from other works (Chung, J. Kim, et al. 2023; Moufad et al. 2025; Yu et al. 2023) and annealed the learning rate. We chose the following simple scheduler:

$$\eta_t = \begin{cases} \eta(1-t) & t > 0.5 \\ 0 & t \le 0.5 \end{cases}, \tag{117}$$

where $\eta = 0.001$ is the base learning rate.

**Runge-Kutta 4.** Additionally, we ran some experiments using RK4 but ran into insurmountable stability issues. Recall that RK4 is given by

$$k_1 = u_\theta\left(t_n, x_n\right),\tag{118}$$

$$k_2 = u_\theta\left(t_n + \frac{h}{2}, x_n + \frac{h}{2}k_1\right),\tag{119}$$

$$k_3 = u_\theta\left(t_n + \frac{h}{2}, x_n + \frac{h}{2}k_2\right),\tag{120}$$

$$k_4 = u_\theta\left(t_n + h, x_n + hk_3\right),\tag{121}$$

$$x_{n+1} = x_n + \frac{h}{6}(k_1 + 2k_2 + 2k_3 + k_4).\tag{122}$$

Using the step size $h = 1 - t$ we encountered large stability issues with the $k_4$ term due to being evaluated at the endpoint of the flow model trajectory. We tried a mixed-solver scheme were we would start with Euler and then switch to RK4, but that did not help. We also tried the common diffusion trick of truncated the time interval to $[0, 1 - \epsilon]$ for some small $\epsilon > 0$, but this did not solve the stability issues either. Ultimately, we abandoned it for this work and left such explorations for future work. It seems reasonable to suppose that schemes which don't evaluate on the endpoint, *e.g.*, Ralston's method, Heun's third-order method, or Ralton's third-order method may fair better.

### I.3 Numerical schemes

We detail the numerical schemes used for posterior guidance beyond Euler.

**Midpoint.** The midpoint scheme used in both experiments is implemented as

$$x_1 = x_t + hu_\theta\left(t + \frac{h}{2}, x_t + \frac{h}{2}u_\theta(t, x_t)\right)\tag{123}$$

with step size $h = 1 - t$.[12]

**2-step Euler.** This scheme used in both experiments is implemented as

$$x_{t + \frac{h}{2}} = x_t + \frac{h}{2}u_\theta(t, x_t),\tag{124}$$

$$x_1 = x_{t + \frac{h}{2}} + \frac{h}{2}u_\theta\left(t + \frac{h}{2}, x_{t + \frac{h}{2}}\right),\tag{125}$$

with step sizes $h = 1 - t$.

### I.4 Hardware and compute cost

**Inverse image problems.** The inverse image problem experiments were run on a single NVIDIA H100 80GB GPU. It took roughly 4 minutes and 78 GB of VRAM to generate 10 images for each inverse problem. As such each experiment took about an 40–50 minutes. Experiments which used the midpoint method, unsurprisingly ran about 90% slower.

**Molecule generation.** The molecule generation experiments were run on a single NVIDIA V100 16GB GPU. It took about 3 minutes and 1.5 GB of VRAM to generate 1 molecule leading to the experiments taking on the order of 300 minutes to complete. Experiments which used the midpoint method, unsurprisingly ran about 90% slower.

## J  Further experimental results

We present additional experimental results that we could not include in the main paper for the sake of space.

---

[12]This is appropriately adjusted for diffusion models with a terminal time of 0.

## J.1 Molecule generation for QM9

In Table 5 we present the *atom stability percentage* (ASP) and *molecule stability percentage* (MSP) per property for each guided generation model. Interestingly, despite their poor quantitative performance in MAE (*cf*. Table 3) the greedy (midpoint) and (2-step Euler) strategies have slightly better stability than DTO.

Table 5: Stability reported in ASP/MSP per property.

| Property | $\alpha$ | $\Delta\varepsilon$ | $\varepsilon_{\mathrm{HOMO}}$ | $\varepsilon_{\mathrm{LUMO}}$ | $\mu$ | $C_v$ |
|---|---|---|---|---|---|---|
| DTO | 94.90/65.00 | 96.20/74.00 | 95.90/67.00 | 96.00/65.00 | 94.60/61.00 | 95.00/67.00 |
| Greedy (Euler) | 94.70/68.80 | 96.40/76.00 | 97.40/79.00 | 98.40/84.80 | 97.60/84.00 | 85.55/21.20 |
| Greedy (midpoint) | 97.46/80.00 | 97.51/83.00 | 97.91/81.00 | 97.77/83.00 | 97.70/81.00 | 97.09/80.00 |
| Greedy (2-step Euler) | 97.67/82.00 | 96.95/74.00 | 98.18/84.00 | 96.29/72.00 | 97.40/93.00 | 97.75/84.00 |
| EquiFM | 98.88/89.00 | | | | | |

## J.2 Further results on inverse image problems

To put the results from Section 7.1 into context we present some detailed comparisons to other works from the domain of inverse problems with diffussion models, namely:

1. DAPS (B. Zhang et al. 2025),
2. DPS (Chung, J. Kim, et al. 2023),
3. DDRM (Kawar et al. 2022),
4. DDNM (Y. Wang, Yu, and J. Zhang 2023),
5. DCDP (Li et al. 2024),
6. FPS-SMC (Dou and Y. Song 2024),
7. DiffPIR (Zhu et al. 2023), and
8. RED-diff (Mardani et al. 2024).

We present the full comparison in Table 6.

Table 6: Additional results for inverse image problems on FFHQ $256 \times 256$.

| Task | Method | PSNR (↑) | SSIM (↑) | LPIPS (↓) | FID (↓) |
|------|--------|----------|----------|-----------|---------|
| Super resolution 4× | Greedy (Euler) | 27.94 | 0.728 | 0.217 | 66.64 |
| | Greedy (midpoint) | 27.98 | 0.727 | 0.224 | 70.96 |
| | Greedy (2-step Euler) | 27.95 | 0.728 | 0.220 | 68.93 |
| | DAPS | 29.07 | 0.818 | 0.177 | 51.44 |
| | DPS | 25.86 | 0.753 | 0.269 | 81.07 |
| | DDRM | 26.58 | 0.782 | 0.282 | 79.25 |
| | DDNM | 28.03 | 0.795 | 0.197 | 64.62 |
| | DCDP | 28.66 | 0.807 | 0.178 | 53.81 |
| | FPS-SMC | 28.42 | 0.813 | 0.204 | 49.25 |
| | DiffPIR | 26.64 | - | 0.260 | 65.77 |
| Inpaint (box) | Greedy (Euler) | 23.74 | 0.732 | 0.187 | 46.87 |
| | Greedy (midpoint) | 24.08 | 0.724 | 0.186 | 44.55 |
| | Greedy (2-step Euler) | 23.88 | 0.720 | 0.188 | 44.09 |
| | DAPS | 24.07 | 0.814 | 0.133 | 43.10 |
| | DPS | 22.51 | 0.792 | 0.209 | 61.27 |
| | DDRM | 22.26 | 0.801 | 0.207 | 78.62 |
| | DDNM | 24.47 | 0.837 | 0.235 | 46.59 |
| | DCDP | 23.89 | 0.760 | 0.163 | 45.23 |
| | FPS-SMC | 24.86 | 0.823 | 0.146 | 48.34 |
| Inpaint (random) | Greedy (Euler) | 30.87 | 0.823 | 0.141 | 40.73 |
| | Greedy (midpoint) | 31.03 | 0.816 | 0.139 | 38.80 |
| | Greedy (2-step Euler) | 30.80 | 0.811 | 0.144 | 39.23 |
| | DAPS | 31.12 | 0.844 | 0.098 | 32.17 |
| | DPS | 25.46 | 0.823 | 0.203 | 69.20 |
| | DDNM | 29.91 | 0.817 | 0.121 | 44.37 |
| | DCDP | 30.69 | 0.842 | 0.142 | 52.51 |
| | FPS-SMC | 28.21 | 0.823 | 0.261 | 61.23 |
| Gaussian deblurring | Greedy (Euler) | 28.01 | 0.766 | 0.182 | 57.04 |
| | Greedy (midpoint) | 28.36 | 0.776 | 0.185 | 58.55 |
| | Greedy (2-step Euler) | 28.18 | 0.774 | 0.181 | 57.18 |
| | DAPS | 29.19 | 0.817 | 0.165 | 53.33 |
| | DPS | 25.87 | 0.764 | 0.219 | 79.75 |
| | DDRM | 24.93 | 0.732 | 0.239 | 92.43 |
| | DDNM | 28.20 | 0.804 | 0.216 | 57.83 |
| | DCDP | 27.50 | 0.699 | 0.304 | 86.43 |
| | FPS-SMC | 26.54 | 0.773 | 0.253 | 67.45 |
| | DiffPIR | 27.36 | - | 0.236 | 59.65 |
| Motion deblurring | Greedy (Euler) | 29.35 | 0.748 | 0.207 | 63.05 |
| | Greedy (midpoint) | 29.73 | 0.762 | 0.207 | 66.21 |
| | Greedy (2-step Euler) | 29.64 | 0.764 | 0.203 | 63.99 |
| | DAPS | 29.66 | 0.847 | 0.157 | 39.49 |
| | DPS | 24.52 | 0.801 | 0.246 | 65.23 |
| | DCDP | 25.08 | 0.512 | 0.364 | 125.13 |
| | FPS-SMC | 27.39 | 0.826 | 0.227 | 48.32 |
| | DiffPIR | 26.57 | - | 0.255 | 65.78 |
| Phase retrieval | Greedy (Euler) | 15.10 | 0.282 | 0.598 | 298.06 |
| | Greedy (midpoint) | 15.10 | 0.286 | 0.595 | 299.45 |
| | Greedy (2-step Euler) | 15.07 | 0.284 | 0.598 | 304.60 |
| | DAPS | $30.63_{\pm 3.13}$ | $0.851_{\pm 0.072}$ | $6.139_{\pm 0.060}$ | 42.71 |
| | DPS | $17.64_{\pm 2.97}$ | $0.441_{\pm 0.129}$ | $0.410_{\pm 0.090}$ | 104.52 |
| | RED-diff | $15.60_{\pm 4.48}$ | $0.398_{\pm 0.195}$ | $0.596_{\pm 0.092}$ | 167.43 |
| | DCDP | $28.65 \pm 8.09$ | $0.781_{\pm 0.217}$ | $0.203_{\pm 0.196}$ | 68.13 |
| Nonlinear deblur | Greedy (Euler) | 24.767 | 0.551 | 0.327 | 79.06 |
| | Greedy (midpoint) | 25.09 | 0.558 | 0.332 | 76.73 |
| | Greedy (2-step Euler) | 24.81 | 0.547 | 0.330 | 76.26 |
| | DAPS | $28.29_{\pm 1.77}$ | $0.783_{\pm 0.036}$ | $0.155_{\pm 0.032}$ | 49.38 |
| | DPS | $23.39_{\pm 2.01}$ | $0.623_{\pm 0.082}$ | $0.278_{\pm 0.060}$ | 91.31 |
| | RED-diff | $30.86_{\pm 0.51}$ | $0.795_{\pm 0.028}$ | $0.160_{\pm 0.034}$ | 43.84 |
| | DCDP | $27.92_{\pm 2.64}$ | $0.779_{\pm 0.067}$ | $0.183_{\pm 0.051}$ | 51.96 |
| High dynamic range | Greedy (Euler) | 24.16 | 0.767 | 0.181 | 43.59 |
| | Greedy (midpoint) | 26.62 | 0.809 | 0.160 | 37.86 |
| | Greedy (2-step Euler) | 25.70 | 0.797 | 0.165 | 37.97 |
| | DAPS | $27.12_{\pm 3.53}$ | $0.752_{\pm 0.041}$ | $0.162_{\pm 0.072}$ | 42.97 |
| | DPS | $22.73_{\pm 6.07}$ | $0.591_{\pm 0.141}$ | $0.264_{\pm 0.156}$ | 112.82 |
| | RED-diff | $22.16_{\pm 3.41}$ | $0.512_{\pm 0.083}$ | $0.258_{\pm 0.089}$ | 108.32 |

## J.3 Sampling trajectories for inverse problems

We present the solution trajectories the different guidance algorithms below for solving the HDR inverse problem. Note that the midpoint and 2-step Euler, unsurprisingly, have better approximations of $\boldsymbol{x}_1$.

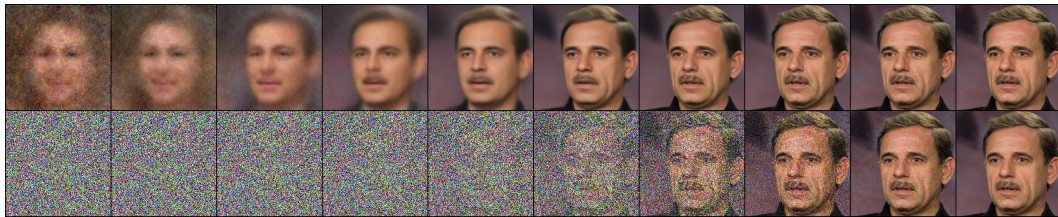

Figure 6: Sampling trajectory for greedy (Euler) solving the HDR inverse problem. Top row is $\boldsymbol{x}_{1|t}^{\theta}(\boldsymbol{x}_t)$ and the bottom row is $\boldsymbol{x}_t$.

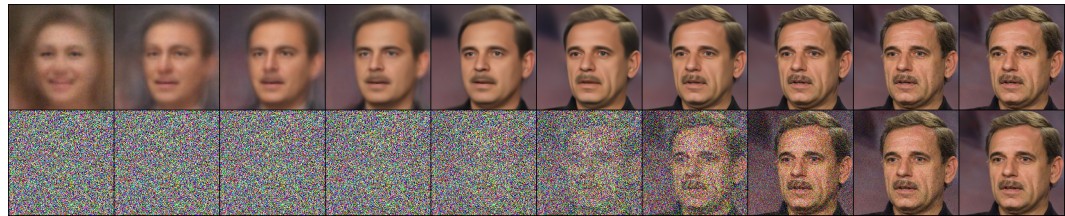

Figure 7: Sampling trajectory for greedy (midpoint) solving the HDR inverse problem. Top row is midpoint estimate and the bottom row is $\boldsymbol{x}_t$.

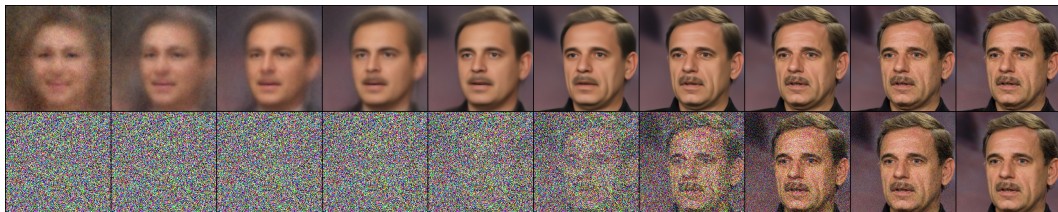

Figure 8: Sampling trajectory for greedy (2-step Euler) solving the HDR inverse problem. Top row is 2-step Euler estimate and the bottom row is $\boldsymbol{x}_t$.

## J.4 More qualitative samples for inverse problems

We showcase some examples generated by the greedy gradient strategy (Euler) on the different inverse problems.

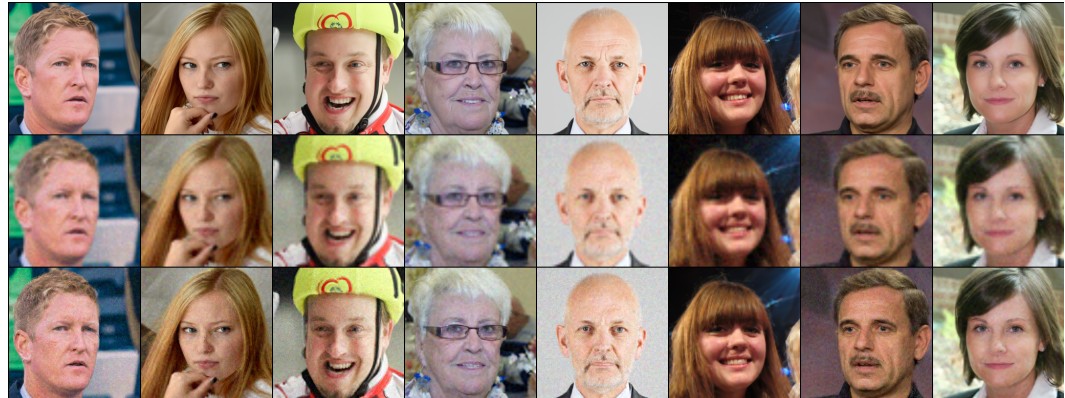

Figure 9: Qualitative visualization of using greedy guidance to solve the super resolution $4\times$ inverse problem. Top row is the ground truth, middle row is the measurement, and the bottom row is the reconstruction.

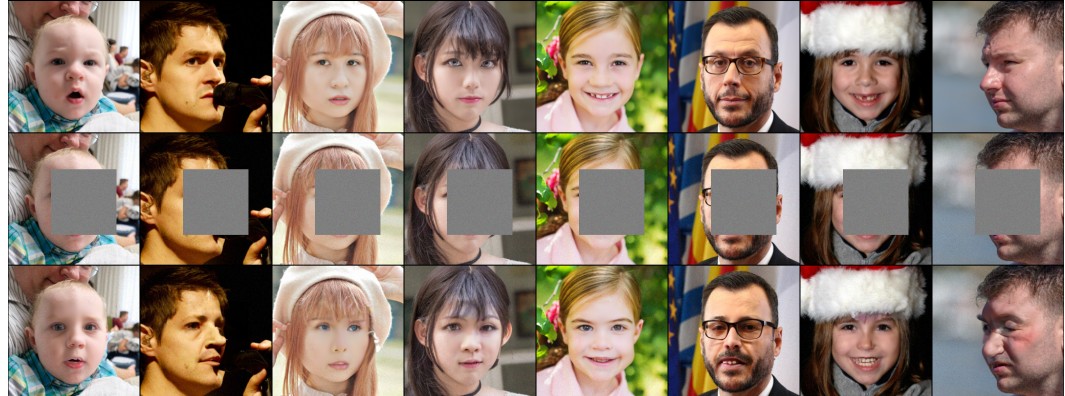

Figure 10: Qualitative visualization of using greedy guidance to solve the super resolution $4\times$ inverse problem. Top row is the ground truth, middle row is the measurement, and the bottom row is the reconstruction.

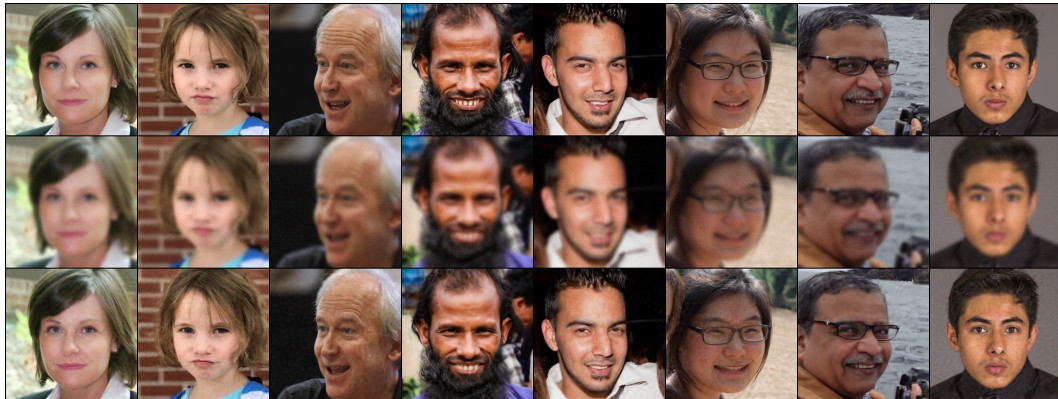

Figure 11: Qualitative visualization of using greedy guidance to solve the Gaussian deblurring inverse problem. Top row is the ground truth, middle row is the measurement, and the bottom row is the reconstruction.

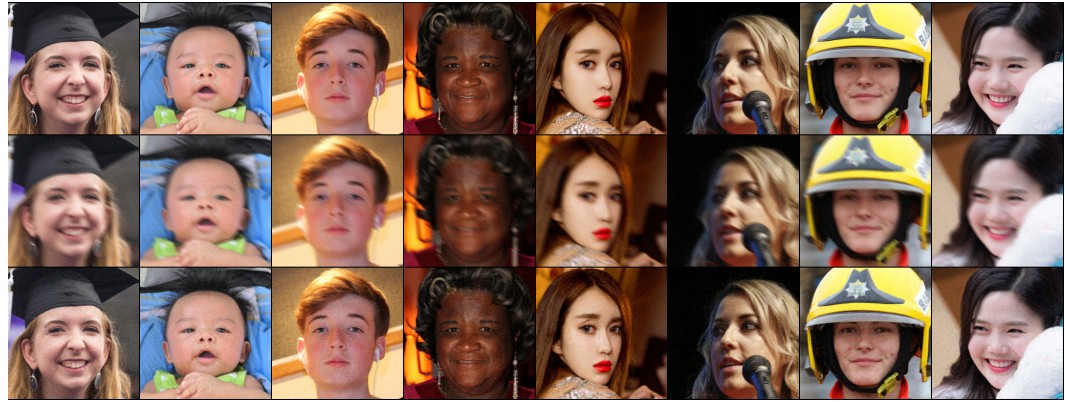

Figure 12: Qualitative visualization of using greedy guidance to solve the motion deblurring inverse problem. Top row is the ground truth, middle row is the measurement, and the bottom row is the reconstruction.

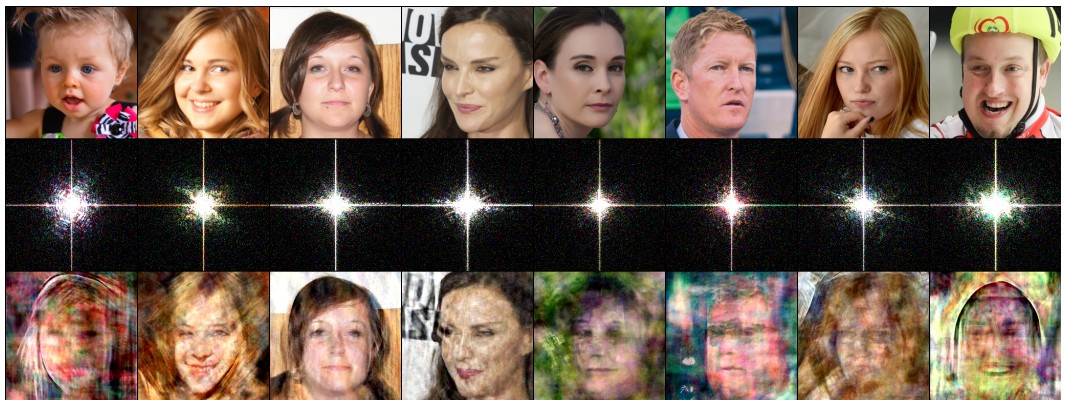

Figure 13: Qualitative visualization of using greedy guidance to solve the Phase retrieval inverse problem. Top row is the ground truth, middle row is the measurement, and the bottom row is the reconstruction.

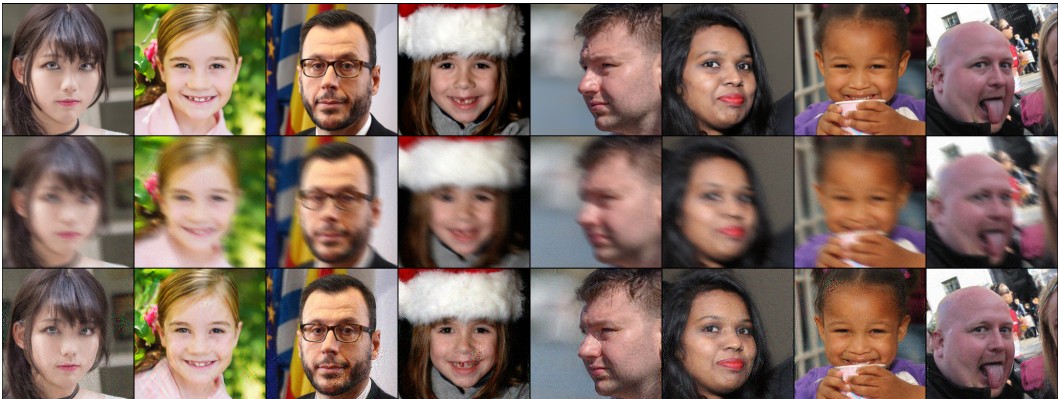

Figure 14: Qualitative visualization of using greedy guidance to solve the nonlinear deblurring inverse problem. Top row is the ground truth, middle row is the measurement, and the bottom row is the reconstruction.

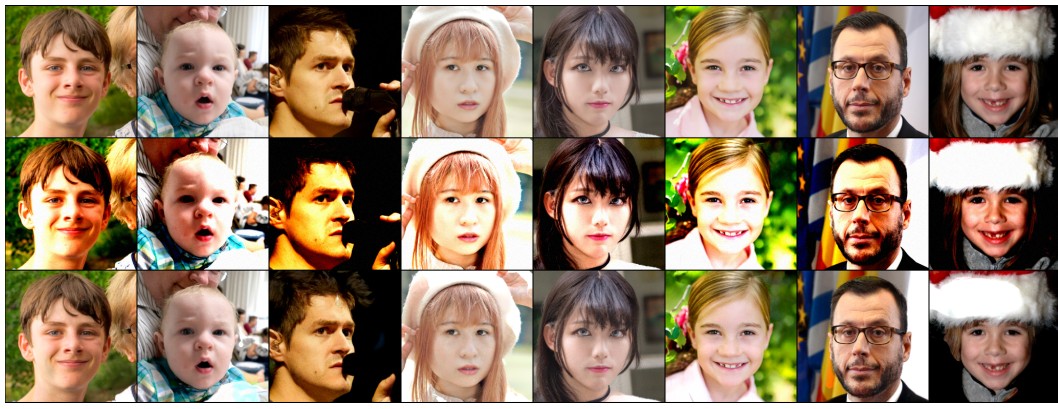

Figure 15: Qualitative visualization of using greedy guidance to solve the HDR inverse problem. Top row is the ground truth, middle row is the measurement, and the bottom row is the reconstruction.

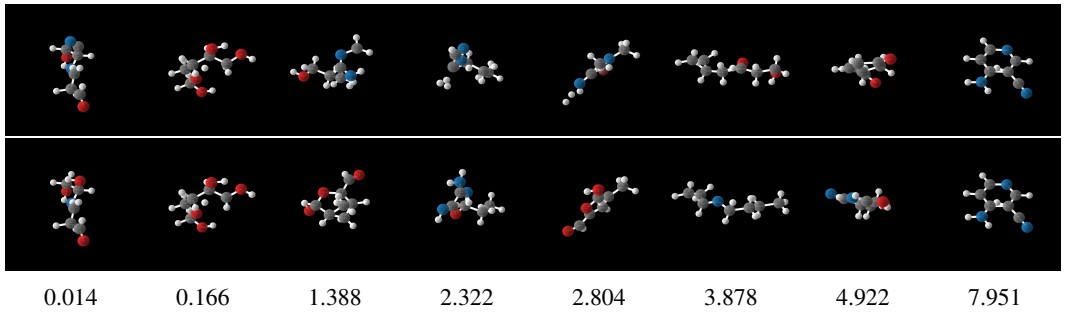

| 0.014 | 0.166 | 1.388 | 2.322 | 2.804 | 3.878 | 4.922 | 7.951 |

Figure 16: Qualitative visualization of controlled generated molecules for various dipole moments ($\mu$). Top row is generated using a end-to-end guidance with a DTO scheme and the bottom row is generated using posterior guidance.

### J.5  More qualitative samples for controlled molecule generation

In Appendix J.4 we present some qualitative results for property-guided molecule generation. In particular, we target different dipole moments.

## K  Discussions

### K.1  Broader Impacts

Controllable generation can be used for many tasks both benign and malicious. The insights from this paper could be used to develop more effective adversarial attacks, generation of harmful content, or other malicious applications.

### K.2  Limitations

As this work is mostly theoretical, our experimental illustrations are limited, serving more to illustrate the key concepts rather than advancing the state-of-the-art within the particular problem. We believe that future work can use these insights to make informed design choices when developing solutions to guided generation problems.

In our controllable molecule generation experiments, we take a naïve strategy for annealing the learning rate leaving performance on the table. Moreover, we don't consider mixed accuracy schemes, *i.e.*, using Euler for certain steps closer to the target and midpoint for steps further away (*cf*. Moufad et al. 2025).

