# OpenReview forum: "Greed is Good: A Unifying Perspective on Guided Generation"
_NeurIPS.cc/2025/Conference — NeurIPS 2025 poster_

### Official Review · Reviewer_Hyh7 · 2025-07-01

**Clarity:** 3
**Significance:** 2
**Originality:** 2
**Rating:** 2
**Confidence:** 4

**Summary:**

The paper shows that posterior guidance is a greedy approximation of end-to-end guidance, and presents a unifying framework for training-free guided generation in flow/diffusion models. The authors derive error bounds (Theorems 5.4, 6.1) linking greedy and exact gradients, and propose higher-order solver to trade off compute vs. accuracy. Experiments on inverse imaging and molecule generation control illustrate the performance of new insight.

**Questions:**

1.Failure Analysis: The greedy method fails badly for some molecule generation. Please analyze why (e.g. non-convexity, learning-rate issues).

2. Baseline Comparison: Explicitly compare to similar recent works(“greedy (midpoint)” v.s. midpoint guidance (Moufad et al., 2025)) to better position your hybrids.

References
Moufad, B., Janati, Y., Bedin, L., Durmus, A. O., Douc, R., Moulines, E., and Olsson, J. (2025). “Variational Diffusion Posterior Sampling with Midpoint Guidance.” Proceedings of the 13th International Conference on Learning Representations. URL: https://openreview.net/forum?id=6EUtjXAvmj (cit. on pp. 17, 18, 32, 34, 42)

**Ethical Concerns:**

["NO or VERY MINOR ethics concerns only"]

**Final Justification:**

After the rebuttal, I now better appreciate that the assumptions made are standard, and I acknowledge the additional analyses and experiments the authors have provided. However, I remain firm in my evaluation that the paper’s empirical support is too weak to fully endorse its significance for real-world guided generation tasks. My critique was always about empirical competitiveness and stability-and on that front, the responses, while informative, show that the proposed approach is still playing catch-up to existing methods (in some cases by a wide margin). At present, given the continued concerns about empirical robustness and competitiveness, I regret that I must maintain my original rating.

**Limitations:**

Yes.

**Quality:**

2

**Strengths And Weaknesses:**

Strengths
1. First to cast posterior guidance as a greedy end-to-end approximation under certain assumptions.
2. Rigorous theoretical analysis with clear theorems on greedy vs. full gradients and then derive error bounds (Theorems 5.4, 6.1).

Weaknesses
1. Results rely on affine paths and smoothness assumptions. Real-world flow/diffusion models often include non-affine terms or non-Gaussian noise, so it’s unclear whether the proofs can extend to more complex settings.
2. Experiments are only proof of concept and do not demonstrate clear practical superiority.Limited empirical evidence of real-world benefit and underperform SOTA.For example, in Table 1 Greedy (2-step Euler) falls behind DAPS(B. Zhang et al. 2024) by only a narrow margin, whereas the basic Greedy algorithm performs significantly worse than D-flow(Ben-Hamu et al.2024).
3. Experimental illustrations are limited,offers error bounds but no empirical runtime vs. quality trade-off Figures to guide method selection.The experimental evaluation lacks sufficient task diversity, omitting important scenarios such as editing and conditional generation.

References
Ben-Hamu, H., Puny, O., Gat, I., Karrer, B., Singer, U., and Lipman, Y. (2024). “D-Flow: Differentiating through Flows for Controlled Generation.” Proceedings of the 41st International Conference on Machine Learning. URL: https://openreview.net/forum?id=SE20BFqj6J (cit. on pp. 1–3, 6, 8, 9, 17–19, 21, 22, 24, 33, 34)
Zhang, B., Chu, W., Berner, J., Meng, C., Anandkumar, A., and Song, Y. (2024). “Improving Diffusion Inverse Problem Solving with Decoupled Noise Annealing.” arXiv preprint arXiv:2407.01521 [cs.LG]. URL: https://arxiv.org/abs/2407.01521 (cit. on pp. 7, 8, 17, 32–35)

---

> ### Author Rebuttal · Authors · 2025-07-30
>
> We thank reviewer Hyh7 for their detailed feedback and constructive suggestions.
> We are glad that they find our contribution of casting "posterior guidance as a greedy approximation of end-to-end guidance" to be novel and that they appreciate our "rigorous theoretical analysis with clear theorems."
> We address the concerns and questions raised by the reviewer below.
>
> > Results rely on affine paths and smoothness assumptions. Real-world flow/diffusion models often include non-affine terms or non-Gaussian noise, so it’s unclear whether the proofs can extend to more complex settings.
>
> We wish to clarify that real world flow/diffusion models **do** in fact use affine terms and a Gaussian source distribution.
> Diffusion models (in the ODE form) all fall within this formulation (remember that diffusion models diffuse the data distribution to a Gaussian distribution) and *most* flow models choose to use a Gaussian source distribution. We refer the reviewer to [1, Section 4.8] for more details.
> Moreover, many models are trained used the model of affine probability paths.
> This analysis is standard and has been used in prior works [2].
>
> Thus these assumptions in **no way** hinder the application or utility of our theoretical results in real world scenarios.
>
> > Experiments are only proof of concept and do not demonstrate clear practical superiority.Limited empirical evidence of real-world benefit and under perform SOTA.For example, in Table 1 Greedy (2-step Euler) falls behind DAPS(B. Zhang et al. 2024) by only a narrow margin, whereas the basic Greedy algorithm performs significantly worse than D-flow(Ben-Hamu et al.2024).
>
> We wish to emphasis that the **core contribution of this paper is not to outperform the SOTA but to connect end-to-end and posterior guidance together.**
> Thus our performance is bounded by these two approaches and we explore the connection between them.
> Thus our core significant contribution is the rigorous mathematical analysis in unifying these two families together.
>
> > Experimental illustrations are limited,offers error bounds but no empirical runtime vs. quality trade-off Figures to guide method selection.The experimental evaluation lacks sufficient task diversity, omitting important scenarios such as editing and conditional generation.
>
> While we did not explore experiments applications such as editing, we do explore conditional generation in **Section 7.2** and in **Appendices H.2 and I.1**.
> Furthermore, we would like to reiterate that our experimental scope is in line with other works in the space [2-5].
>
> To explore performance vs complexity tradeoff we would like to direct the reviewer's attention to **Theorem 5.1, Proposition 5.2, and Theorem 6.1** which explore several facets of the problem.
>
> **1.** In **Theorem 5.1** and **Proposition 5.2** we show that the greedy guidance strategy results in a covariance projection of the loss function; however, this projection is weaker and less significant as $t \to 1$. Thus for later time steps the greedy gradient is more likely to push samples off the data manifold.
> We posit this is why there was little stability concerns for the inverse image problems; whereas the the controlled molecule generation problem is less resilient to being pushed away from the data manifold. This makes intuitive sense as the very small changes could result in invalid molecules.
> This corroborates with the analyses of prior work [2].
>
> **2.** Then in **Theorem 6.1** we show that the accuracy of the gradients is bound by the local truncation error of the underlying numerical scheme; however, for large step sizes higher-order solvers may be *less* suitable.
> This however, depends on the the straightness of the underlying ODE, if the true underlying ODE is perfectly straight (highly idealized, but presented for illustrative purposes) then all the schemes would have no error.
>
> Additionally, we have run some additional experiments to help clear up this demonstration empirically.
>
> | Property                | $\alpha$          | $\Delta \varepsilon$ | $\varepsilon_{\mathrm{HOMO}}$ | $\varepsilon_{\mathrm{LUMO}}$ | $\mu$ | $C_v$                                                |
> |:-------------------------|:-------------------:|:----------------------:|:-------:|:-------:|:-------:|:------------------------------------------------------:|
> | Greedy (Euler)          | 11.282            | 1265                 | 725                           | 1092                          | 1.559 | 6.469                                                |
> | Greedy (midpoint)       | 5.313             | 1196                 | 599                           | 1057                          | 1.417 | 2.967                                                |
> | Greedy (2-step Euler) | 5.377             | 1275                 | 560                           | 1204                          | 1.563 | 2.975                                                |
> | Greedy (3-step Euler)   | 5.098             | 1152                 | 600                           | 1152                          | 1.384 | 3.229                                                |
> | Greedy (5-step Euler)   | 4.177             | 1083                 | 571                           | 939                           | 1.328 | 2.332                                                |
> | DTO                     | 1.404             | 401                  | 176                           | 373                           | 0.372 | 0.866     |
> | EquiFM                  | 9.525             | 1494                 | 622                           | 1523                          | 1.628 | 6.689                                                |
> | Lower bound             | 0.10              | 64                   | 39                            | 46                            | 0.043 | 0.040                                                |
>
> We note that generally, taking more discretization steps resulted in a more *useful* gradient. This corroborates with recent work which explored taking two Euler steps [1] (*N.B.*, they approach the problem from a different perspective and use different terminology, but the equivalence is valid).
>
>
> > Failure Analysis: The greedy method fails badly for some molecule generation. Please analyze why (e.g. non-convexity, learning-rate issues).
>
> We discuss the instability more in Appendix H.2, but to summarize it relates to **Proposition 5.2** and the greedy gradient being pushed off the data manifold for $t \gg 0$.
> We provide a more detailed response below:
>
> In **Theorem 5.1** and **Proposition 5.2** we show that the greedy guidance strategy results in a covariance projection of the loss function; however, this projection is weaker and less significant as $t \to 1$. Thus for later time steps the greedy gradient is more likely to push samples off the data manifold.
> We posit this is why there was little stability concerns for the inverse image problems; whereas the the controlled molecule generation problem is less resilient to being pushed away from the data manifold. This makes intuitive sense as the very small changes could result in invalid molecules.
> This corroborates with the analyses of prior work [2].
>
> > Baseline Comparison: Explicitly compare to similar recent works(“greedy (midpoint)” v.s. midpoint guidance (Moufad et al., 2025)) to better position your hybrids.
>
> The recent work of Moufad et al., 2025 can be viewed as analagous to the "Greedy (2-step Euler)" experimental runs and fits nicely within our framework.
> We wish to note that this work was approached from a different perspective than ours; however, it ties in nicely to our analysis in the "Beyond Euler" section.
> We discuss this connection in greater detail in **Appendix A.1**.
>
> [1] Lipman, Y., Havasi, M., Holderrieth, P., Shaul, N., Le, M., Karrer, B., Chen, R. T., Lopez-Paz,
> D., Ben-Hamu, H., and Gat, I. (2024). “Flow Matching Guide and Code”. In: arXiv preprint
> arXiv:2412.06264
>
> [2] Ben-Hamu, H., Puny, O., Gat, I., Karrer, B., Singer, U., and Lipman, Y. (2024). “D-Flow: Differentiating through Flows for Controlled Generation.” Proceedings of the 41st International Conference on Machine Learning. URL: https://openreview.net/forum?id=SE20BFqj6J
>
> [3] Zhang, B., Chu, W., Berner, J., Meng, C., Anandkumar, A., and Song, Y. (2024). Improving Diffusion
> Inverse Problem Solving with Decoupled Noise Annealing. arXiv: 2407.01521 [cs.LG]. URL:
> https://arxiv.org/abs/2407.01521
>
> [4] Blasingame and Liu, C. (2024). “AdjointDEIS: Efficient Gradients for Diffusion Models”. In:
> Advances in Neural Information Processing Systems. Ed. by A. Globerson, L. Mackey, D. Bel-
> grave, A. Fan, U. Paquet, J. Tomczak, and C. Zhang. Vol. 37. Curran Associates, Inc., pp. 2449–
> 2483.
>
> [5] Wang, L., Cheng, C., Liao, Y., Qu, Y., and Liu, G. (2025). “Training Free Guided Flow-Matching
> with Optimal Control”. In: The Thirteenth International Conference on Learning Representations.
> URL: https://openreview.net/forum?id=61ss5RA1MM
>
> ## Closing remarks
> We thank reviewer Hyh7 for their suggestions on how to improve our work.
> We hope that our detailed responses helped clarify the strength of our contributions and alleviate any concerns that the reviewer had concerning our experimental work.
> We are more than happy to consider any additional questions or further suggestions.
>
> We kindly ask the reviewer to consider increasing their score if our responses address their concerns satisfactorily.

---

> > ### Comment · Reviewer_Hyh7 · 2025-08-03
> >
> > 1. Clarifying the Focus: Empirical Utility vs. Theoretical Scope
> >
> > “Thus these assumptions in no way hinder the application or utility of our theoretical results in real world scenarios.”I agree with this assessment. My initial note about these assumptions was not a fundamental critique, but rather a prompt to confirm the theory’s practical scope. I am satisfied that the theoretical framework is sound, and I have no concerns about the theoretical assumptions.
> >
> > I appreciate the rigorous theory - my primary concern has always been the empirical side. The rebuttal suggests the authors interpreted my critique as expecting state-of-the-art performance or questioning the theoretical setup. To clarify: I fully recognize that the core contribution is a unifying theoretical perspective, not beating SOTA benchmarks. My critique was that while the theory is elegant and general, the experimental evidence provided did not convince me that the approach works robustly or competitively in real-world settings. I agree with the authors that surpassing SOTA is not the goal here; however, for the contribution to have a practical impact, the method’s reliability and performance should at least be convincingly strong (even if not absolute SOTA) on representative tasks.
> >
> > 2. Empirical Results and Performance Gap
> >
> > The authors’ analysis of the molecule generation instability is well taken.Small deviations in molecule space can indeed yield invalid molecules, so a greedy, approximate update is more prone to produce chemically incorrect results - an observation the authors note is consistent with prior work.I appreciate this theoretical explanation (based on Proposition 5.2) as it offer insight on why the method struggled on that task. However, it also reinforces a practical limitation: for domains like molecular design (which are highly sensitive to off-manifold perturbations), the greedy approach may inherently be at a disadvantage stability-wise.In other words, the insight that greedy guidance is prone to instability in molecule generation highlights the need for further research (perhaps integrating constraints or adaptive step sizes) before this approach can be considered robust for those use cases.
> >
> >  The experimental results remain a point of concern. In particular, the guided generation method’s performance on the molecule generation task (Table 2 of the paper) was quite weak, which undermines confidence in its real-world utility. The rebuttal confirms that our approach did not demonstrate clear practical superiority over existing methods. The updated results provided by the authors reinforce this: the greedy (Euler) approach for molecule property control produces errors that are much higher than those of baseline methods. Even after introducing multi-step guidance, there remains a considerable performance gap. The authors’ own data show, for instance, that a 5-step Greedy Euler reduces the error relative to 1-step, yet α is still about 4.177 vs. 1.404 for the baseline DTO method. This is an improvement over the single-step greedy (which was 11.282), but it is still nearly 3 times higher than DTO approach.
> >
> > This persistent gap in empirical performance is the crux of my concern: it suggests that the unified perspective, while intellectually valuable, has not yet translated into a method that one would choose over existing techniques for at least 1 challenging task.
> >
> > 3. Experimental Scope and Practical Validation
> >
> > The authors addressed some of my comments on experimental scope by noting that they did include a conditional generation experiment (in Section 7.2 and Appendix) and that their chosen tasks align with related works. I acknowledge this, and it partially mitigates my concern.
> >
> > However, other aspects of practical validation are still lacking. Notably, as I originally pointed out, there were no empirical runtime-versus-quality trade-off figures provided. The authors direct attention to theoretical results (Theorems 5.1, 6.1) for insights on this trade-off, which is intellectually satisfying, but it is not a substitute for concrete empirical evidence. For a practitioner or reader, it would be very useful to see a plot or table illustrating how increasing the number of guidance steps or using higher-order solvers improves output quality versus the computational cost incurred(partly offered in this rebuttal). Such an experiment would directly demonstrate the method’s value proposition of trading compute for accuracy. Similarly, while matching the scope of prior works is reasonable, exploring an additional scenario (e.g., an image editing task or a different conditional generation setting) would have strengthened the paper’s empirical section.

---

> > > ### Author Response · Authors · 2025-08-04
> > >
> > > We thank the reviewer for their detailed response and would like to make a few clarifications.
> > >
> > > ---
> > > > 2.  Empirical Results and Performance Gap
> > >
> > > We are happy that the reviewer appreciates our analysis of the molecule generation instability.
> > >
> > > To address the concerns about Table 2, we actually believe that it **illustrates the strength** of our paper. *I.e.*, it shows by moving between the posterior guidance strategy towards the end-to-end guidance strategy we approach the upper bound (full DTO) in performance.
> > > As the reviewer stated "the core contribution is a unifying theoretical perspective, not beating SOTA benchmarks." and thus being able to move between posterior guidance and end-to-end guidance we are able to illustrate the trade-off empirically.
> > >
> > > To alleviate concerns about the empirical performance we include the table from our response to reviewer oWM7 which further explores this trade-off. We explore only taking the greedy step at the initial time $t = 0$ and then explore adding more steps until reaching the **upper bound** of the full 50 for the standard DTO. (Note we still use 50 for the final sampling once the optimal $\boldsymbol x_0^*$ has been found)
> > >
> > > | Property                | $\alpha$          | $\Delta \varepsilon$ | $\varepsilon_{\mathrm{HOMO}}$ | $\varepsilon_{\mathrm{LUMO}}$ | $\mu$ | $C_v$                                                |
> > > |-------------------------|-------------------|----------------------|-------------------------------|-------------------------------|-------|------------------------------------------------------|
> > > | DTO (1-step)            | 13.049            | $989 \times 10^{12}$ | 681                           | 86.512                        | 1.666 | 15.144                                               |
> > > | DTO (2-step)            | 6.113             | 1359                 | 666                           | 1199                          | 1.533 | 3.757                                                |
> > > | DTO (4-step)            | 6.115             | 1294                 | 668                           | 1190                          | 1.406 | 2.829                                                |
> > > | DTO (8-step)            | 4.549             | 1070                 | 608                           | 1078                          | 1.247 | 2.594                                                |
> > > | DTO (16-step)           | 3.454             | 817                  | 608                           | 939                           | 1.177 | 2.003                                                |
> > > | DTO (32-step)           | 2.912             | 750                  | 410                           | 666                           | 0.721 | 1.566                                                |
> > > | DTO (40-step)           | 2.384             | 625                  | 372                           | 556                           | 0.719 | 1.425                                                |
> > > | DTO (50-step)                     | 1.404             | 401                  | 176                           | 373                           | 0.372 | 0.866                                                |
> > >
> > > We would like to emphasis that DTO with the full 50 steps is **an upper bound to performance** within our design axes.
> > >
> > > We hope this helps address the concerns the reviewer has over the performance gap.
> > >
> > > ---
> > > > 3. Experimental Scope and Practical Validation
> > >
> > > Unfortunately, we are not able to include such figures during review process. We address the concerns in more detail below concerning this trade-off:
> > >
> > > * We believe our results *clearly demonstrate* this trade-off. We point to the table above and in the tables in the main paper we compare the amount of compute (number of discretization steps) compared to empirical performance.
> > >
> > > * Additionally, in **Appendix E, Table 3** we discuss the compute and memory requirements in more detail.
> > >
> > > * The compute costs can be straightforwardly computed from the number of steps, we will include this in the final paper and can be appended to the tables to illustrate the runtime-versus-quality dynamics.
> > > ---
> > > > it suggests that the unified perspective, while intellectually valuable, has not yet translated into a method that one would choose over existing techniques for at least 1 challenging task.
> > >
> > > Our **unified analysis** shows that both theoretically and empirically we can exchange compute for improved gradient accuracy which lies at the heart of our method. In **Appendix I.2, Table 5** we see that our methods often keep up in line with the SOTA despite being not considering additional design components like Langevin dynamics.
> > >
> > > We hope that these answers clarify any concerns about the empirical portion of our work. We would be happy to discuss further and answer any more questions about our work.

---

### Official Review · Reviewer_oWM7 · 2025-07-03

**Clarity:** 2
**Significance:** 3
**Originality:** 3
**Rating:** 4
**Confidence:** 3

**Summary:**

The paper provides a unifying theoretical framework for two seemingly distinct families of training-free guided generation techniques used in flow and diffusion models: posterior guidance and end-to-end guidance. Both techniques control the generative process, often by injecting gradients from a target prediction model to steer the generation towards desired properties. The authors analyze both approaches and show that posterior guidance can be framed as a greedy gradient step within the end-to-end scheme.

**Questions:**

- Some reference to prominent approaches are missing. Can the authors explain the connections between the analysed methods and e.g., \[1\] and \[2\]?

\[1\] [\[2311.16424\] Manifold Preserving Guided Diffusion](https://arxiv.org/abs/2311.16424)

\[2\] [Pseudoinverse-Guided Diffusion Models for Inverse Problems](https://openreview.net/forum?id=9_gsMA8MRKQ)

**Ethical Concerns:**

["NO or VERY MINOR ethics concerns only"]

**Final Justification:**

The authors have been responsive and addressed my concerns.
With the suggested modifications and additions to the manuscript (add pseudo code in main paper, additional experiments exploring further the interpolation space between the methods, etc.) I think the paper present a valuable analysis.

**Limitations:**

yes

**Quality:**

3

**Strengths And Weaknesses:**

**Strengths**

- Theoretical Unification: The paper's primary strength lies in the theoretical framework that unifies posterior guidance and end-to-end guidance. This provides a deeper understanding of their relationship and is a significant conceptual contribution to the field of guided generation.

**Weaknesses**

- Experiments:
  - While the authors show results of experiments for 3 variants in the interpolation space (Euler, midpoint, 2-Euler), I find the experimental setup lacking. Given the theoretical connection that were drawn between the posterior guidance approach and the end-to-end approach, a more elaborate ablation on the various interpolation intermediate steps between the two would greatly improve the understanding of the practical implications of the theoretical findings. Do the experimental results behave as one would expect? Is there anything surprising, as many theoretical results do not transfer to practice in high dimensional regims, and with neural networks.
  - One of the claims is that this interpolation allows to trade quality and efficiency, it is not clearly demonstrated in the paper.
- It would be valuable to better position this paper in existing literature, see e.g., [\[2410.00083\] A Survey on Diffusion Models for Inverse Problems](https://arxiv.org/abs/2410.00083)
- Clarity: it is hard to parse what the different methods are, adding algorithm boxes with pseudo code would improve ease of understanding.

---

> ### Author Rebuttal · Authors · 2025-07-31
>
> We thank reviewer oWM7 for their detailed feedback and constructive suggestions.
> We are happy that appreciated that the "paper's primary strength lies in the theoretical framework that unifies posterior guidance and end-to-end guidance."
> Below we provide the suggested ablations and experiments along with answering the reviewer's questions.
>
> >  While the authors show results of experiments for 3 variants in the interpolation space (Euler, midpoint, 2-Euler), I find the experimental setup lacking. Given the theoretical connection that were drawn between the posterior guidance approach and the end-to-end approach, a more elaborate ablation on the various interpolation intermediate steps between the two would greatly improve the understanding of the practical implications of the theoretical findings. Do the experimental results behave as one would expect? Is there anything surprising, as many theoretical results do not transfer to practice in high dimensional regims, and with neural networks.
>
> We agree with the reviewer that such ablations would be a valuable addition to the paper.
> We ran some additional experiments on the controlled molecule generation task with high-order solvers and found that increasing the number of steps did improve the performance of the model at the cost of more compute.
> See our discussion with reviewer XpGA for a more detailed analysis of these results.
>
>
> **Table 1.** We compare the effect of taking more discretization steps in the greedy guidance framework.
>
> | Property                | $\alpha$          | $\Delta \varepsilon$ | $\varepsilon_{\mathrm{HOMO}}$ | $\varepsilon_{\mathrm{LUMO}}$ | $\mu$ | $C_v$                                                |
> |:-------------------------|:-------------------:|:----------------------:|:-------:|:-------:|:-------:|:------------------------------------------------------:|
> | Greedy (Euler)          | 11.282            | 1265                 | 725                           | 1092                          | 1.559 | 6.469                                                |
> | Greedy (midpoint)       | 5.313             | 1196                 | 599                           | 1057                          | 1.417 | 2.967                                                |
> | Greedy (2-step Euler) | 5.377             | 1275                 | 560                           | 1204                          | 1.563 | 2.975                                                |
> | Greedy (3-step Euler)   | 5.098             | 1152                 | 600                           | 1152                          | 1.384 | 3.229                                                |
> | Greedy (5-step Euler)   | 4.177             | 1083                 | 571                           | 939                           | 1.328 | 2.332                                                |
> | DTO                     | 1.404             | 401                  | 176                           | 373                           | 0.372 | 0.866     |
>
> We also ran an additional set of experiments wherein we explored a differing number of discretization steps within a DTO guidance scheme, ranging from 1 (*i.e.* posterior guidance only applied to $\boldsymbol x_0$) to the full 50 used in the D-Flow model.
> These results further showed that we can exchange compute cost with performance (and accuracy of gradients).
>
> **Table 2.** We compare the effect of a different number of discretization steps for optimizing $\boldsymbol x_0$.
>
> | Property                | $\alpha$          | $\Delta \varepsilon$ | $\varepsilon_{\mathrm{HOMO}}$ | $\varepsilon_{\mathrm{LUMO}}$ | $\mu$ | $C_v$                                                |
> |:-------------------------|:-------------------:|:----------------------:|:-------:|:-------:|:-------:|:------------------------------------------------------:|
> | DTO (1-step)            | 13.049            | $989 \times 10^{12}$ | 681                           | 86.512                        | 1.666 | 15.144                                               |
> | DTO (2-step)            | 6.113             | 1359                 | 666                           | 1199                          | 1.533 | 3.757                                                |
> | DTO (4-step)            | 6.115             | 1294                 | 668                           | 1190                          | 1.406 | 2.829                                                |
> | DTO (8-step)            | 4.549             | 1070                 | 608                           | 1078                          | 1.247 | 2.594                                                |
> | DTO (16-step)           | 3.454             | 817                  | 608                           | 939                           | 1.177 | 2.003                                                |
> | DTO (32-step)           | 2.912             | 750                  | 410                           | 666                           | 0.721 | 1.566                                                |
> | DTO (50-step)                     | 1.404             | 401                  | 176                           | 373                           | 0.372 | 0.866                                                |
>
> > One of the claims is that this interpolation allows to trade quality and efficiency, it is not clearly demonstrated in the paper.
>
> We justify this claim in **Theorems 5.3 and 6.1**.
> We hope that our additional experiments discussed above help illustrate this further experimentally.
>
> > It would be valuable to better position this paper in existing literature, see e.g., [2410.00083] A Survey on Diffusion Models for Inverse Problems
>
> We thank the reviewer for suggesting the survey paper, it is quite interesting.
> We wish to note, however, that beyond already referencing several of the papers mentioned in the survey, that the focus of our paper lies beyond just diffusion models or inverse problems as we are interested more generally in flow models.
> We also wish to remind the reviewer that we did positions our paper in light of several papers on posterior guidance and end-to-end guidance, see **Appendix A** and **Figure 5** in the paper.
> We would be happy to add this survey to our literature review if accepted.
>
> > Clarity: it is hard to parse what the different methods are, adding algorithm boxes with pseudo code would improve ease of understanding.
>
> We thank the reviewer for this suggestion and will incorporate this feedback if the paper is accepted.
> We wish to note that we already detailed the numerical schemes in **Appendix H.3.**
>
> > Some reference to prominent approaches are missing. Can the authors explain the connections between the analysed methods and e.g., [1] and [2]?
>
> We thank the reviewer for highlighting these works.
>
> **1. MPGD.**
> From [1, Theorem 1] we see that *Manifold Preserving Guided Diffusion* (MPGD) update rule can be written in our notation as
>
> $\begin{equation}
> \boldsymbol x_{1|t}' = \boldsymbol x_{1|t}^\theta(\boldsymbol x_t) - c_t \nabla_{\boldsymbol x_1}\mathcal L\left(\boldsymbol x_{1|t}^\theta(\boldsymbol x)\right).
> \end{equation}$
>
> Thus we see that MPGD applies the gradient to the output of the denoiser.
> We can then apply this to the data prediction formulation (*cf.* **Proposition 4.1**, *n.b.*, MPGD is originally discussed in the context of the discretized diffusion SDE but we focus on flow models in our work).
>
> We actually discuss nearly the same algorithm in **Appendix F**, but from another perspective (and of course in the context of ODEs).
> This can be viewed as an instance of greedy *control signal optimization* which we discuss in **Appendix F.**
> In our discussion in **Appendix F** we discuss how applying this technique to the denoiser results in rescaling $\nabla_{\boldsymbol x_1}\mathcal L(\boldsymbol x_1)$ by $b_t$.
> We will update our discussion to include a reference to MPGD.
> We thank the reviewer to pointing out this connection.
>
>
> **2. Pseudoinverse guidance.**
> From [2, Eqs. (6) and (7)] we see that [2] focuses specifically on solving inverse problems and uses a known measurement matrix $\boldsymbol H$ to project the Jacobian $\nabla_{\boldsymbol x}\boldsymbol x_{1|t}^\theta(\boldsymbol x)$ to time $t$. That is, this a specialized formulation which amounts to projecting the loss by the Jacobian of the denoiser, this is quite similar to the greedy gradient discussed in our paper.
>
> We would also like to remind the reviewer that we discuss [2] in **Appendix A / Figure 5.**
>
> **Summary.**
> We wish to note that both works are focused on learning the score function $\nabla_{\boldsymbol x_t} \log p(\boldsymbol y|\boldsymbol x_t)$ whereas our work takes an **entirely different** perspective and considers flow models in addition to the ODE formulation of diffusion models.
>
> ## Closing remarks
> We thank the reviewer of the detailed feedback and questions.
> We especially appreciate the additional related work the reviewer highlighted.
> We hope our responses helped answer the questions and clarified the strengths of our contributions.
> We would be happy to consider any additional questions or further suggestions.
>
> We kindly ask the reviewer to consider increasing their score if our responses address their concerns satisfactorily.

---

> > ### Comment · Reviewer_oWM7 · 2025-08-05
> >
> > I thank the author for their response and effort addressing my questions.
> >
> > I have a few follow-up comments/questions:
> >
> > 1. With regards to results shown in Table 1 and Table 2 above, does a similar trend appear for inverse problems on images where the dimension of the data is higher? i.e., results improve with higher order solvers?
> >
> > 2. Have the authors checked how DTO performs on images with growing number of steps like in Table 2 in reponse above?
> >
> > 3. Clarity
> > > We thank the reviewer for this suggestion and will incorporate this feedback if the paper is accepted. We wish to note that we already detailed the numerical schemes in Appendix H.3.
> >
> > I am aware of these, but I believe a simple pseudo code in the essence of the code the authors shared with Reviewer RAY9, would greatly improve the readability the clarity of the paper.
> >
> > 4. Another question I am interested in is - whether there are types of problems where DTO would fail, but the greedy approach will succeed? I would imagine in setups where little imformation is given in the observable for images for inatcne, that this may be the case? say where half of the image is masked.

---

> > > ### Author Response · Authors · 2025-08-05
> > >
> > > We are happy the reviewer appreciated our initial response and we are happy to address the follow up questions.
> > >
> > > ---
> > > > 1. With regards to results shown in Table 1 and Table 2 above, does a similar trend appear for inverse problems on images where the dimension of the data is higher? i.e., results improve with higher order solvers?
> > > > 2. Have the authors checked how DTO performs on images with growing number of steps like in Table 2 in reponse above?
> > >
> > > During the rebuttal process we began with the controlled molecule generation experiments. We are currently running experiments for inverse problems but due to compute limits we do not yet have the results. We will update this comment if we obtain them in time.
> > >
> > > ---
> > > > 3. I am aware of these, but I believe a simple pseudo code in the essence of the code the authors shared with Reviewer RAY9, would greatly improve the readability the clarity of the paper.
> > >
> > > We agree and would be happy to include these in the final paper.
> > >
> > > ---
> > > > 4. Another question I am interested in is - whether there are types of problems where DTO would fail, but the greedy approach will succeed? I would imagine in setups where little imformation is given in the observable for images for inatcne, that this may be the case? say where half of the image is masked.
> > >
> > > Theoretically, **no**, in practice **maybe?** To clarify, the full DTO scheme *is the perfect and optimal upper bound of performance*, with respect to the estimation of the gradient. So within our design axes it is indeed the upper bound. However, *in practice*, DTO is **very memory intensive** with a memory cost of $\mathcal O(nd^2)$ where $n$ is the number of steps and $d$ is the dimensionality of the data. The memory cost of greedy is much smaller as we take fewer steps. So in some scenarios the memory cost could become prohibitive. We discuss these considerations more in **Appendix E**, but for the purposes of our paper DTO is the **upper bound** of performance when it is computational tractable.
> > >
> > > We would be happy to discuss further and answer any more questions about our work.

---

> > > > ### Comment · Reviewer_oWM7 · 2025-08-06
> > > >
> > > > > During the rebuttal process we began with the controlled molecule generation experiments. We are currently running experiments for inverse problems but due to compute limits we do not yet have the results. We will update this comment if we obtain them in time.
> > > >
> > > > I would appreciate it if the author post these results if they get them before discussion period ends. I think these will give a complete image on how and if theoretical findings transfer to empirical results in various domains (i.e., molecules and images).
> > > >
> > > > If authors can commit to adding these results to an updated manuscript I will lean towards accepting the paper, as I think the thoretical work is important and having the suitable complamentray experimental results will benefit the paper greatly.

---

> > > > > ### Author Response · Authors · 2025-08-09
> > > > > **Additional experiments on inverse image problems**
> > > > >
> > > > > We were able to finish several experimental runs on the *non-linear* inverse problem of high dynamic range (HDR) reconstruction introduced in [1]. The problem is to "recover a higher dynamic range image (factor of 2) from a low dynamic range image" [1].
> > > > >
> > > > > We present our more complete ablations (running more than 2-steps) below along with the prior works of DPS, DAPS, and RED-diff from the original table. Additionally, unlike our original table we reran the Euler and 2-step Euler for 4 runs as suggested by the DAPs paper for reporting results on *non-linear* problems.
> > > > >
> > > > > **Table 3.** Results on HDR inverse image problem of greedy strategy with varying number of discretization steps.
> > > > >
> > > > > | **Method** | **PSNR** $(\uparrow)$ | **SSIM** $(\uparrow)$ | **LPIPS** $(\downarrow)$ | **FID** $(\downarrow)$ |
> > > > > |:-----------|:-------:|:-------:|:-------:|:-------:|
> > > > > | DAPS | ${2 7 . 1 2}_{ \pm 3.53}$ | ${0 . 7 5 2}_{ \pm 0.041}$ | ${0 . 1 6 2}_{ \pm 0.072}$ | 42.97|
> > > > > | DPS | ${22.73}_{ \pm 6.07}$ | ${0.591}_{ \pm 0.141}$ | $0.264_{ \pm 0.156}$ | 112.82|
> > > > > | RED-diff | $22.16_{ \pm 3.41}$ | $0.512_{ \pm 0.083}$ | ${0.258}_{ \pm 0.089}$ | ${108.32}$|
> > > > > | Greedy (Euler) | $25.07_{\pm 4.25}$ | $0.776_{\pm 0.126}$ | $0.173_{\pm 0.070}$ | 43.25|
> > > > > | Greedy (2-step Euler) | $26.32_{\pm 4.34}$ | $0.802_{\pm 0.111}$ | $0.173_{\pm 0.065}$ | 38.64|
> > > > > | Greedy (3-step Euler) | $27.17_{\pm 4.21}$ | $0.820_{\pm 0.096}$ | $0.154_{\pm 0.062}$ | 36.07|
> > > > > | Greedy (4-step Euler) | $27.89_{\pm 4.10}$ | $0.828_{\pm 0.092}$ | $0.151_{\pm 0.061}$ | 36.94|
> > > > > | Greedy (5-step Euler) | $\mathbf{28.27} _{\pm 4.01}$ | $\mathbf{0.831}_{\pm 0.088}$ | $\mathbf{0.149}_{\pm 0.059}$ | **35.35**|
> > > > >
> > > > > We notice that:
> > > > > * The Greedy (5-step Euler) performs very well beating the SOTA DAPS (CVPR 2025) algorithm, even the 2-step and 3-step perform variants as well or better.
> > > > > * Increasing the number of discretization steps (unsurprisingly as predicted by our theory) leads to better performance.
> > > > > * Interestingly the standard deviation (reported as the subscript plusminus term) decreases as well with the results becoming more consistent.
> > > > >
> > > > > Now with regards to an analogue to Table 2 from our initial reply for controlled molecule generation we were only able to generate a few runs due to compute limitations. This helps to provide additional clarity to our initial response to
> > > > > > Another question I am interested in is - whether there are types of problems where DTO would fail, but the greedy approach will succeed? I would imagine in setups where little imformation is given in the observable for images for inatcne, that this may be the case? say where half of the image is masked.
> > > > >
> > > > > While the gradient provided by full DTO is **exactly** accurate it can be intractable to calculate it, as we were unable to without encountering OOM errors. As we discuss in **Appendix E**, there are alternatives around this, the adjoint method (inexact), reversible solvers (poor stability in the ODE solve), and recursive check-pointing (more compute time) all of which were beyond the scope of this ablation and our current resources during this discussion. As such we only were able to run up to a DTO scheme with 4 Euler steps. We have 8 steps currently running, but we doubt that will make the deadline for the discussion period.
> > > > >
> > > > > **Table 4.** Results on HDR inverse image problem of a DTO backprop strategy for optimizing $\boldsymbol x_0$
> > > > >
> > > > > | **Method** | **PSNR** $(\uparrow)$ | **SSIM** $(\uparrow)$ | **LPIPS** $(\downarrow)$ | **FID** $(\downarrow)$ |
> > > > > |:-----------|:-------:|:-------:|:-------:|:-------:|
> > > > > | DTO (1-step) | $13.16 _{\pm 1.15}$ | $0.372_{\pm 0.083}$ | $0.521_{\pm 0.059}$ | 108.39|
> > > > > | DTO (2-step) | $14.91 _{\pm 1.23}$ | $0.372_{\pm 0.080}$ | $0.483_{\pm 0.061}$ | 98.93|
> > > > > | DTO (4-step) | $16.37 _{\pm 1.38}$ | $0.455_{\pm 0.082}$ | $0.457_{\pm 0.066}$ | 93.52|
> > > > >
> > > > > We do observe a trend of increasing performance as we increase the number of steps, however, it far under-performs the greedy strategy for a similar compute budget.
> > > > >
> > > > > So to answer the reviewer's question: in the presence of limited compute and time it seems that the greedy strategy is preferable to full DTO on this problem (we noticed similar trends for other inverse image problems).
> > > > >
> > > > > We sincerely thank reviewer oWM7 for encouraging us to pursue these additional ablations and making our paper stronger.
> > > > >
> > > > > [1] Zhang, B., Chu, W., Berner, J., Meng, C., Anandkumar, A., and Song, Y. (2025). Improving Diffusion
> > > > > Inverse Problem Solving with Decoupled Noise Annealing. CVPR

---

### Official Review · Reviewer_RAY9 · 2025-07-03

**Clarity:** 2
**Significance:** 2
**Originality:** 2
**Rating:** 4
**Confidence:** 4

**Summary:**

The authors explore the theoretical connections between these two families and provide an in-depth theoretical understanding of these two techniques relative to the continuous ideal gradients. Motivated by this analysis, they then show a method for interpolating between these two families enabling a trade-off between compute and accuracy of the guidance gradients. they validate this work on some inverse image problems and property-guided molecular generation.

**Questions:**

1. How the theoretical analysis close the gap between practice and theory? The experimental results leave room for improvement.
2. Could more state-of-the-art methods be unified under the proposed framework to better demonstrate its practical relevance? Additionally, how does the framework concretely indicate potential directions for future advancements?

**Ethical Concerns:**

["NO or VERY MINOR ethics concerns only"]

**Final Justification:**

I appreciate the theoretical efforts of this paper and the detailed explanation from the authors. I have increased my score accordingly.

**Limitations:**

Yes, the authors have discussed the limitations and future work in the Appendix

**Quality:**

2

**Strengths And Weaknesses:**

### Strength

1. The theoretical part of the paper is structured clearly and presented in a concise and readable manner.
2. The paper presents substantial theoretical analysis on posterior guidance and end-to-end optimization, which could potentially inspire future advances in method design.

### Weakness

1. Including more illustrations and pseudocode would help readers better understand the overall workflow of the paper and how the proposed theoretical insights lead to concrete improvements.
2. While training-free guidance methods are known to face substantial limitations—including limited controllability, off-manifold generation, longer sampling requirements, and sensitivity to hyper-parameters—the paper does not provide further analysis or exploration toward addressing these challenges, which could have strengthened the contribution.
3. While the paper outlines future directions involving more hyperparameters and case-specific design choices, it is worth noting that training-free methods are already known to be highly sensitive to hyperparameters, which raises concerns regarding the feasibility and robustness of such approaches.

---

> ### Author Rebuttal · Authors · 2025-07-30
>
> We thank reviewer RAY9 for their time, feedback, and helpful suggestions in more clearly communicating our ideas.
> We are happy that the reviewer found the theoretical part of the paper to be "structured clearly and presented in a concise and readable manner", in addition to appreciating our "substantial theoretical analysis".
> Below, we address the reviewer's questions and concerns.
>
> > Including more illustrations and pseudocode would help readers better understand the overall workflow of the paper and how the proposed theoretical insights lead to concrete improvements.
>
> Yes, we agree that including more illustrations and pseudocode would strengthen the paper.
> Whilst, we cannot attach new figures in the rebuttal we can illustrate some pseudocode
> ```python
> # posterior step
> def posterior_step(t, x)
>     tlist = [0, 1]
>     dt = tlist[1] - t
>
>     if method == 'euler':
>         # Euler step
>         x = x + model(t, x) * dt
>     elif method == 'midpoint':
>         # Midpoint step
>         x = x + dt * model(t + 0.5 * dt, x + 0.5 * dt * model(t, x))
>     elif method == 'ralston3':
>         # Ralston's third order
>         k1 = model(t, x)
>         k2 = model(t + 0.5 * dt, x + 0.5 * dt * k1)
>         k3 = model(t + 0.75 * dt, x + 0.75 * dt * k2)
>
>         x = x + dt * (2/9 * k1 + 1/3 * k2 + 4/9 * k3)
>     elif method == 'heun3':
>         # Heun's third order
>         k1 = model(t, x)
>         k2 = model(t + 1/3 * dt, x + 1/3 * dt * k1)
>         k3 = model(t + 2/3 * dt, x + 2/3 * dt * k2)
>
>         x = x + dt * (1/4 * k1 + 3/4 * k3)
>
>     elif method == 'rk4':
>         # RK-4 step
>         k1 = model(t, x)
>         k2 = model(t + 0.5*dt, x + 0.5*dt*k1)
>         k3 = model(t + 0.5*dt, x + 0.5*dt*k2)
>         k4 = model(t + dt, x + dt*k3)
>
>         x = x + dt * (1/6 * k1 + 1/3 * k2 + 1/3 * k3 + 1/6 * k4)
>
>     return x
>
> # posterior guidance pseudocode
> # assumed dt and loss are defined
>
> xt = x0
>
> for t in timesteps:
>     # quick fix
>     xt_opt = xt.detach().clone().requires_grad_(True)
>
>     if optim == 'sgd':
>         optimizer = torch.optim.SGD([xt_opt], lr=lr)
>     else:
>         optimizer = torch.optim.LBFGS([xt_opt], max_iter=max_iter, lr=lr, line_search_fn='strong_wolfe')
>
>     for step in range(opt_steps):
>         optimizer.zero_grad()
>
>         x1hat = posterior_step(t, xt_opt)
>
>         loss(x1hat).mean().backward()
>         optimizer.step()
>
>     with torch.no_grad():
>         xt = xt_opt + model(t, xt_opt) * dt
>
> return xt
> ```
>
> Additionally, in **Appendix H.3** we outline the numerical schemes used for guidance.
> We would also like to note that our code is provided in the supplementary material.
>
> > While training-free guidance methods are known to face substantial limitations—including limited controllability, off-manifold generation, longer sampling requirements, and sensitivity to hyper-parameters—the paper does not provide further analysis or exploration toward addressing these challenges, which could have strengthened the contribution.
>
> We agree that these are all pertinent topics of great interest and appreciate the suggestions.
> However, we would like to respectfully argue that we did address some of these issues---albeit we need to make this more clear in the writing.
>
> **1. Limited controllability.**
> In **Theorem 5.4 and Corollary D.1.1** we show put a bound on *controllability* in the sense of how well the gradient can steer the output of the flow model towards a desired solution.
>
> **2. Off-manifold generation.**
> We address this concern in **Proposition 5.2**, wherein we show that the guidance strategy applied to earlier time steps $t \ll 1$ applies a covariance projection to the guidance signal encouraging samples to stay on the data manifold.
>
> **3. Longer sampling requirements.**
> While not fully explored in this work, we believe our analysis on exchanging the compute for calculating the guidance gradient and the accuracy of the gradient does shed some **new light** on this discussion.
>
> > While the paper outlines future directions involving more hyperparameters and case-specific design choices, it is worth noting that training-free methods are already known to be highly sensitive to hyperparameters, which raises concerns regarding the feasibility and robustness of such approaches.
>
> Yes, our work highlights another potential design axis introducing more hyperparameters.
> We believe the benefit of this work is **unifying** two widely used families for designing guided generation algorithms into a single family.
> Thus we inherit the robustness and feasibility of the two families.
> We believe our "substantial theoretical analysis" sheds some light on this.
>
> > How the theoretical analysis close the gap between practice and theory? The experimental results leave room for improvement.
>
> The experimental results are meant to illustrate the **connection between posterior and end-to-end guidance** rather than outperforming the SOTA.
> Thus the theoretical results is meant to explain the performance gap between the two techniques and then show how the performance can be bridged by taking more steps in approximating the gradient.
> We would like to emphasis that this perspective is **novel** and could help frame future research into the topics.
>
> > Could more state-of-the-art methods be unified under the proposed framework to better demonstrate its practical relevance? Additionally, how does the framework concretely indicate potential directions for future advancements?
>
> Several state-of-the-art models (D-Flow [1], AdjointDEIS [2], OC-Flow [3], and MGPS [4]) are all subsumed by this analysis, see **Appendix A.2** for D-Flow and AdjointDEIS, **Appendix F** for OC-Flow, and **Appendix A.1** for MGPS.
>
> As the difference between the inverse image problems and controlled molecule generation experiments illustrated, certain problem are more sensitive to samples being moved off the data manifold *cf.* **Proposition 5.2** which would indicate a DTO-type of strategy would be more promising.
> Moreover, one could explore more sophisticated posterior guidance schemes which use higher or lower order solvers dependent on the step size $h$.
>
> [1] Ben-Hamu, H., Puny, O., Gat, I., Karrer, B., Singer, U., and Lipman, Y. (2024). “D-Flow: Differentiating through Flows for Controlled Generation.” Proceedings of the 41st International Conference on Machine Learning. URL: https://openreview.net/forum?id=SE20BFqj6J
>
> [2] Blasingame and Liu, C. (2024). “AdjointDEIS: Efficient Gradients for Diffusion Models”. In:
> Advances in Neural Information Processing Systems. Ed. by A. Globerson, L. Mackey, D. Bel-
> grave, A. Fan, U. Paquet, J. Tomczak, and C. Zhang. Vol. 37. Curran Associates, Inc., pp. 2449–
> 2483.
>
> [3] Wang, L., Cheng, C., Liao, Y., Qu, Y., and Liu, G. (2025). “Training Free Guided Flow-Matching
> with Optimal Control”. In: The Thirteenth International Conference on Learning Representations.
> URL: https://openreview.net/forum?id=61ss5RA1MM
>
> [4] Moufad, B., Janati, Y., Bedin, L., Durmus, A. O., Douc, R., Moulines, E., and Olsson, J. (2025). “Variational Diffusion Posterior Sampling with Midpoint Guidance”. In: The Thirteenth International Conference on Learning Representations. URL: https://openreview.net/forum?id=6EUtjXAvmj
>
>
> ## Closing remarks
> We thank reviewer RAY9 for their suggestions on how to improve this paper.
> We hope that our responses fully address all the questions and concerns raised by the reviewer and we are happy to consider any additional questions.
>
> We kindly ask the reviewer to consider increasing their score if our responses address their concerns satisfactorily.

---

> > ### Comment · Reviewer_RAY9 · 2025-08-04
> > **Thanks for the detailed response !**
> >
> > I appreciate the efforts the authors has put in the rebuttal phase. I think the theoretical analysis in this paper would be beneficial but limited for developing new guidance methods. I want to kindly point out that the interpolation between posterior and end-to-end guidance is straightforward and empirically validated[1]. Currently I share similar perspective with Reviewer Hyh7. I appreciate the values of theoretical papers and if advantages over existing guidance methods are more clear (speed, robustness, performance, etc), the paper will be stronger.
> >
> >
> > [1] CoDe: Blockwise Control for Denoising Diffusion Models

---

> > > ### Author Response · Authors · 2025-08-04
> > >
> > > We are glad the reviewer appreciated our rebuttal. We also thank reviewer RAY9 for pointing out [1], since it was *recently* published during the time of our original submission we were unaware of it. It was quite interesting and we will include it in our discussion.
> > >
> > > We would like to clarify that while [1, Algorithm 1] does offer a perspective to move between posterior guidance and a form of end-to-end guidance via the $B$ parameter, to the best of our knowledge it does so from an *orthogonal perspective.* In particular, our work ties together traditional posterior guidance and works focusing on full backprop through diffusion models via DTO or the adjoint method.
> > > We noticed that the connection to this second family was absent from their analysis as that paper focused on other interesting areas.
> > >
> > > We attach an updated table with new results that we have used in some of our other discussion which we hope clears up any concerns on the empirical front. We explore only taking the greedy step at the initial time $t = 0$ and then explore adding more steps until reaching the **upper bound** of the full 50 for the standard DTO. (Note we still use 50 for the final sampling once the optimal $\boldsymbol x_0^*$ has been found)
> > >
> > > | Property                | $\alpha$          | $\Delta \varepsilon$ | $\varepsilon_{\mathrm{HOMO}}$ | $\varepsilon_{\mathrm{LUMO}}$ | $\mu$ | $C_v$                                                |
> > > |-------------------------|-------------------|----------------------|-------------------------------|-------------------------------|-------|------------------------------------------------------|
> > > | DTO (1-step)            | 13.049            | $989 \times 10^{12}$ | 681                           | 86.512                        | 1.666 | 15.144                                               |
> > > | DTO (2-step)            | 6.113             | 1359                 | 666                           | 1199                          | 1.533 | 3.757                                                |
> > > | DTO (4-step)            | 6.115             | 1294                 | 668                           | 1190                          | 1.406 | 2.829                                                |
> > > | DTO (8-step)            | 4.549             | 1070                 | 608                           | 1078                          | 1.247 | 2.594                                                |
> > > | DTO (16-step)           | 3.454             | 817                  | 608                           | 939                           | 1.177 | 2.003                                                |
> > > | DTO (32-step)           | 2.912             | 750                  | 410                           | 666                           | 0.721 | 1.566                                                |
> > > | DTO (40-step)           | 2.384             | 625                  | 372                           | 556                           | 0.719 | 1.425                                                |
> > > | DTO (50-step)                     | 1.404             | 401                  | 176                           | 373                           | 0.372 | 0.866                                                |
> > >
> > > Thus our theoretical analysis is demonstrated as we converge towards to the **upper bound on performance** within our design axes, *i.e.*, DTO with the full 50 steps.
> > >
> > > We believe that this in conjunction with our results in **Appendix I.2, Table 5** helps illustrate that we can the ability of this analysis to exchange between compute costs and accuracy of gradients.
> > >
> > > We hope that these answers clarify any concerns about the empirical portion of our work. We would be happy to discuss further and answer any more questions about our work.

---

> > > > ### Author Response · Authors · 2025-08-09
> > > >
> > > > With regards to the point raised here:
> > > > > I appreciate the values of theoretical papers and if advantages over existing guidance methods are more clear (speed, robustness, performance, etc), the paper will be stronger.
> > > >
> > > > We would like to share our additional ablations on the high-dynamic range (HDR) reconstruction *non-linear* inverse image problem.
> > > >
> > > > **Table 3.** Results on HDR inverse image problem of greedy strategy with varying number of discretization steps.
> > > >
> > > > | **Method** | **PSNR** $(\uparrow)$ | **SSIM** $(\uparrow)$ | **LPIPS** $(\downarrow)$ | **FID** $(\downarrow)$ |
> > > > |:-----------|:-------:|:-------:|:-------:|:-------:|
> > > > | DAPS | ${2 7 . 1 2}_{ \pm 3.53}$ | ${0 . 7 5 2}_{ \pm 0.041}$ | ${0 . 1 6 2}_{ \pm 0.072}$ | 42.97|
> > > > | DPS | ${22.73}_{ \pm 6.07}$ | ${0.591}_{ \pm 0.141}$ | $0.264_{ \pm 0.156}$ | 112.82|
> > > > | RED-diff | $22.16_{ \pm 3.41}$ | $0.512_{ \pm 0.083}$ | ${0.258}_{ \pm 0.089}$ | ${108.32}$|
> > > > | Greedy (Euler) | $25.07_{\pm 4.25}$ | $0.776_{\pm 0.126}$ | $0.173_{\pm 0.070}$ | 43.25|
> > > > | Greedy (2-step Euler) | $26.32_{\pm 4.34}$ | $0.802_{\pm 0.111}$ | $0.173_{\pm 0.065}$ | 38.64|
> > > > | Greedy (3-step Euler) | $27.17_{\pm 4.21}$ | $0.820_{\pm 0.096}$ | $0.154_{\pm 0.062}$ | 36.07|
> > > > | Greedy (4-step Euler) | $27.89_{\pm 4.10}$ | $0.828_{\pm 0.092}$ | $0.151_{\pm 0.061}$ | 36.94|
> > > > | Greedy (5-step Euler) | $\mathbf{28.27} _{\pm 4.01}$ | $\mathbf{0.831}_{\pm 0.088}$ | $\mathbf{0.149}_{\pm 0.059}$ | **35.35**|
> > > >
> > > > We notice that:
> > > > * The Greedy (5-step Euler) performs very well beating the SOTA DAPS (CVPR 2025) algorithm, even the 2-step and 3-step perform variants as well or better.
> > > > * Increasing the number of discretization steps (unsurprisingly as predicted by our theory) leads to better performance.
> > > > * Interestingly the standard deviation (reported as the subscript plusminus term) decreases as well with the results becoming more consistent.
> > > >
> > > > We thank reviewer RAY9 for their comments and interest in making our empirical results stronger.

---

> > > > > ### Comment · Reviewer_RAY9 · 2025-08-09
> > > > > **Thanks for your reply**
> > > > >
> > > > > I appreciate the theoretical efforts of this paper and the detailed explanation from the authors. I will update my score accordingly.

---

### Official Review · Reviewer_XpGA · 2025-07-03

**Clarity:** 3
**Significance:** 3
**Originality:** 3
**Rating:** 5
**Confidence:** 4

**Summary:**

This paper presents a novel and elegant theoretical framework that unifies two major families of training-free guided generation: posterior guidance (e.g., DPS) and end-to-end guidance (e.g., D-Flow). The central contribution is the formal demonstration that posterior guidance can be viewed as a "greedy," first-order Euler approximation of the more computationally expensive end-to-end guidance.

The authors provide a rigorous mathematical analysis to support this claim, showing the equivalence of the posterior guidance gradient to a single explicit Euler step in a DTO scheme and to the first iteration of a fixed-point method for an implicit Euler discretization in an OTD scheme. Furthermore, the paper provides a formal error analysis bounding the difference between the "greedy" gradient and the "true" gradient from end-to-end backpropagation.

Building on this unified view, the authors propose a framework, suggesting that higher-order numerical solvers can be used to interpolate between the two guidance families, offering a potential trade-off between computational cost and gradient accuracy.

**Questions:**

Stabilty of results: In Table 2, the euler results, seem very unstable, can the authors confirm that this is true and not originated from some bug?

**Ethical Concerns:**

["NO or VERY MINOR ethics concerns only"]

**Final Justification:**

I had some concerns, they were addressed during the rebuttal.

**Limitations:**

i dont believe there are potential negative societal impact

**Quality:**

3

**Strengths And Weaknesses:**

**Strengths**

To the best of my knowldege, the unification of posterior and end-to-end guidance is novel. The "greedy" perspective is intuitive, and likely to have a good view of the field. The theoretical claims seem valid and are supported by the proofs in the appendix (I didnt fully validated the proofs). Lastly, the proposed framework is a native extenstion of the theory.

**Weaknesses**

* Explore more solvers: I wonder how the results will look like with higher-order solvers. Does the trend being maintined?
* Limited scope of experiments: I think that the experiments can benefit from a deeper analysis of task complexity vs. method being used.
* Limitations: I couldnt find any refernfce to limitation of the work.

I believe the theoretical contribution of this paper is strong and the community will benefit from this work.

---

> ### Author Rebuttal · Authors · 2025-07-30
>
> We thank reviewer XpGA for their interest in our work and thoughtful feedback. We are encouraged that they find the unification framework to be novel and supported by strong theoretical contributions. We are especially excited that reviewer XpGA believes that *"the community will benefit from this work."* We now address the questions and suggestions raised by reviewer XpGA below.
>
> > Explore more solvers: I wonder how the results will look like with higher-order solvers. Does the trend being maintained?
>
> We ran some more experiments for controlled molecule generation which we summarize in the table below.
>
> | Property                | $\alpha$          | $\Delta \varepsilon$ | $\varepsilon_{\mathrm{HOMO}}$ | $\varepsilon_{\mathrm{LUMO}}$ | $\mu$ | $C_v$                                                |
> |:-------------------------|:-------------------:|:----------------------:|:-------:|:-------:|:-------:|:------------------------------------------------------:|
> | Greedy (Euler)          | 11.282            | 1265                 | 725                           | 1092                          | 1.559 | 6.469                                                |
> | Greedy (midpoint)       | 5.313             | 1196                 | 599                           | 1057                          | 1.417 | 2.967                                                |
> | Greedy (2-step Euler) | 5.377             | 1275                 | 560                           | 1204                          | 1.563 | 2.975                                                |
> | Greedy (3-step Euler)   | 5.098             | 1152                 | 600                           | 1152                          | 1.384 | 3.229                                                |
> | Greedy (5-step Euler)   | 4.177             | 1083                 | 571                           | 939                           | 1.328 | 2.332                                                |
> | DTO                     | 1.404             | 401                  | 176                           | 373                           | 0.372 | 0.866     |
> | EquiFM                  | 9.525             | 1494                 | 622                           | 1523                          | 1.628 | 6.689                                                |
> | Lower bound             | 0.10              | 64                   | 39                            | 46                            | 0.043 | 0.040                                                |
>
> We note that generally, taking more discretization steps resulted in a more *useful* gradient. This corroborates with recent work which explored taking two Euler steps [1] (*N.B.*, they approach the problem from a different perspective and use different terminology, but the equivalence is valid).
> In addition to our discussion on using Runge-Kutta 4 in **Appendix H.2** we also tried Ralston's third order method. However, we found similar issues.
> Intuitively, this make sense as by **Theorem 6.1** we found that the error of the gradient lies within the $\mathcal O(h^{\alpha + 1})$ neighborhood. Thus for large step sizes (like in the beginning of posterior guidance) such methods are *likely* to be unsuitable.
> For future work one could explore a more clever scheme using a mix of different order solvers. We discuss this more in **Appendix J.2.**
>
> > Limited scope of experiments: I think that the experiments can benefit from a deeper analysis of task complexity vs. method being used.
>
> We agree that we could have discussed this part better. The main analysis for these two aspects lies within **Theorem 5.1**, **Proposition 5.2**, and **Theorem 6.1**.
>
> **1.** In **Theorem 5.1** and **Proposition 5.2** we show that the greedy guidance strategy results in a covariance projection of the loss function; however, this projection is weaker and less significant as $t \to 1$. Thus for later time steps the greedy gradient is more likely to push samples off the data manifold.
> We posit this is why there was little stability concerns for the inverse image problems; whereas the the controlled molecule generation problem is less resilient to being pushed away from the data manifold. This makes intuitive sense as the very small changes could result in invalid molecules.
> This corroborates with the analyses of prior work [2].
>
> **2.** Then in **Theorem 6.1** we show that the accuracy of the gradients is bound by the local truncation error of the underlying numerical scheme; however, for large step sizes higher-order solvers may be *less* suitable.
> This however, depends on the the straightness of the underlying ODE, if the true underlying ODE is perfectly straight (highly idealized, but presented for illustrative purposes) then all the schemes would have no error.
>
> This analysis is further supplemented by an additional corollary of **Theorem 6.1** and **Proposition 5.2** which we state below.
>
> **Corollary.** *Consider the standard affine Gaussian probability paths model trained to zero loss. Let $\boldsymbol \Phi$ be an explicit Runge-Kutta solver of order $\alpha > 0$ of a flow model with flow $\Phi_{s,t}^\theta(\boldsymbol x)$. The Gateaux differential of $\boldsymbol x$ at some time $t \in [0, 1]$ in the direction of gradient $\nabla_{\boldsymbol x} \mathcal L \left(\boldsymbol \Phi_{t,1}(\boldsymbol x)\right)$ is given by*
>
> $\begin{equation}
> \delta_{\boldsymbol x}^{\boldsymbol \Phi} \Phi_{t,1}^\theta(\boldsymbol x) = -\nabla_{\boldsymbol x} \Phi_{t,1}^\theta(\boldsymbol x) \nabla_{\boldsymbol x} \boldsymbol \Phi_{t,1} (\boldsymbol x)^\top \nabla_{\boldsymbol x_1}\mathcal L(\boldsymbol x_1)
> \end{equation}$
>
> This corollary provides similar analysis tools for studying the higher-order solvers.
>
> > Limitations: I couldn't find any reference to limitation.
>
> We discuss the limitations in **Appendix J.2** (page 42 of the submission).
>
> > Stability of results: In Table 2, the euler results, seem very unstable, can the authors confirm that this is true and not originated from some bug?
>
> We discuss the instability more in **Appendix H.2**, but to summarize it relates to **Proposition 5.2** and the greedy gradient being pushed off the data manifold for $t \gg 0$. Additionally, our code is provided in the supplementary material.
>
> [1] Moufad, B., Janati, Y., Bedin, L., Durmus, A. O., Douc, R., Moulines, E., and Olsson, J. (2025).
> “Variational Diffusion Posterior Sampling with Midpoint Guidance”. In: The Thirteenth Interna-
> tional Conference on Learning Representations. URL: https://openreview.net/forum?id=
> 6EUtjXAvmj
>
> [2] Ben-Hamu, H., Puny, O., Gat, I., Karrer, B., Singer, U., and Lipman, Y. (2024). “D-Flow: Differ-
> entiating through Flows for Controlled Generation”. In: Forty-first International Conference on
> Machine Learning. URL: https://openreview.net/forum?id=SE20BFqj6J
>
> ## Closing remarks
> We thank reviewer XpGA for their questions, which gave us the opportunity to clarify  our contributions and improve our work.
> It is our hope that our answers fully address all the important questions raised by the reviewer, and we are happy to answer any additional questions and consider further suggestions.
>
> We kindly ask the reviewer to consider increasing their score if our responses address their concerns satisfactorily.

---

> > ### Comment · Reviewer_XpGA · 2025-08-05
> >
> > I thank the reviewers for taking the time to reply. I will keep my positive score.

---

### Note · Authors · 2025-08-13

We thank the reviewers for their feedback and advice on how to improve our paper. We provide a brief summary of the discussion period.

### Theoretical contributions
---
We are happy that all the reviewers appreciated our *core theoretical contributions*, *e.g.*, "The theoretical claims seem valid and are supported by the proofs in the appendix" (reviewer XpGA), "I appreciate the theoretical efforts of this paper and the detailed explanation from the authors" (reviewer RAY9), "The paper's primary strength lies in the theoretical framework that unifies posterior guidance and end-to-end guidance" (reviewer oWM7), and "I appreciate the rigorous theory" (reviewer Hyh7).

### Experimental contributions
----
Incorporating the suggestions and advice from the reviewers we ran *a number* of additional ablations to better illustrate our core theoretical claims.
To summarize during the discussion period we:
* Performed additional experiments on posterior guidance with multiple sampling steps ranging (up to 5) on QM9 which confirms our theoretical analysis.
* Ran ablations on a DTO-esque guidance scheme for finding the optimal $\boldsymbol x_0$ on QM9 with a variety of discretization steps showing that as the number of steps increases we tend towards the upper bound of the full DTO scheme.
* We performed a number of experimental runs on the high-dynamic range (HDR) *non-linear* inverse problem, achieving SOTA results with discretization steps > 2. This is in spite of not using more advanced techniques like noise annealing (or other SDE related techniques) which other approaches made use of (as this was outside our design axes).

Due to the helpful suggestions from the reviewers our empirical contributions are now significantly more compelling and further supplement our core theoretical contributions.

### Miscellanea
---
We will add additional clarity to the paper through the addition of:
* pseudocode,
* algorithms, and
* additional experimental illustrations as alluded to through our tables.

### Closing
---
We again thank the reviewers for their time and engaged conversations throughout the discussion period. We hope this summary is helpful for the AC and the reviewers as they have their internal discussions.

Best,

The authors

---

### Decision · Program_Chairs · 2025-09-17

**Decision:**

Accept (poster)

**Comment:**

The paper presents a unifying framework for posterior guidance and end-to-end guidance, two techniques used in training-free guided generation in flow and diffusion models. The main contribution is that posterior guidance can be viewed as a greedy first-order Euler approximation of end-to-end guidance. This sheds light on potential trade-offs between computational cost and gradient accuracy.
The theory is validated on inverse image problems and property-guided molecular generation.
During the rebuttal, the authors addressed the reviewers' questions by providing more experiments and clarifications.
While there was some discussion around the fact that the results do not outperform SOTA, as pointed out by reviewer Hyh7 in particular, the authors have clarified that their point was not SOTA but to explain the connection between posterior and end-to-end guidance, which the experiments address.